# Descending networks transform command signals into population motor control

Jonas Braun[1,2], Femke Hurtak[1,2], Sibo Wang-Chen[1] & Pavan Ramdya[1✉]

To convert intentions into actions, movement instructions must pass from the brain to downstream motor circuits through descending neurons (DNs). These include small sets of command-like neurons that are sufficient to drive behaviours[1]—the circuit mechanisms for which remain unclear. Here we show that command-like DNs in *Drosophila* directly recruit networks of additional DNs to orchestrate behaviours that require the active control of numerous body parts. Specifically, we found that command-like DNs previously thought to drive behaviours alone[2–4] in fact co-activate larger populations of DNs. Connectome analyses and experimental manipulations revealed that this functional recruitment can be explained by direct excitatory connections between command-like DNs and networks of interconnected DNs in the brain. Descending population recruitment is necessary for behavioural control: DNs with many downstream descending partners require network co-activation to drive complete behaviours and drive only simple stereotyped movements in their absence. These DN networks reside within behaviour-specific clusters that inhibit one another. These results support a mechanism for command-like descending control in which behaviours are generated through the recruitment of increasingly large DN networks that compose behaviours by combining multiple motor subroutines.

Animals, including humans, are capable of generating a remarkable variety of behaviours ranging from stereotyped movements—such as escape reflexes needed to rapidly evade a predator—to more elaborate actions such as navigating over unpredictable, rugged terrain. All of these behaviours require the active control of multiple joint degrees of freedom by motor circuits in the vertebrate spinal cord or invertebrate ventral nerve cord (VNC). In addition to the important role of spinal circuits in the execution of movements, a relatively small population of DNs projecting from the brain to motor circuits regulate the selection, initiation and online steering of many behaviours.

We still lack mechanistic understanding of how DNs as a population drive and coordinate behaviours, in part due to the technical difficulty of comprehensively recording and manipulating DNs in behaving mammals: there are more than 1 million in the human pyramidal tract[5] and approximately 70,000 in the mouse corticospinal tract[6]. By contrast, the adult fly, *Drosophila melanogaster*, has approximately 1,300 DNs linking the brain to motor centres in the VNC[7]. Despite this numerical simplicity, flies can generate various complex behaviours including legged locomotion[8], flight[9], courtship[10] and aggression[11]. Several tools facilitate the investigation of descending control in the fly including connectomes for quantifying the synaptic connectivity of every neuron in the brain[7] and VNC[12,13], as well as genetic tools for repeatedly targeting identified descending neurons[14,15] across individual animals for experimental recordings (electrophysiological[16] or optical[17]) and manipulations (activation[18] or silencing[19]).

One notable discovery derived using these tools is that, despite the abundance of DNs in the fly brain, artificial activation of pairs of 'command-like' DNs (comDNs) can be sufficient to drive a complete behaviour (but not also necessary as is required to be considered 'command' neurons[20]). For example, DNs have been identified whose artificial activation trigger forwards walking[3], grooming[4,21], backwards walking[2], escape[16], egg-laying[22] and components of courtship[23,24]. The capacity of some DNs to act as command-like neurons appears to be general across species including invertebrates[25,26] and mammals[27]. Command-like descending control has also been leveraged to design controllers for robots[28].

The concept of command-like control raises a fundamental question regarding to what extent each pair or small set of DNs drives a distinct action. Several lines of evidence have suggested that this is unlikely. Most directly, for many DNs, sparse optogenetic activation does not clearly and reliably drive a coordinated behaviour[18]. In addition, previously, we observed the co-activation of many DNs during walking[29], and others have shown that a group of 15 DNs can modulate wing beat amplitude[30] and that the activation of individual DNs has a lower probability of eliciting take-off than the co-activation of multiple DNs[31]. Furthermore, beyond controlling kinematics, DNs can also be neuromodulatory[32,33]. All of these observations imply that DN control of a given behaviour rather than being via one class of DNs conveying a simple but reliable drive signal could instead depend on multiple classes of DNs working together as a population. In this model, individual DNs would represent single dimensions of a high-dimensional control signal, which are combined to construct complete behaviours from simpler motor primitives.

[1]Neuroengineering Laboratory, Brain Mind Institute & Interfaculty Institute of Bioengineering, EPFL, Lausanne, Switzerland. [2]These authors contributed equally: Jonas Braun, Femke Hurtak. ✉e-mail: pavan.ramdya@epfl.ch

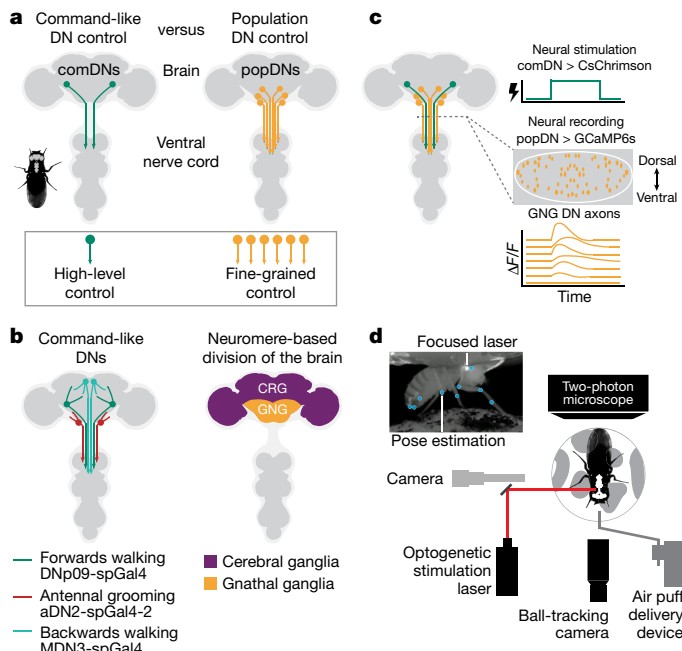

**Fig. 1 | Optical approach to probe the relationship between comDNs and popDNs in behaving animals. a**, Schematic of the *Drosophila* nervous system showing a pair of DNs that project from the brain to motor circuits in the VNC (left). Activation of small sets of comDNs (green) can drive complete behaviours. Thus, comDNs are thought to send simple, high-level control signals to the VNC, where they are transformed into complex, multi-joint movements. However, larger popDNs (orange) are also known to become active during natural behaviours (right). Therefore, in another model, individual DNs contribute to complex behaviours by sending low-level signals that control the fine-grained movements of individual or sparse sets of joints. **b**, We stimulated three sets of comDNs to elicit three distinct behaviours: forwards walking (DNp09, green)[3,14], antennal grooming (aDN2, red)[4] and backwards walking (MDN, cyan) (left)[2]. DN cell body locations are schematized. Two coarse subdivisions of the adult *Drosophila* brain are the cerebral ganglia (CRG; previously known as the supraoesophageal ganglion) and the GNG (also known as the suboesophageal ganglion) (right)[59]. We recorded from DNs within the GNG, which houses most DNs[14]. **c**, We recorded neural activity in the axons of GNG DN populations (orange) during optogenetic stimulation of different sets of comDNs (green). The grey dashed line denotes a coronal section region of interest in the thoracic cervical connective illustrating DN axon cross-sections (orange ellipses). **d**, A system for recording behaviour, GNG DN neural activity[29] and optogenetically stimulating comDN axons in the neck connective (schema not to scale). The inset shows a camera image of a fly with focused laser light on its neck. Superimposed on the camera image are pose estimation key points (light blue).

At first glance, these two models—comDN versus population-based DN behavioural control—appear to be conflicting. However, we can envision at least two scenarios in which they can be unified. First, comDNs or non-comDNs may simply target different downstream motor circuits (in the spinal cord or VNC) that can or cannot generate complete behaviours, respectively. Alternatively, comDNs may be privileged in that they can recruit additional DN populations to drive complete behaviours. This latter possibility is supported by the fact that, in addition to projecting to the VNC, 85% of all DNs have axon collaterals and thus may engage one another in the gnathal ganglia (GNG) of the brain, a location where most DNs are found[14].

Here we investigated the degree to which known comDNs interact with other DNs in the brain to generate complete behaviours. When optogenetically activating three sets of comDNs, we observed the co-activation of additional DN populations in the GNG. This functional recruitment covaries with and can be explained at least in part by monosynaptic excitatory connections between comDNs and downstream DN

networks. Through decapitation experiments, we found that behaviours triggered by strongly connected DNs require the engagement of larger DN networks, whereas comDNs engaging smaller networks do not. We then identified nine additional sets of comDNs that allowed us to experimentally test and validate this model of DN recruitment for behavioural control. Finally, we performed a comprehensive analysis of all DN–DN interconnectivity in the brain and found that DN networks form predominantly excitatory clusters associated with distinct actions that mutually inhibit one another. In summary, these findings suggest a new framework that can reconcile the two dominant models of DN control: comDNs drive complete behaviours by recruiting additional downstream DN populations, which combine and coordinate multiple motor subroutines.

## From comDNs to DN populations

We set out to explore the relationship between two prominent models for how DNs control behavioural kinematics. In the first model, the artificial activation of a few comDNs—a simple high-level descending signal—engages downstream motor circuits in the VNC to drive a complete behaviour (for example, walking or grooming) (Fig. 1a, left 'comDNs'). In the second model, a larger population of DNs must become co-active to orchestrate a given behaviour. Each DN within this population would be responsible for controlling or modulating a particular movement or motor primitive. The combined activity of the entire population would yield a complete behaviour (Fig. 1a, right 'popDNs').

These two scenarios can be distinguished by the degree to which activation of comDNs further co-activates other DNs. We tested this using an all-optical experimental strategy in the adult fly *D. melanogaster*. We activated three sets of comDNs that drive a wide range of behaviours including forwards walking (DNp09 (ref. 3), green), antennal grooming (aDN2 (ref. 34), red) or backwards walking (MDN3 (ref. 2), cyan) (Fig. 1b, left) via cell-specific expression of the light-activated ion channel CsChrimson[35] (comDN-*spGAL4 > UAS-CsChrimson*; Extended Data Fig. 1a,d) and laser light stimulation. Simultaneously, we recorded the activity of DN populations by expressing the genetically encoded calcium indicator GCaMP6s[36] (*Dfd-LexA > LexAOp-opGCaMP6s*), in the GNG, the most caudal region of the fly brain (Fig. 1b, right 'GNG DNs', and Extended Data Fig. 1b), but not in our comDNs (Extended Data Fig. 1c). To further restrict our neural recordings to DNs, we performed two-photon microscopy of DN axons passing through the thoracic cervical connective[29] (Fig. 1c). We further increased the specificity of comDN optogenetic activation by restricting stimulation of DN axons to the neck connective (Fig. 1d, red, and Extended Data Fig. 1e,f).

## ComDNs recruit additional DNs

Using these tools, we examined whether additional DNs in the GNG might be recruited upon optogenetic activation of comDNs. We used an open-loop trial structure in which 5-s periods of optogenetic stimulation were interleaved with 10-s periods of spontaneous animal behaviour. This approach elicited robust behavioural responses, which we quantified through trial averaging (Fig. 2a). We observed a clear increase in GNG DN activity during the stimulation of any of the three sets of comDNs in individual animals: DNp09, aDN2 and MDN (Supplementary Video 1) (Fig. 2b–d). This result was also consistent across multiple animals (Fig. 2e,f). We did not observe pronounced activation of GNG DNs in control animals lacking an *spGAL4* transgene (Fig. 2b–f, rightmost, and Supplementary Video 1). Thus, GNG DN populations become active due to comDN stimulation as, for all three sets of comDNs tested, the number and fraction of GNG DNs activated were significantly higher than for control animals (Fig. 2g,h; $P = 0.018$ (DNp09), $P = 0.040$ (aDN2) and $P = 0.008$ (MDN)).

We found that GNG DNs were recruited in a spatially distinct manner across the cervical connective depending on which class of comDNs

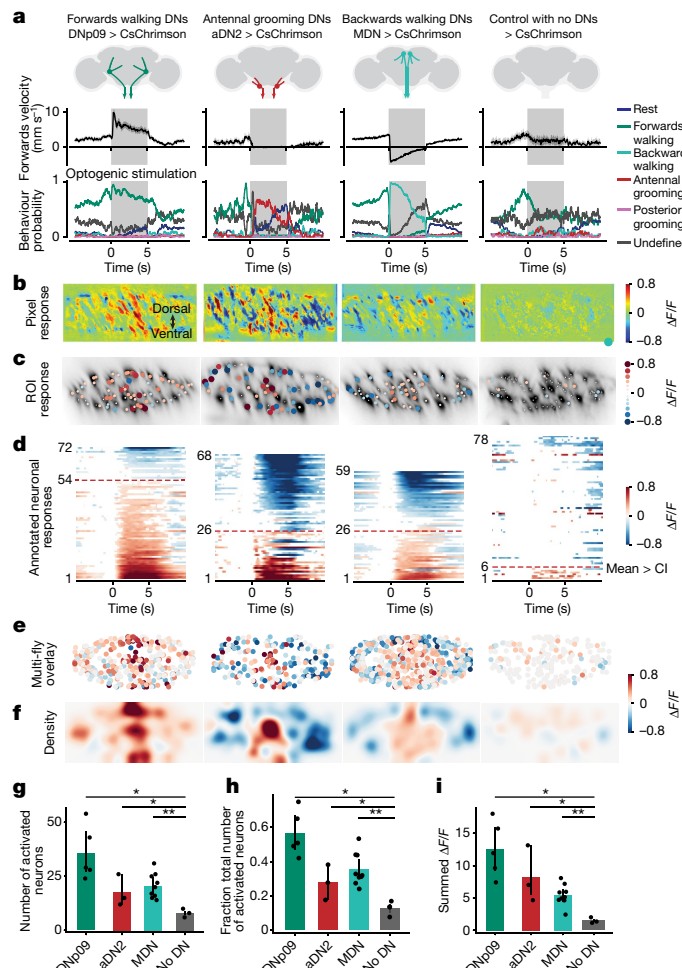

**Fig. 2 | Activation of comDNs recruits larger, distinct DN populations.**
Optogenetic stimulation of comDNs: DNp09 (forwards walking, *n* = 5 flies, 120
stimulation trials), aDN2 (antennal grooming, *n* = 3 flies, 34 trials) and MDN
(backwards walking, *n* = 9 flies, 271 trials). Control: no DN expression (*n* = 3 flies,
47 trials). **a**, Forwards walking velocities (top) and the probability of classified
behaviours (bottom) during optogenetic stimulation (grey bar). **b**, Images
illustrating GNG DN population activity upon comDN stimulation. For each,
one representative animal is shown (as in Supplementary Video 1; *n* = 33, 10, 97
and 10 trials for DNp09, aDN2, MDN and control flies, respectively). The same
flies are shown in panels **c**,**d**. **c**, Single-neuron responses to DN stimulation.
Circles are scaled or colour coded to represent the maximum change in
fluorescence (normalized Δ*F/F*) of one detected DN axon or region of interest
(ROI). The small white dots indicate responses smaller than the 95% CI of the
trial mean. **d**, Trial-averaged single ROI responses across time, ordered by
response magnitude. Response magnitude is colour coded or white if smaller
than the 95% CI. The red dashed line indicates the number of activated ROIs
(that is, positive response larger than the 95% CI). **e**,**f**, A registered overlay (**e**)
or density visualization (**f**) of the data from multiple flies analysed as in **c**. The
number of flies or trials is identical to **a**. **g**–**i**, Statistical comparison of the number
of activated ROIs (that is, red dashed line in **d**) (**g**), the fraction of activated ROIs
(that is, divided by the number of visible ROIs) (**h**) and the strength of activation
(that is, the sum of the normalized Δ*F/F* for positively activated neurons) (**i**) using
two-sided Mann–Whitney *U*-tests (*n* as in **a**; *P* values for each comparison to
control: DNp09 = 0.018, aDN2 = 0.040 and MDN = 0.008). The shaded areas in
**a** and the error bars in **g**–**i** represent 95% CI of the mean. \*\*\**P* < 0.001, \**P* < 0.05.

was activated (Fig. 2e,f). Stimulation of forwards walking (DNp09) and
antennal grooming (aDN2) increased the activity of DNs localized in dis-
tinct regions of the medial cervical connective: the entire dorsal–ven-
tral axis for forwards walking, and the medial and ventral connective for
grooming. Activation of backwards walking (MDN) led to weaker GNG

DN recruitment localized to the medial connective. We quantified the
strength of GNG DN recruitment as the summed responses of neurons
that were positively activated during optogenetic stimulation (Fig. 2i),
a quantity that was significantly higher for comDN stimulation than for
controls (*P* = 0.018 (DNp09), *P* = 0.040 (aDN2) and *P* = 0.008 (MDN)). In
addition, we observed a recruitment gradient among comDNs: DNp09
stimulation resulted in very strong recruitment of GNG DNs, aDN2 in
slightly weaker recruitment and MDN the weakest.

Co-activation of GNG DNs by optogenetic stimulation may be
non-ethological rather than reflecting what is seen during natural
behaviour. For example, when animals groom their antennae to remove
debris, aDN2 will have a specific firing rate with a specific temporal
activity pattern. This may not be well reflected by the potentially high
firing rate and relatively static temporal activity pattern driven by
optogenetic stimulation of the same neurons. Thus, an unusually high
firing rate might be responsible for recruiting other DNs. To address this
concern, we compared the activity of GNG DN populations in the same
individual animals during both optogenetic stimulation and the corre-
sponding natural behaviour. Specifically, we compared neural activity
during both DNp09 stimulation and bouts of spontaneous forwards
walking (Extended Data Fig. 2a and Supplementary Video 2), aDN2
stimulation and air-puff-induced anterior grooming (Extended Data
Fig. 2b and Supplementary Video 2), as well as MDN stimulation and
spontaneous backwards walking on a cylindrical treadmill (Extended
Data Fig. 2c and Supplementary Video 2). In each case, we observed
that populations of GNG DNs were recruited during both optogenetic
stimulation and natural behaviour. For backwards walking, these pat-
terns were largely similar across optogenetic and natural conditions
(Extended Data Fig. 2c). However, for forwards walking (Extended Data
Fig. 2a) and, to a lesser extent, for anterior grooming (Extended Data
Fig. 2b), there were some differences. DNp09 stimulation consistently
and strongly activated a small subset of DNs located in the medial–
dorsal and medial–ventral connective, which were not active during
spontaneous forwards walking (Extended Data Fig. 2d–f). However, the
remaining largest fraction of DNs were active in a similar manner
during optogenetic DNp09 stimulation and during spontaneous for-
wards walking (Extended Data Fig. 2e, white region).

We next considered how comDNs might recruit additional GNG DNs.
On the one hand, it could be through connections within the brain. On
the other hand, it could be indirectly via the VNC. For example, a DN
might target (or indirectly drive) an interneuron in the VNC, which in
turn ascends to the brain and engages GNG DNs. To determine whether
DN recruitment can arise from brain connections alone, we resected
the VNC in the anterior-most prothoracic (T1) neuromere to sever
axonal projections of DNs to the VNC and of ascending neurons to
the brain. We then performed functional imaging of GNG DNs during
optogenetic stimulation of DNp09 (Extended Data Fig. 3a and Supple-
mentary Video 1) and observed that GNG DNs were still co-activated
in T1-severed animals (Extended Data Fig. 3b–e) but not in control
flies without a DN driver (Extended Data Fig. 3f–j). This confirms that
connections in the brain can be sufficient for DN recruitment.

Together, these data show that optogenetic stimulation of com-
DNs leads to the recruitment of many additional DNs in a manner that,
particularly for backwards walking and antennal grooming, is similar
to DN population activity during natural behaviour.

## ComDNs connect to DN networks

The functional recruitment of GNG DNs by comDNs could arise from
various circuit mechanisms in the brain. Broadly speaking, it might
either result from direct, monosynaptic excitatory connections or
indirectly via local interneurons. We investigated these possibilities by
examining DN–DN connectivity within the female adult fly brain con-
nectome[7,37,38]. There, we identified our three sets of comDNs—DNp09,
aDN2 and MDN (Fig. 3a)—and all of their downstream partners. We

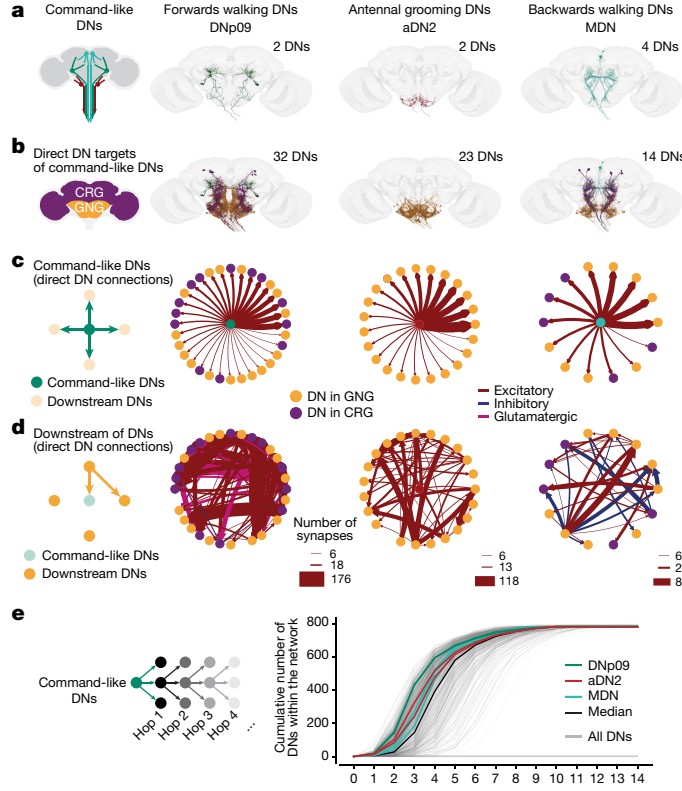

**Fig. 3 | ComDNs connect to other DNs, forming larger DN networks. a,** The neuronal morphologies of three sets of comDNs in the female adult fly brain connectome: DNp09 (left), aDN2 (middle) and MDN (right)[7]. **b,** The location and morphologies of DNs directly (monosynaptically) targeted by comDNs. DNs are colour coded based on their cell body localization in the GNG (orange) or CRG (purple). Command-like neurons are colour coded as in **a. c,** ComDNs form monosynaptic excitatory connections to downstream DN targets. Edge weights reflect the number of synapses as shown in **d,** with consistent scaling across all plots. Edge colours denote whether synapses are excitatory (red), inhibitory (blue) or glutamatergic (pink), which can be excitatory or inhibitory depending on the receptor type[60]. DNs are colour coded as in **b. d,** Network connectivity among downstream DNs shows strong recurrence and minimal feedback to comDNs (only in aDN2). **e,** The cumulative number of downstream DNs that three sets of command-like neurons—DNp09 (green lines; 2 DNs), aDN2 (red lines; 2 DNs), MDN (cyan lines; 4 DNs)—connect to across an increasing number of DN–DN synapses or 'hops'. This is compared with the number of DNs accessible over an increasing number of hops for all DNs (grey lines) and the median of all DNs (black line). Many DNs do not connect to any other DN, and 455 DNs only receive inputs from maximally one other DN, limiting the maximum number of recruited DNs to approximately 800.

found that each comDN has direct, monosynaptic connections to other DNs (Fig. 3b).

On the basis of the predictions from electron microscopy images, our three sets of comDNs are cholinergic[7,39]. Thus, they probably form excitatory connections with downstream DNs (Fig. 3c, red arrows). These connections are predominantly feedforward with only sparse feedback connections for aDN2 (Fig. 3d). By contrast, among their downstream DNs, we observed strong recurrent interconnectivity, including some inhibition (Fig. 3d, blue arrows). Of note, the three sets of comDNs connect to a variable number of downstream DNs, which mirrors their differential recruitment of GNG DNs during our functional imaging experiments (Fig. 2i): those for forwards walking (DNp09) have the most downstream DNs (32), whereas those for antennal grooming (aDN2) have fewer (23) and those for backwards walking (MDN) have the fewest (14). This ordering also holds for polysynaptic connections to downstream DNs (Fig. 3e). These data support a mechanism in which

comDNs engage additional DN populations in the brain via direct excitatory connections.

## Behavioural requirement of DN recruitment

We next asked to what extent the recruitment of additional DN populations is necessary for comDNs to drive complete behaviours. To do this, we needed to stimulate comDNs while preventing the recruitment of additional DN populations. Sensory neurons in the brain provide inputs to help initiate and regulate natural behaviours, whereas DNs are thought to integrate these signals to drive specific motor actions. In this experiment, we aimed to identify which elements of behavioural kinematics result solely from optogenetic stimulation of comDNs alone, without also recruiting sensory inputs to the brain or other downstream DNs in the brain (Fig. 4a, right). We achieved this by studying animals that were carefully decapitated with their exposed necks sealed. Following decapitation, flies can survive and generate behaviours for hours[40]. A less invasive approach—acute optogenetic inhibition of GNG DNs using GtACR1 (ref. 41)—would inhibit only a fraction of all DNs and, when tested, caused animals to groom even at low light intensities (Extended Data Fig. 1g), obstructing analysis of comDN-driven behaviours.

Using this approach, we compared the behaviours of intact and headless animals upon optogenetic activation of comDNs. As for our previous experiments, stimulation of DNp09, aDN2 and MDN in intact animals drove forwards walking, antennal grooming and backwards walking, respectively (Fig. 4b–d, black traces), with no reliable behaviour generated in control animals (Supplementary Video 3) (Fig. 4e, black traces). After decapitating these same animals, we found that the activation of MDN in headless flies still drove backwards walking. This confirms that decapitation does not trivially impair movement generation (Fig. 4d; $P = 0.265$ comparing the backwards walking probabilities of headless versus intact flies). By contrast, decapitation had a different effect on the other two comDNs: DNp09 and aDN2 stimulation in headless animals did not elicit forwards walking (Fig. 4b; $P = 0.006$) or antennal grooming (Fig. 4c; $P = 0.006$), respectively. However, these headless animals could still exhibit behaviours distinct from control animals; optogenetic stimulation of DNp09 and aDN2 in headless flies reliably elicited stereotyped abdomen contraction for DNp09 (Fig. 4f; $P = 0.006$ comparing headless DNp09 versus headless control animals) and front leg approach for aDN2 animals (Fig. 4g; $P = 0.030$ comparing the distance between the tibia–tarsus joint and neck in headless aDN2 versus headless control animals). These observations confirm that DN axons in the VNC alone are capable of activating downstream VNC motor circuits in headless animals and led us to posit that differences in optogenetically driven behaviours between intact and headless flies result from the failure to recruit additional, downstream DN networks in the brain. The fact that functional recruitment of DN populations is necessary for comDNs to drive some behaviours (that is, forwards walking and antennal grooming via DNp09 and aDN2 stimulation, respectively), but not others (backwards walking via MDN stimulation), implies several distinct modes of DN behavioural control that we next set out to explore.

## Network size predicts behavioural necessity

Our results thus far revealed a correlation between three properties of comDNs (Fig. 5a, top): (1) the functional recruitment of other DNs (Fig. 2), (2) the degree of monosynaptic connectivity to downstream DNs (Fig. 3), and (3) the necessity of recruiting downstream DNs to generate complete optogenetically driven behaviours (Fig. 4). Together, these properties suggest that comDNs may lay on a continuum. 'Broadcaster' DNs, such as DNp09, have a large number of downstream DNs that must be recruited to generate behaviours, possibly by combining multiple motor primitives[42,43]. By contrast, 'standalone' DNs, such as MDN, have few or no downstream DNs and may by themselves be

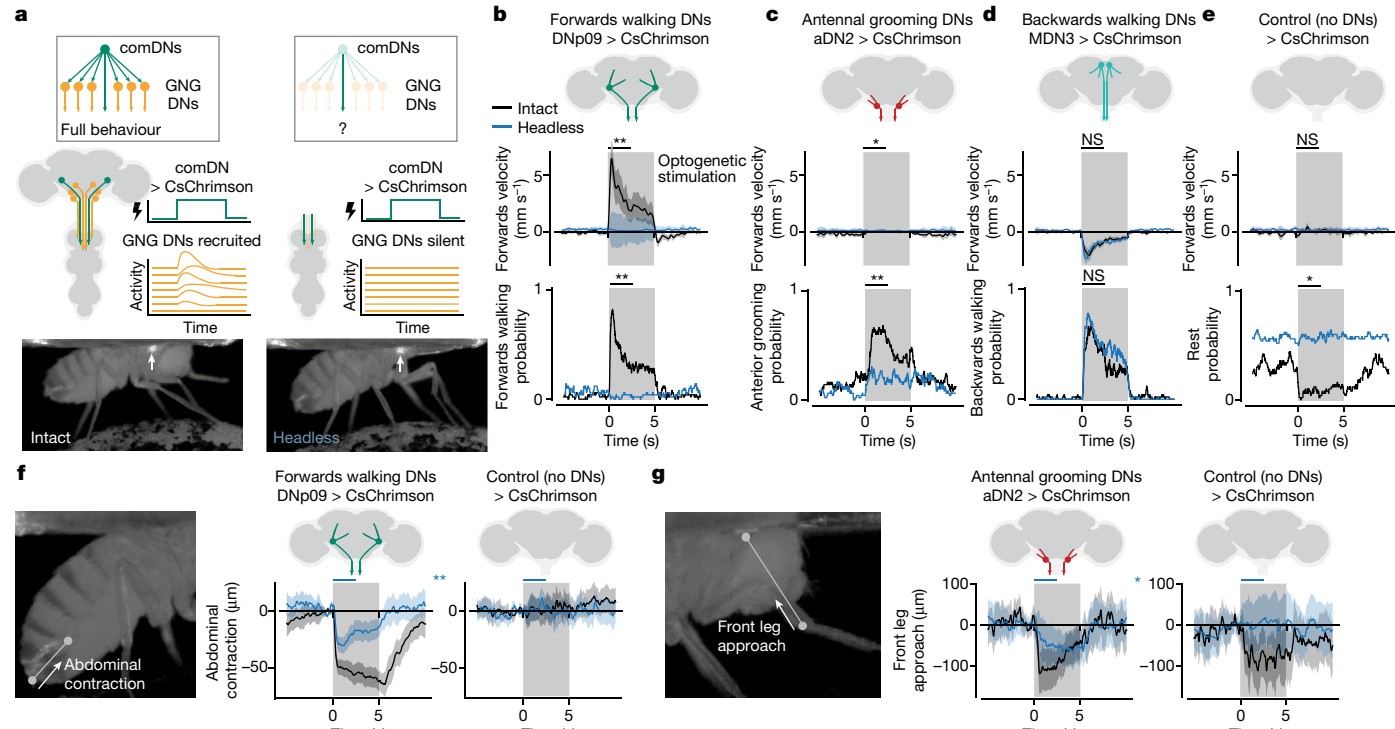

**Fig. 4 | Recruited DN networks are required for forwards walking and grooming, but not for backwards walking. a**, In intact animals (left), activation of a comDN (green) recruits other DNs (orange) and leads to the execution of a complete behaviour. In headless animals (right), the axons of comDNs (green) can still be activated in the VNC. However, other DN axons (orange) cannot be recruited in the brain and remain silent. This comparison between intact and headless animals allows one to isolate the necessity of downstream DN networks to generate complete behaviours. **b**–**e**, Forwards walking velocities and behaviour probabilities for DNp09 (**b**), aDN2 (**c**), MDN (**d**) or control (**e**) flies. Mann–Whitney $U$-tests compare the difference between the means of the first 2.5 s of optogenetic stimulation across intact (black traces) versus headless (blue traces) animals. **f**, DNp09 stimulation in both intact and headless animals

leads to abdominal contraction (change in Euclidian distance between the anal plate and the ventral side of the most posterior stripe). Mann–Whitney $U$-test compares the mean of the first 2.5 s of stimulation (blue bars) for headless DNp09 versus headless control animals (blue traces). **g**, aDN2 stimulation in both intact and headless animals leads to front leg approach (change in Euclidian distance between the front leg tibia–tarsus joint and the neck). Mann–Whitney $U$-test compares the first 2.5 s of stimulation (blue bars) between headless aDN2 and headless control animals (blue traces). All plots in **b**–**g** show data from $n = 5$ flies with 10 trials each (trial mean and 95% CI (shaded area)). Two-sided Mann–Whitney $U$-tests compare the trial mean across different flies. ***$P < 0.001$, **$P < 0.01$, *$P < 0.05$ and not significant (NS) $P > 0.05$. For exact $P$ values, see Supplementary Table 5.

sufficient to drive behaviours that are largely dependent on VNC circuitry alone (Fig. 5a). Thus, for a given comDN, one might be able to predict the behavioural outcome of optogenetic stimulation in intact versus headless animals based on the number of downstream DNs that it is connected to. Specifically, broadcaster or standalone DNs should show, respectively, either a strong or weak degradation of their associated optogenetically driven behaviours following decapitation (Fig. 5a, light blue box).

To test this hypothesis, we examined direct DN–DN connectivity across all DNs in the brain connectome[38] to identify additional broadcaster and standalone DNs. We observed a continuum of interconnectivity for DNs across the brain (Fig. 5b, grey) that was also present for connections to GNG-based DNs specifically (Fig. 5b, orange): a few DNs have dozens of DN partners, whereas hundreds of others have no downstream DN partners. This continuum ranging from well-connected broadcaster DNs to sparsely connected standalone DNs held true even when accounting for both excitatory and inhibitory connections (Extended Data Fig. 4a–c), excitatory connections alone (Extended Data Fig. 4d–f) or inhibitory connections alone (Extended Data Fig. 4g–i). These differences also persisted when accounting for disynaptic connections via another DN (Extended Data Fig. 4b,e,h) or via any other brain interneuron (Extended Data Fig. 4c,f,i).

Our three sets of comDNs lie in the middle of this continuum with higher connectivity than most DNs (median number of connected DNs: all DNs (4), MDN (9), aDN2 (15) and DNp09 (23); Fig. 5b, inset). Of note,

consistent with our model, giant fibre neurons, which are known to drive relatively stereotyped, ballistic escape behaviours in both intact and headless animals[44,45], have only a few DN partners (three and four for the left and right giant fibre neurons, respectively; Fig. 5a, grey circle). We selected an additional nine sets of DNs along this continuum of connectivity (Fig. 5c, squares in colour) based on specific connectivity criteria (see Methods) and the availability of transgenic driver lines for optogenetic stimulation[14,15].

Data from optogenetically stimulating these nine sets of DNs in both intact and headless animals confirmed our predictions: DNs with many downstream DN partners drove behaviours that were lost in headless animals (Extended Data Fig. 5), whereas DNs with few or no downstream DN partners elicited simple, stereotyped movements (for example, abdominal curling and ovipositor extension) that persisted following decapitation (Extended Data Fig. 6). Among broadcasters, this degradation of behaviour was most profound for DNb02, which connects to 20 other DNs (Fig. 5d,e) and drives turning in intact animals. In headless animals, DNb02 stimulation does not elicit turning (Fig. 5f; $P = 0.001$ comparing intact and headless flies), but instead drives flexion of the front legs upon stimulation onset (Supplementary Video 4). This is noticeable as a small spike in forwards velocity in headless animals (Extended Data Fig. 5d). Similarly, for other broadcasters, we observed a loss of backwards retreat in DNp42 (Extended Data Fig. 5a and Supplementary Video 4) and turning in DNa01 (Extended Data Fig. 5c and Supplementary Video 4) and DNa02 (Extended Data Fig. 5e and

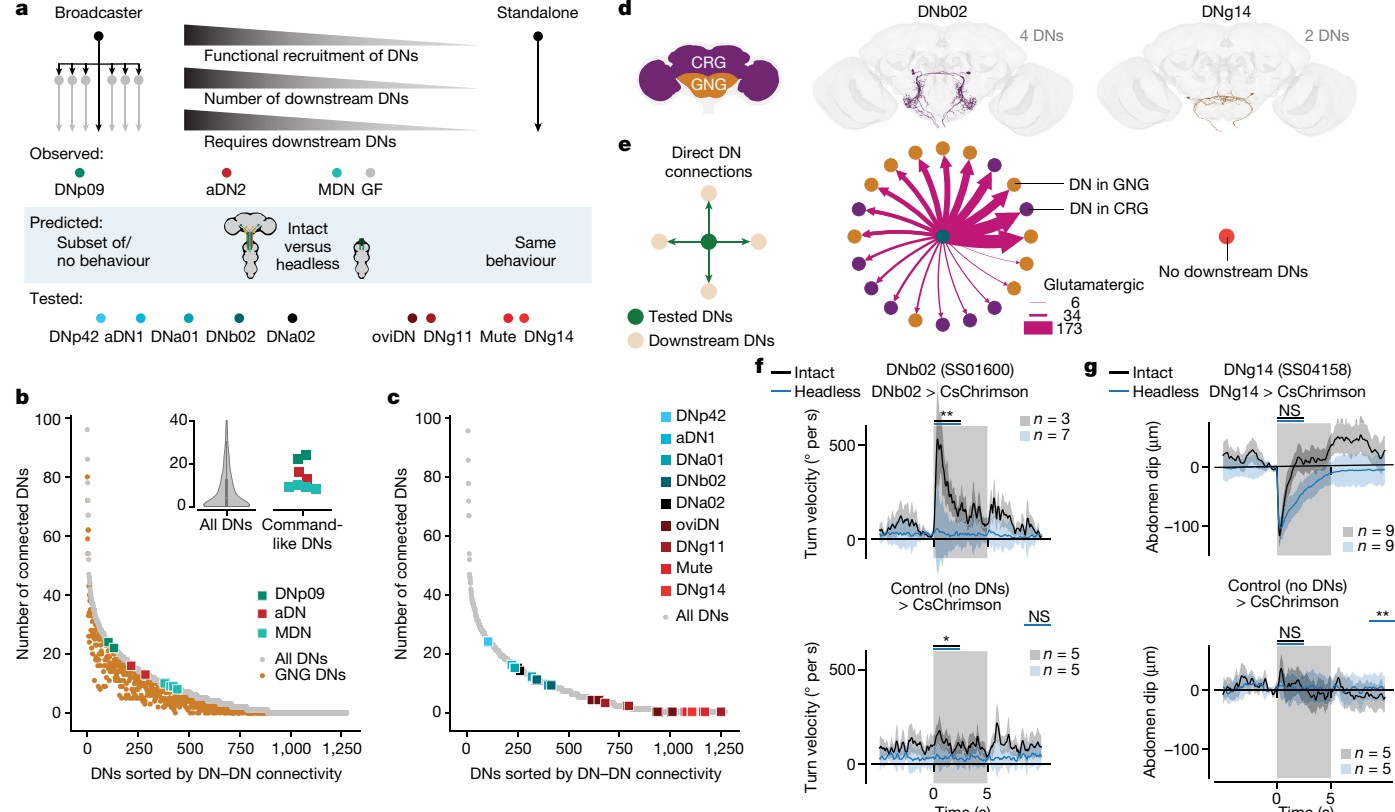

**Fig. 5 | Network connectivity accurately predicts the necessity for downstream DNs to drive behaviour. a**, For the comDNs investigated, three important properties covary in a continuum that spans from broadcaster DNs to standalone DNs. Schematized along this continuum are our three comDNs, giant fibre (GF) neurons and nine additional tested neurons: DNp42, aDN1, DNa01, DNb02, DNa02, oviDN, DNg11, Mute and DNg14. **b**, For each *Drosophila* DN, the total (grey) or GNG-based (orange) number of monosynaptically downstream DNs. ComDNs are colour coded. The inset shows median and 25% and 75% quantiles (left violin plot, *n* = 1,303) comparing all DNs to DNp09, aDN2 and MDN. **c**, The number of DNs directly downstream of nine additional sets of DNs (colour-coded circles as in **a**) for which connectome-based experimental predictions are made. All DNs (grey) shown are as in **b**. **d**, The morphology of two sets of DNs (DNb02 and DNg14) in the female adult fly brain connectome.

**e**, Monosynaptic connectivity for two tested DNs (DNb02 and DNg14). Edge weights denote the number of glutamatergic synapses (pink). **f**, Absolute, undirected turn velocity for DNb02 (top) and control (bottom) animals upon laser stimulation. **g**, Abdomen dipping for DNg14 (top) and control (bottom) animals upon laser stimulation (change in anal plate vertical position). In **f**,**g**, data are shown for intact (black traces) and headless (blue traces) animals. The number of animals is indicated for each condition. Each fly was optogenetically stimulated ten times. Traces show the average and 95% CI across *n* × 10 trials. Two-sided Mann–Whitney *U*-tests comparing the trial mean of intact and headless animals (black bars) or comparing headless experimental with headless control flies (blue bars, between top and bottom plots). \*\**P* < 0.01, \**P* < 0.05 and NS *P* > 0.05. For exact *P* values, see Supplementary Table 5.

Supplementary Video 4) headless animals. aDN1 animals retained only uncoordinated front leg movements following decapitation (Extended Data Fig. 5b and Supplementary Video 4).

Among standalone DNs, the maintenance of stereotyped movements was most clear for DNg14, which do not directly synapse upon any other DN (Fig. 5e). These neurons drive a subtle dip and vibration of the abdomen in both intact and headless animals (Fig. 5g and Extended Data Fig. 6d; *P* = 0.144; Supplementary Video 5). Similarly, for other standalone DNs, in both intact and headless animals, we observed a downward curling of the abdomen in oviDN flies (Extended Data Fig. 6a and Supplementary Video 5), foreleg rubbing in DNg11 flies (Extended Data Fig. 6b and Supplementary Video 5) and ovipositor extension in Mute flies (Extended Data Fig. 6c and Supplementary Video 5). Thus, our experiments on a total of 12 sets of DNs support a model in which the connectivity of a comDN to other DNs is predictive of its necessity for network recruitment to generate behaviour.

## Network clusters correlate with behaviour

Our investigation of the brain connectome revealed that DN–DN connectivity lies on a continuum: a few DNs have very high connectivity

(for example, with more than 80 downstream DNs), whereas 567 (44%) target only two or fewer DNs (Fig. 5b). This overall structure of DN networks has implications for how information flows between neurons, motivating us to examine the large-scale structure of the entire DN network. We compared the DN network derived from the fly brain connectome with a shuffled network having the same number of neurons and interconnections, but with individual connections randomly assigned. We found that the connectivity degree distribution (that is, the distribution of how many other DNs each DN connects to) is dramatically different ($R^2$ = −0.04 comparing connectivity distributions) for real (Fig. 6a, black) versus shuffled (Fig. 6a, red) DN networks. This is largely because very strongly connected DNs (more than 30 partners) and very weakly connected DNs (fewer than 5 partners) only appear in the real DN network but not in the shuffled network. That the original DN network can be fit better by an exponential ($R^2$ = 0.92; Fig. 6a, green) or a power law ($R^2$ = 0.79; Fig. 6a, blue) degree distribution indicates that it has intrinsic network structure. A power law connectivity degree distribution is the defining feature of a scale-free network[46,47] and hints that DNs may be linked via well-connected 'hub' neurons.

Inherent structure within this network also implies the existence of subnetworks, or clusters, with unique properties. To explore this

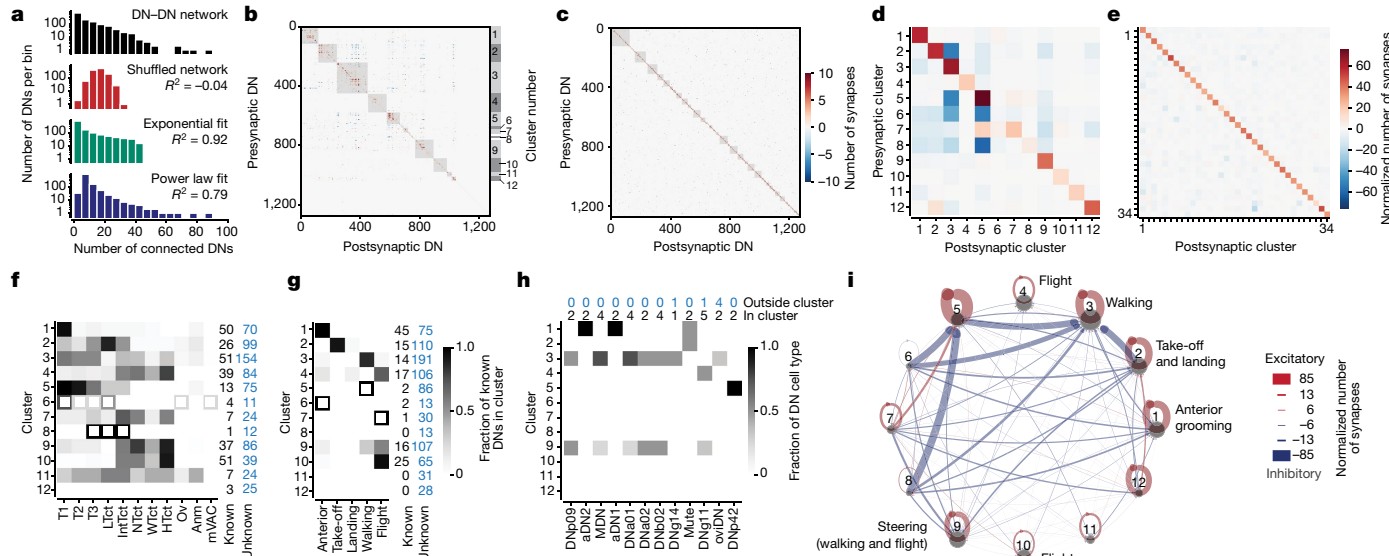

**Fig. 6 | Networks of DNs for similar behaviours excite one another and inhibit those for other behaviours. a**, The connectivity distribution of the DN–DN network (black), the same data after shuffling individual connections (red), the best exponential fit (green) or the best power law fit (blue). **b**, DN–DN connectivity clusters (grey squares) indicating excitatory (red) and inhibitory (blue) connectivity between presynaptic DNs (rows) and postsynaptic DNs (columns). The numbers on the right side indicate cluster numbers in **d,f–i. c**, As in **b**, but for a network with shuffled DN–DN connectivity. **d**, The number of synapses (excitatory minus inhibitory) between any two clusters normalized by the number of DNs in the postsynaptic cluster. **e**, As in **d**, but for the shuffled network in **c. f**, Fraction of known DNs within each cluster projecting to different VNC neuropil regions. Anm, abdominal neuromere; HTct, haltere tectulum; IntTct, intermediate tectulum; LTct, lower tectulum; mVAC, medial ventral association centre; NTct, neck tectulum; Ov, ovoid; T1–T3, leg neuropils; WTct, wing tectulum. Data are from ref. 13. **g**, Fraction of known DNs within each cluster associated with distinct behaviours. Data are taken from the literature (Supplementary Table 8). Open squares indicate clusters containing fewer than five known DNs (**f,g**). **h**, The distribution of experimentally investigated DNs across DN clusters. **i**, A network visualization of clusters in **d** with associated behaviours from **g**. There are predominantly excitatory (red) connections within each DN cluster and inhibitory (blue) connections between clusters.

possibility, we identified clusters of DNs in the fly brain by applying the Louvain method, a community detection algorithm[48]. Indeed, we could reliably identify multiple clusters of DNs with strong interconnectivity (Fig. 6b, grey boxes). When we applied the same algorithm to our shuffled network, we only inconsistently found small clusters (Fig. 6c, grey boxes). This was apparent in the number of DNs in the five largest clusters for the original DN–DN network (726 ± 42 neurons) versus the shuffled DN–DN network (581 ± 51 neurons; mean ± s.d., $P < 0.001$ comparing 100 repetitions of the Louvain method). Within clusters, we observed predominantly strong excitatory connections (Fig. 6d, diagonal elements). By contrast, connectivity between clusters was dominated by inhibition (Fig. 6d, off-diagonal elements). In the shuffled DN–DN network, this inhibition was weaker and more uniformly distributed (Fig. 6e, off-diagonal elements).

Distinct excitatory clusters imply parallel DN modules with distinct anatomical and/or functional properties. We investigated this possibility by first asking whether DN clusters (with similar connectivity in the brain) connect to similar targets in the VNC. Specifically, we studied the projections of known DNs[2,14] within the VNC connectome of an adult male fly[13]. This analysis revealed very specific projection patterns including, for example, that cluster 1 predominantly projects to a neuropil controlling the front legs (T1), cluster 2 predominantly to the lower tectulum (LTct), clusters 3 and 5 most strongly to all three leg neuropils (T1, T2 and T3), and clusters 4, 7, 9 and 10 predominantly to dorsal neuropils involved in wing, haltere and neck control (WTct, HTct and NTct, respectively) (Fig. 6f).

These results strongly suggest that specific excitatory DN clusters may also regulate distinct behaviours. To investigate this possibility, we identified 132 known DNs that have been shown or are predicted to be involved in anterior movements, walking, take-off, flight and landing (Supplementary Table 8). Indeed, we found that clusters included DNs with known links to specific behaviours and VNC projections (Fig. 6g). For example, as might be expected, DNs related to anterior grooming—DNg10 (ref. 21), DNg12 (ref. 21), aDN1 (ref. 4) and aDN2 (ref. 4)—were predominantly in cluster 1 targeting the T1 neuropil controlling the front legs. ComDNs that we studied experimentally were also in behaviourally consistent clusters (Fig. 6h). aDN1 and aDN2 are in the 'anterior grooming' cluster 1, whereas DNp09, MDN, DNa01, DNa02 and DNb02 are in the 'walking' or 'steering' clusters 3 and 9, with neurons in the right hemisphere being assigned mainly to cluster 3 and those in the left hemisphere being assigned to cluster 9 (Extended Data Fig. 7).

These data support the model that DNs form networks to orchestrate particular behaviours. A closer look at the comDNs that we tested experimentally supports this community-based inference (Extended Data Fig. 8a). First, DNp09 neurons driving forwards walking have direct excitatory connections with both DNa02 and DNb02 (Extended Data Fig. 8b), which, when optogenetically activated, elicit turning (Extended Data Fig. 5d,e). Second, aDN2 antennal grooming neurons connect directly to aDN1 neurons (Extended Data Fig. 8c), which also elicit antennal grooming (Extended Data Fig. 5b). Third, MDN backwards walking neurons connect to DNa01 neurons (Extended Data Fig. 8d), which, when activated, elicit turning (Extended Data Fig. 5c). Fourth, beyond DNs that we tested experimentally, we found that BDN2 and oDN1 (ref. 49)—two sets of recently discovered comDNs that drive walking—have similar DN connectivity patterns (Extended Data Fig. 8e) and interconnectivity to DNp09 (Extended Data Fig. 8f–h). In addition, we observed similar (Extended Data Fig. 8i) and mutual (Extended Data Fig. 8j,k) connectivity among DNs known to drive antennal grooming (aDN1 and aDN2). Together, these data support a model in which distinct behaviours are orchestrated by specific excitatory DN networks.

Of note, some clusters receive strong inhibition from other clusters. For example, cluster 2 related to take-off inhibits cluster 3 related to walking (Fig. 6i). Within these two clusters, excitatory connections prevail (Extended Data Fig. 9a,b). However, inhibitory DNs within cluster

2 project strongly to cluster 3 (Extended Data Fig. 9c,d). In particular, four cluster 2 'web' DNs[15] inhibit a large number of cluster 3 DN targets (96, 86, 45 and 41 DNs) (Extended Data Fig. 9d, asterisks). These inhibitory connections are well poised to contribute to action selection and the suppression of conflicting behaviours.

## Discussion

Here, by combining optogenetic activation, functional imaging and brain connectome analysis, we have resolved two seemingly conflicting observations: the activation of a few comDNs is sufficient to drive complete behaviours such as forwards walking even though many more DNs are co-active when the same behaviour is generated naturally. To explain this discrepancy, we have found that precise stimulation of multiple classes of comDNs recruits activity in many additional DNs. Thus, the 'command' signal is not only conveyed directly to the VNC, but can also be sent to other brain neurons that convey additional descending signals. There are a number of circuit motifs that could give rise to DN–DN interactions. Although we focus on monosynaptic connectivity, we have also shown that comDNs (DNp09, aDN2 and MDN) ultimately reach—and may potentially co-activate—hundreds of other DNs within only a few synapses. Future work may map the identity of recruited DNs by matching volumetric imaging data to anatomical templates from connectomes[50].

Our experiments and brain connectivity analyses for 12 sets of comDNs show that they lie along a continuum of interconnectivity in which those targeting larger downstream DN populations require network recruitment to generate a complete behaviour, whereas those with fewer DN partners largely do not. These results are consistent with a descending control model in which most DNs drive relatively simple body part kinematics. Other privileged DNs (for example, comDNs) can then directly recruit an assortment of such DNs to construct a full behaviour. This resembles the proposal drawn from work in other insects that descending fibres 'act in consensus' to assemble a complete behaviour[51]. Each of these individual fibres may drive distinct 'motor primitives'—fundamental kinematic elements which, when combined, have been suggested to underlie both innate and learned behaviours in vertebrates and mammals[42,43,52–54]. Consistent with this framework, a recent study of DN control during walking in *Drosophila* has shown that specific DN classes control limb movement 'gestures' akin to motor primitives[55].

For a given comDN, we speculate that the number of actively controlled joints or appendages engaged to generate its behaviour may be reflected by the size of its downstream DN network (Extended Data Fig. 10a). Consistent with this, we found that behaviours driven by stimulating broadcaster DNs (for example, walking and turning) appear more complex than movements driven by stimulating standalone DNs (for example, abdomen curling and ovipositor extension). A similar distinction has been suggested for the descending control of complex (for example, forwards walking) versus simple, stereotyped (for example, stridulation) behaviours in Orthoptera[56]. To take a quantitative example from our own study, DNp09 requires its large downstream DN network to drive forwards walking, but MDN does not require a relatively small downstream DN network to drive backwards walking. We found that MDN-driven backwards walking only depends on active movements of the two hindlegs[57] (Extended Data Fig. 10b and Supplementary Video 6), whereas DNp09-driven forwards walking can be controlled by active movements of any two pairs of the six legs (Extended Data Fig. 10c–e and Supplementary Video 6).

A framework in which comDNs recruit additional DNs to generate complete behaviours suggests an efficient substrate for the evolution of new behaviours or the diversification of existing behaviours (for example, species-specific courtship displays) through the de novo coupling or uncoupling of DNs and their associated motor primitives. This mechanism is therefore likely also used for descending control in other species including mammals[27,52] and suggests new avenues for the design of more flexible artificial controllers in engineering and robotics[58].

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

# Methods

## Fly stocks and husbandry

All experiments were performed on female adult *D. melanogaster* raised at 25 °C and 50% humidity on a 12-h light–dark cycle. The day before optogenetic experiments (22–26 h prior), we transferred experimental and control[61] flies to a vial containing food covered with 20 µl all *trans*-retinal (ATR) solution (100 mM ATR in 100% ethanol; Sigma Aldrich R2500, Merck) and wrapped in aluminium foil.

**Functional imaging and behaviour experiments.** We generated transgenic flies expressing LexAop-opGCaMP6s (a gift from O. Akin[62]) under the control of a *Dfd-LexA* driver (a gift from J. Simpson[63]) and having a copy of *UAS-CsChrimson* (Bloomington ID 55135) (Supplementary Table 1, ID 1). We also generated flies that additionally had the *LexAop-tdTomato* transgene (Bloomington ID 77139) (Supplementary Table 1, ID 2). For most experiments, we used flies without tdTomato expression.

MDN-spGAL4 flies (also known as MDN3 from ref. 2) were used to drive backwards walking. aDN2-spGAL4 flies (also known as aDN2-spGAL4-2 from ref. 4) were used to drive antennal grooming. DNp09-spGAL4 flies (from ref. 3) were used to drive forwards walking. Their genotypes[2–4,14,15,22,64] are listed at the top of Supplementary Table 2.

For all experiments in Figs. 2 and 4, we crossed spGAL4 flies or wild-type flies (Phinney Ridge flies, Dickinson laboratory) with one of our stable transgenic driver lines for imaging (Supplementary Table 1, ID 1 or ID 2). For Fig. 2, flies were 2–9 days post-eclosion and experiments were performed at Zeitgeber time 7–13 (ZT7–13). For Fig. 4, flies were 2–9 days post-eclosion and experiments were performed at ZT4–7. For Fig. 5, Extended Data Figs. 5, 6 and 10, we crossed spGAL4 lines with *20XUAS-CsChrimson.mVenus* (*attP40*) flies (Bloomington ID 55135). Control experiments were performed by crossing wild-type flies (Phinney Ridge flies, or Canton S) to *20XUAS-CsChrimson.mVenus* (*attP40*). The exact genotypes of the split lines and the source stocks are listed in Supplementary Table 2. All experiments were performed on flies 4–8 days post-eclosion at ZT4–7.

**Confocal imaging experiments.** We generated flies with stable *Dfd*-driven expression of membrane-targeted tdTomato or nuclear-targeted mCherry based on flies generated by the McCabe laboratory (EPFL) (Supplementary Table 1, IDs 3 and 4). For the three spGAL4 driver lines targeting comDNs (MDN, DNp09 and aDN2), we generated stable lines expressing CsChrimson (Supplementary Table 1, IDs 5, 6 and 7). We crossed flies expressing a red fluorescent protein variant with flies expressing CsChrimson in a spGAL4 driver line to visualize the expression patterns using confocal microscopy (Extended Data Fig. 1).

**Recording from DNs using a Dfd driver line.** We leveraged a genetic-optical intersectional approach to selectively record from GNG DNs. We chose to record from GNG DNs because we found that 73% of all DN–DN synapses in the brain connectome are in the GNG. In addition, the GNG houses 60% of all DNs and 85% of all DNs have axonal output in the GNG[14]. However, the Hox gene *Dfd* does not include the entirety of all GNG DNs: it excludes those driven by the Hox gene Sex combs reduced (*Scr*)[65]. Sterne et al.[15] have estimated that 550 cells in the GNG are *Dfd* positive and 1,100 are *Scr* positive, with only a small fraction expressing both. We show, for example, that aDN2, although localized to the GNG, is *Dfd* negative and thus most likely *Scr* positive (Extended Data Fig. 1c). In our study, functional imaging of DNs using an *Scr* driver line proved difficult because *Scr* expression extends into the neck and anterior VNC[63]. Specifically, we observed strong expression of GCaMP in the tissues surrounding the thoracic cervical connective (potentially ensheathing glia[66]), making it very hard to record the activity of DN axons. We expect that some *Scr*-positive DNs will also be recruited by

comDNs. Thus, we probably under-report the number of recruited GNG DNs.

**Limitations of selected spGAL4 driver lines.** In addition to descending neurons, our aDN2-spGAL4 driver line (*aDN2-GAL4.2* (ref. 4)) contains two more groups of neurons. One pair is on the anterior surface of the brain and, based on our control experiments, is probably not or only weakly activated by targeted optical stimulation of the neck (and not at all activated by thoracic stimulation). Another is a set of neurons in the anterior VNC. Because other driver lines targeting aDN2 neurons with more, different off-target neurons have the same behavioural phenotype as our aDN2 driver[4], we are confident that the effects that we observed are due to stimulating aDN2 neurons.

Different studies have reported variable behavioural phenotypes for stimulating the DNp09-spGAL4 driver line: some saw forwards walking[3], whereas others observed stopping or freezing[18,67]. We observed both: at our standard 21-µW optogenetic stimulation power, heterozygous animals mostly walked forwards. Occasionally, flies would only transiently walk forwards and then stop, or alternate rhythmically between walking and stopping. With higher expression levels of CsChrimson (that is, *DNp09-spGAL4 > UAS-CsChrimson* homozygous animals), we observed mostly freezing. We used heterozygous animals for our study.

## Immunofluorescence tissue staining and confocal imaging

We dissected brains and VNCs from 3 to 6 days post-eclosion female flies as described in ref. 68.

For samples in Extended Data Fig. 1a,c, we fixed flies in 4% paraformaldehyde (PFA; 441244-1KG, Sigma Aldrich, Merck) in 0.1 M PBS (Gibco PBS, pH 7.4, 10010-015, Thermo Fisher Scientific). We then washed them six times for 10 min with 1% Triton (Triton X-100, X100-100ML, Sigma Aldrich, Merck) in PBS (hereafter named 1% PBST) at room temperature. We then transferred them to a solution of 1% PBST, 5% natural goat serum (goat serum from controlled donor herd, G6767-100ML, Sigma Aldrich, Merck) and primary antibodies (see Supplementary Table 3) and left them overnight at 4 °C. We then washed the samples six times for 10 min with 1% PBST at room temperature. We transferred them to a solution of 1% PBST, 5% natural goat serum and secondary antibodies (see Supplementary Table 3) and left them for 2 h at room temperature. We then washed the samples six times for 10 min with 1% PBST at room temperature. We mounted the samples on glass slides using SlowFade (SlowFade Gold Antifade Mountant, S36936, Thermo Fisher Scientific) and applied a coverslip. To space the slide and the coverslip, we placed a small square of two layers of double-sided tape at each edge. We sealed the edges of the coverslip with nail polish.

For samples in Extended Data Fig. 1b, we fixed flies in 4% PFA in PBS and transferred them to 1% PBST and left them overnight at 4 °C. We then washed the samples three times for 15 min with 1% PBST at room temperature. We transferred them to a solution of 1% PBST, 5% natural goat serum and primary antibodies (see Supplementary Table 3) and left them overnight at 4 °C. We then washed the samples three times for 15 min with 1% PBST at room temperature. We transferred them to a solution of 1% PBST, 5% natural goat serum and secondary antibodies (see Supplementary Table 3) and left them overnight at 4 °C. We then washed the samples three times for 15 min with 1% PBST at room temperature. We mounted the samples on glass slides using SlowFade and applied a coverslip. To space the slide and the coverslip, we applied a small square of two layers of double-sided tape at each edge. We sealed the edges of the coverslip with nail polish.

We imaged samples using a Leica SP8 Point Scanning Confocal Microscope with the following settings: ×20, 0.75 NA HC PL APO dry objective, 2× image averaging, 1,024 × 1,024 pixels, 0.52 × 0.52-µm pixel size, 0.5-µm *z*-step interval; green channel 488-nm excitation, 50–540-nm emission bandpass; red channel (imaged separately to avoid cross-contamination) 552-nm excitation, 570–610-nm emission bandpass; and infrared channel (nc82, imaged in parallel with the

green channel) 638-nm excitation, 650–700-nm emission bandpass. We summed confocal image stacks along the z-axis and rotated and translated the images to centre the brain/VNC using Fiji[69].

## Optogenetic stimulation system and approach

We used a 640-nm laser (Coherent OBIS 1185055 640 nm LX 100 mW, Edmund Optics) as an optogenetic excitation light source. We reduced the light intensity using neutral density filters (Thorlabs) and controlled the light intensity with mixed analogue and digital control signals coming from an Arduino with custom software. A digital signal was used to turn the laser on and off. An analogue signal (PWM output from Arduino and RC low-pass filtered) was used to modulate the power. Both of those signals were sent in parallel to the laser and acquisition board and were recorded alongside the two-photon microscope signals using ThorSync 3.2 software (Thorlabs). The light was directed towards the fly with multiple mirrors. Fine control of the target location was achieved using a kinematic mount (KM100, Thorlabs) and a galvanometric mirror (GVS011/M, Thorlabs). We manually optimized targeting of the laser onto the neck/thorax before each experiment. The light was focused onto the fly using a plano-convex lens with $f$ = 75.0 mm (LA1608, Thorlabs) placed at the focal distance from the fly. For stimulation of the inhibitory opsin GtACR1, we used the same system, but with a 561-nm laser (Coherent OBIS 1280720 561 nm LS 150 mW, Edmund Optics) instead of a 640-nm laser to better match the optical excitation spectrum of GtACR1.

We note that, although comDNs have axon collaterals in the GNG, none of the comDNs in this study were among the DN populations that we imaged: DNp09-spGAL4 and MDN-spGAL4 lines drive expression in neurons with cell bodies in the cerebral ganglia and not in the GNG (Extended Data Fig. 1a). The DN cell bodies of the aDN2-spGAL4 line are within the GNG but do not overlap with Dfd driver line expression (Extended Data Fig. 1c). Thus, we could be certain that any active DNs would be recruited through synaptic connections and not optogenetically. We identified laser light intensities that could elicit robust forwards walking, anterior grooming and backwards walking (Fig. 2a and Extended Data Fig. 1d).

We used different laser intensities to stimulate MDN (21 μW), DNp09 (21 μW) and aDN2 (41.6 μW) animals because 21-μW stimulation power mostly causes aDN2 animals to stop (Extended Data Fig. 1d). Activation of MDN in the head, neck and thorax was sufficient to trigger backwards walking (Extended Data Fig. 1e). Although some tissue scattering of laser light can be expected, in control experiments, we found that activation of the head capsule, but not the thorax, could strongly elicit forwards walking in the 'bolt protocerebral neurons' of the brain—these neurons are known to drive robust and fast forwards walking[3] (Extended Data Fig. 1f). Stimulation (21 μW) was more specific than 41.6 μW, which is why we selected 21-μW stimulation for MDN and DNp09 as well as the spGAL4 lines tested (Fig. 5f,g and Extended Data Figs. 5 and 6). We regularly calibrated the laser intensity by measuring it with a power metre (PM100D, Thorlabs) and adjusting the analogue gain of the laser.

## In vivo two-photon calcium imaging experiments

We performed two-photon microscopy with a ThorLabs Bergamo II two-photon microscope augmented with a behavioural tracking system as described in ref. 29. In brief, we recorded a coronal section of the thoracic cervical connective using galvo-resonance scanning at around 16-Hz frame rate. In addition, optogentic stimulation was performed as described above. We only recorded the green PMT channel (525 ± 25 nm) because the red PMT channel would be saturated by red laser illumination of the fly. In parallel, we recorded animal behaviour at 100 frames per second (fps) using two infrared cameras placed in front and to the right of the fly.

Flies were dissected to obtain optical access to the VNC and thoracic cervical connective as described in ref. 70. In brief, we mounted the fly to a custom stage by gluing its thorax and anterior head to the holder and removed its wings. Then, we opened the dorsal thorax using a syringe needle and waited for indirect flight muscles to degrade for approximately 1.5 h. We pushed aside the trachea and resected the gut and salivary glands. For some flies, where the trachea was obstructing the view, we placed a V-shaped implant[71] into the thoracic cavity to push the trachea aside. We then placed the fly over an air-suspended spherical treadmill marked with a pattern visible on infrared cameras for ball tracking (air flow at 0.6 l min⁻¹). While the fly was adapting to this new environment (approximately 15 min), the imaging region was identified and the optogenetic stimulation laser was centred onto the neck.

We used ThorImage 3.2 to record and ThorSync 3.2 software to synchronize imaging data. We recorded 10,000 microscopy frames (around 10 min) while also recording behavioural data using cameras placed around the fly and presenting optogenetic stimuli. During a typical 10-min recording session, we presented 40 stimuli (5-s stimulation and 10-s inter-stimulus intervals). Whenever the recording quality was still good enough (that is, many neurons were visible and the fly still behaved healthily), we recorded multiple sessions to increase the number of stimulation trials. Many GNG DNs were active during spontaneous behaviour in the absence of optogenetic stimulation. Thus, to distinguish between GNG DN activity due to comDN stimulation versus the spontaneous initiation of behaviours, we only analysed trials for which flies were walking immediately before optogenetic stimulation. Because flies were quite spontaneously active, analysing trials for which flies were previously walking instead of resting increased the data available for trial averaging. It also allowed us to avoid laser light causing quiescent control animals to behave, obscuring our analyses.

## Investigating natural behaviours

In Extended Data Fig. 2, we compared optogenetically elicited neural activity to activity observed during natural behaviours: forwards walking, anterior grooming and backwards walking. Natural forwards walking is frequently spontaneously generated by the flies. By contrast, we needed to stimulate the antennae with 5-s puffs of humidified air to increase the probability of natural grooming (Extended Data Fig. 2b). We provided humidified air puffs with an olfactometer (220A, Aurora Scientific) using the following parameters: 80 ml min⁻¹ air flow, 100% humidity, 5-s duration and 20-s inter-stimulus interval. To have humid air puffs (that is, an abrupt change in flow rate) instead of a switch from dry air to humidified air—the default olfactometer configuration—we only connected the 'odour' tube to the final valve and not the 'air' tube. Furthermore, to increase the likelihood of spontaneous backwards walking (Extended Data Fig. 2c), we replaced the spherical treadmill with a custom cylindrical treadmill that we found increases the motivation to backwards walk. Specifically, we designed a 10-mm diameter, 80-mg 3D-printed wheel (RCP-30 resin) and printed it using stereolithography through digital light processing (Envisiontec Perfactory P4 Mini XL). This wheel was mounted on a low-friction jewel-bearing holder (ST-3D sapphire shafts, VS-40 sapphire bearings, Freudiger SA). We marked the sides of the wheel with infrared-visible dots to facilitate infrared camera tracking of rotations and calculations of velocity to classify bouts of backwards walking. When using the wheel, we added an additional third infrared camera to the left of the wheel, where dot markers were visible.

## Recording neuronal activity of DNs after resecting the VNC

To record neuronal activity in Dfd DNs after cutting the VNC, we first mounted and dissected flies as described above for intact animals. We verified that the animal was responding to optogenetic stimulation where appropriate and that the animal was still healthy. Then, we used a pair of microsciss (FST, Clipper Neuro Scissors, no. 15300-00, Fine Science Tools GmbH) to cut the entire VNC in the T1 neuromere. We cut just posterior to the fat bodies surrounding the cervical connective. We verified that the VNC was cut by pulling on its posterior region with forceps. We then performed two-photon imaging and optogenetic

stimulation as in experiments with intact flies (that is, laser stimulation of the neck while recording a cross-section of the cervical connective). We recorded 5,000 microscopy frames (around 5 min) with 20 stimulation repetitions. Flies were hanging freely from the stage and not placed on the spherical treadmill because the VNC was injured resulting in no notable leg movements. Post-hoc, we recorded a volume stack of the cervical connective and T1 neuromeres to verify the location of the cut.

## Behavioural experiments in leg-amputated animals

To investigate the number of actively controlled appendages involved in forwards and backwards walking, we mounted flies to the same stages used for imaging and behaviour experiments. We recorded ten trials of responses to optogenetic stimulation on the spherical treadmill, leaving 25 s between each stimulation. We then used cold anaesthesia to amputate the legs of the flies, before letting the flies recover for at least 10 min. The amputation was performed bilaterally for either the front legs, mid-legs or hindlegs, using clipper scissors (FST, Clipper Neuro Scissors, no. 15300-00, Fine Science Tools GmbH). We amputated the legs at the level of the tibia–tarsus joint to minimize the lesion while removing tarsal adhesion. Once they recovered, we recorded flies again on the spherical treadmill for ten trials. The control flies used to investigate walking phenotypes were Canton S, in accordance with previous work on locomotion—in particular DNp09 (ref. 3).

## Behavioural experiments in headless animals

For behavioural experiments, we mounted flies to the same stages used for two-photon imaging, but without gluing the anterior part of the head to the holder. Then, without further dissection, we placed animals onto the spherical treadmill. After recording ten trials of responses to optogenetic stimulation in intact animals, we decapitated the fly by inverting the holder and pushing a razor blade onto the neck. To achieve this, we mounted a splinter of the razor blade onto the tip of a pair of dissection forceps for finer control. We took care not to injure the legs of the fly and to make a clean cut without pulling out thoracic organs passing through the neck connective. To limit desiccation, we then sealed the stump of the neck with a drop of UV-curable glue. We only continued experiments on flies if their limbs were moving following decapitation. We then placed the headless flies onto the spherical treadmill and let them recover for at least 10 min. Then, we recorded ten trials of responses to optogenetic stimulation on the spherical treadmill and ten trials in which the fly was hanging from the holder without contacting the spherical treadmill. In experiments for testing connectome-based predictions, we slightly modified this experimental procedure. Because intact control animals become aroused by optogenetic stimulation, to avoid false positives and to discover behavioural phenotypes for less well-studied DNs, we attempted to reduce the spontaneous movements of flies. First, instead of 10 s between optogenetic stimulation trials, we used 25 s. Second, we filled the fly holder with room temperature saline solution to buffer heating from infrared illumination. For Extended Data Figs. 5 and 6, control flies (no DN > CsChrimson) were of the Phinney Ridge genetic background except for the later-studied DNp42, oviDN and DNg11, which were compared with control flies of the Canton S genetic background.

## Data exclusion

We manually scored the quality of neural recordings (signal-to-noise ratio, occlusions, and so on) and the behaviour of the fly (rigidity, leg injury, among others) on a scale from 1 to 6 (where 1 is very good, 3 is satisfying and 6 is insufficient) for each 10-min recording session. We only retained sessions in which both criteria were at least at a 'satisfying' quality level. Unless indicated otherwise, we analysed trials in which the fly was walking before stimulus onset. Thus, we did not retain data from flies with less than ten trials of walking before stimulation. We chose to do this for several reasons: (1) GCaMP6s decays very slowly. Even if the fly was moving approximately 2 s before stimulation, we

still observed residual fluorescence signals, increasing the variability of changes upon stimulation. There were only very few instances in which the animal was robustly resting for more than 2 s, making the inverse analysis impossible. (2) We observed that control flies became aroused upon laser light stimulation. Thus, they may begin moving if they were resting before stimulation, indirectly driving DN activity and making it harder to discriminate between optogenetically induced versus arousal-induced activity. Data from flies that were resting before stimulation exhibit recruitment patterns that are similar, although not identical (see data at https://doi.org/10.7910/DVN/HNGVGA). DNp09 shows strong activation in the medial cervical connective (as for when the fly was walking before stimulation) and additional activation in lateral regions. The central neurons characteristic of aDN2 activation in animals that were previously walking are also active in animals that were previously resting. In addition, we observed more widespread, weaker activation. DN signals upon MDN activation were slightly more spread out when the fly was resting before stimulation.

For experiments with headless animals, we excluded data from flies in which one of the legs was visibly immobile after decapitation, when at least one leg was not displaying spontaneous coordinated movements, or when the abdomen was stuck to the spherical treadmill such that other movements became impossible.

## Behavioural data analysis

For analysis, we used a custom Python code unless otherwise indicated. Code for behavioural data preprocessing can be found in the 'twoppp' Python package on GitHub (https://github.com/NeLy-EPFL/twoppp) previously used in ref. 71. Code for more detailed analysis can be found in the GitHub repository (https://github.com/NeLy-EPFL/dn_networks) for this paper.

**Velocity computation.** As a proxy for walking velocities, we tracked rotations of the spherical treadmill using Fictrac[72]. Data from an infrared camera placed in front of the fly were used for these measurements as described in ref. 29. Raw velocity traces acquired at 100 Hz were noisy and thus low-pass filtered with a median filter (width = 5 = 0.05 s) and a Gaussian filter ($\sigma = 10 = 0.1$ s).

The velocity of the cylindrical treadmill was computed as follows. First, the wheel was detected in a camera on the left side of the fly using Hough circle detection. For each frame, we extracted a line profile along the surface of the wheel showing the dot pattern painted on its side. We then compared this line profile to the line profile of the previous frame to determine the most likely rotational shift. We converted this shift to a difference in wheel angle and then transformed this into a linear velocity in millimetres per second to make it comparable to quantification of spherical treadmill rotations. This image processing was prone to high-frequency noise. Therefore, we filtered raw velocities with a Gaussian filter ($\sigma = 20 = 0.2$ s).

**2D pose estimation.** We tracked nine keypoints from a camera on the right side of the fly: anal plate, ovipositor, most posterior stripe, neck, front leg coxa, front leg femur tibia joint, front leg tibia–tarsus joint, mid-leg tibia–tarsus joint and hindleg tibia–tarsus joint (see Fig. 1d) using SLEAP (v1.3.0)[73].

**Behaviour classification.** We classified behaviours using an interpretable classifier based on heuristic thresholds on the walking velocity, limb motion energy and front leg height. For example, we classified forwards and backwards walking as having a forwards velocity of more than 1 mm s$^{-1}$ and − 1 mm s$^{-1}$ or less, respectively. All parameters are shown in Supplementary Table 4. If none of the conditions was fulfilled, we classified the behaviour as undefined.

Anterior grooming was composed of a logical 'OR' of two conditions: (1) the front leg was lifted up high, or (2) the front leg was moving with high motion energy. Front leg height was computed as the vertical

distance between the front leg tibia–tarsus joint and the median position of the coxa. Pixel coordinates start from the top of the image. Thus, it is positive when the front leg is low (for example, during resting) and negative when the front leg is high (for example, during head grooming). Motion energy (ME) of the front legs, mid-legs and hindlegs was computed based on the movements of the respective tibia–tarsus joint as follows: $ME = \sqrt{(\Delta x_t)^2 + (\Delta y_t)^2}$, where $\Delta x_t$ and $\Delta y_t$ are the difference in $x$ and $y$ between two consecutive frames. We then computed the moving average of the motion energy within a 0.5-s (that is, 50 samples) window to focus on longer timescale changes in motion energy.

## Two-photon microscopy image analysis

We used a custom Python code unless otherwise indicated. For all image analysis, the $y$ axis is dorsal–ventral along the body of the fly, and the $x$ axis is medial–lateral. Image and filter kernel sizes are specified as $(y,x)$ in units of pixels. Code for two-photon data preprocessing can be found in the 'twoppp' Python package on GitHub (https://github.com/NeLy-EPFL/twoppp) previously used in ref. 71. Code for more detailed analysis can be found in the GitHub repository (https://github.com/NeLy-EPFL/dn_networks) for this paper.

**Motion correction.** Recordings from the thoracic cervical connective suffer from large inter-frame motion including large translations, as well as smaller, non-affine deformations. Contrary to motion-correction procedures used before for similar data[71], here we made use of the high baseline fluorescence seen in *Dfd > LexAop-GCaMP6s* animals instead of relying on an additional, red colour channel for motion correction. Thus, we performed motion correction directly on the green GCaMP channel. We compared the performance for data where a red channel was available and could only find negligible differences in ROI signals. Whether a neuron was encoding walking or resting was unchanged irrespective of whether we used the GCaMP channel or recordings from an additional red fluorescent protein.

We performed centre-of-mass registration on every microscopy frame to compensate for large cervical connective translations. We cropped the microscopy images (from 480 × 736 to 320 × 736 pixels). Then, we computed the motion field for each frame relative to one selected frame per fly using optic flow. We corrected the frames for this motion using bi-linear interpolation. The algorithm for optic flow motion correction was previously described in ref. 70. We only used the optic flow component to compute the motion fields and omitted the feature matching constraint. We regularized the gradient of the motion field to promote smoothness ($\lambda = 800$).

**ROI detection.** For each pixel, we computed the standard deviation image across time for the entire recording. This gives a good proxy of whether a pixel belongs to a neuron: it has high standard deviation because the neuron was sometimes active. We used this image as a spatial map of the recording to inform ROI detection. Example standard deviation images are also used as the background image for Fig. 2c.

We applied principal component analysis (PCA) on a subset of all pixels in the two-photon recording. We then projected the loadings of the first five principal components back into the image space. This gave us additional spatial maps integrating functional information to identify neurons. We then used a semi-automated procedure to detect ROIs; we performed peak detection in the standard deviation map. We visually inspected these peaks for correctness by looking at both the standard deviation map and the PCA maps. We manually added ROIs that the peak detection algorithm had missed, for example, because the neuron was only weakly active. The functional PCA maps allowed us to discriminate between nearby neurons with dissimilar functions. They might show up as one big peak in the standard deviation map, but would clearly be assigned to different principal components. We were able to annotate between 50 and 80 ROIs for each fly. The number of visible neurons varies due to GCaMP6s expression levels,

dissection quality, recording quality and the behavioural activity level of the fly.

## Neural signal processing

We extracted fluorescence values for each annotated ROI by averaging all pixels within a rhomboid shape placed symmetrically over the ROI centre (11 pixels high and 7 pixels wide). This gave us raw fluorescence traces across time for each neuron/ROI. We then low-pass filtered those raw fluorescence traces using a median filter (width = 3 = ~0.185 s) and a Gaussian filter ($\sigma = 3 = ~0.185$ s).

**Δ$F$/$F$ computation.** Because of variable expression levels among cells, GCaMP fluorescence is usually reported as a change in fluorescence relative to a baseline fluorescence. Here we were mostly interested whether neurons were activated. To have a quantification that was comparable across neurons, we also normalized fluorescence of each neuron to its maximum level. Thus, we computed $\Delta F/F = \frac{F - F_0}{F_{max} - F_0}$, where $F$ is the time-varying fluorescence of a neuron, $F_0$ is its fluorescence baseline and $F_{max}$ is its maximum fluorescence. We computed $F_{max}$ as the 95% quantile value of $F$ across the entirety of the recording. In rare instances, neurons would get occluded, or slight glitches of the motion-correction algorithm would result in some residual movement. Both of these make it challenging to estimate the minimum fluorescence. When the fly is resting, nearly all neurons are at their lowest levels (aside from several[29]) and there is usually less movement of the nervous system. Thus, we computed $F_0$ as a 'resting baseline' as follows. First, using our behavioural classifier, we identified the onset of prolonged resting (at least 75% of 1 s after onset classified as resting and at least 1 s after the previous onset of resting) outside of optogenetic stimulation periods. For each neuron, we then computed the median fluorescence across repetitions aligned to resting onset. We then searched for the minimum value in time over the 2 s following rest onset. Taking the median across multiple instances of resting provided a more stable way to compute the baseline than by simply taking the minimum fluorescence. For flies that were not behaving (that is, those with resected VNCs shown in Extended Data Fig. 3), we could not compute a resting baseline and instead used the 5% quantile value as $F_0$. The normalization using $F_0$ and $F_{max}$ provided a way to compare fluorescence across multiple neurons with similar units. Thus, whenever we report absolute $\Delta F/F$, a value of 0 refers to neural activity during resting and 1 refers to the 95% quantile of neural activity. When we report $\Delta F/F$ relative to pre-stimulus values (Fig. 2b–f,i and Extended Data Fig. 2), the unit of $\Delta F/F$ persists and a value of 0.5 means that the neuron has changed its activity level half as much as when it would go from a resting state to its 95% quantile state.

**Video data processing.** To process the raw fluorescence videos shown in Supplementary Videos 1 and 2 and in Fig. 2b, we first low-pass filtered the data with the same temporal filters as for ROI signals (median filter width = 3 = ~0.185 s, Gaussian filter $\sigma = 3 = ~0.185$ s). In addition, we applied spatial filters (median filter width = [3,3] pixels, Gaussian filter $\sigma = [2,2]$ pixels). We then applied the same $\Delta F/F$ computation method described above, but for each individual pixel instead of for individual ROIs. Thus, the units used in the videos are identical to the units used for ROI signals in Fig. 2 and Extended Data Fig. 1.

**Synchronization of two-photon imaging and camera data.** We recorded two different data modalities at two different sampling frequencies: two-photon imaging data were recorded at approximately 16.23 Hz and behavioural images from cameras were acquired at 100 Hz. We synchronized these recordings using a trigger signal acquired at 30 kHz. When it was necessary to analyse neural and behavioural data at the same sampling rate (for example, Supplementary Videos 1 and 2), we downsampled all measurements to the two-photon imaging frame rate by averaging all behavioural samples acquired during

one two-photon frame. In the figures, we report data at its original sampling rate.

## Stimulus-triggered analysis of neural and behavioural data

We proceeded in the same way irrespective of whether the trigger was the onset of optogenetic stimulation (Figs. 2, 4 and 5 and Extended Data Figs. 1, 3, 5, 6 and 10) or the onset of a natural (spontaneous or puff elicited) behaviour (Extended Data Fig. 2). To compute stimulus-triggered averages, we aligned all trials to the onset of stimulation and considered the times between 5 s before the stimulus onset and 5 s after stimulus offset. In Fig. 2, we only considered trials in which the fly was walking in the 1 s before stimulation (behaviour classification applied to the mean of the 1-s pre-stimulus interval). We only considered flies with at least ten trials of walking before stimulation. Behavioural responses in Figs. 2a, 4b–g and 5f,g and Extended Data Fig. 1d–f, 2a–c, 5, 6 and 10 show the average across all trials (including multiple animals) and the shaded area indicates the 95% confidence interval of the mean across trials. When behavioural probabilities are shown, the fraction of trials that a certain behaviour occurs at a specific time after stimulus onset is shown. Neural responses over time in Fig. 2d and Extended Data Figs. 2a–c and 3c,h show average responses across all trials for one animal. To visualize the change in neural activity upon stimulation, the mean of neural activity in the 1 s before stimulation is subtracted for each neuron. If the absolute value of the mean across trials for a given neuron at a given time point was less than the 95% confidence interval of the mean, the data were masked with 0 (that is, it is white in the plot). This procedure allowed us to reject noisy neurons with no consistent response across trials. Because we subtracted the baseline activity before stimulus onset, we also observed DNs that became less active upon optogenetic stimulation (neurons appearing blue). However, GCaMP6s fluorescence does not reliably reflect neural inhibition. Thus, we cannot claim that this reduced activation in some neurons is due to inhibition. Instead, because the flies were walking before stimulation onset, those neurons most likely encode walking and became less active when the fly stopped walking forwards.

Individual neuron responses in Fig. 2c and Extended Data Fig. 2a–c,f and 3b,g show the maximum response of a single neuron/ROI. We detected the maximum response during the first half of the stimulus (2.5 s). We then computed the mean response of this neuron during 1 s centred around the time of its maximum response. If during at least half of that 1 s the mean was confidently different from 0 (that is, |mean| > CI), we considered the neuron to be responsive, otherwise we masked the response to zero to reject noisy neurons with no consistent response across trials. Figure 2b shows the same as Fig. 2c, but with this processing applied to pixels rather than individual neurons/ROIs. Contrary to previous ROI processing, pixels are not masked to 0 in case they are not responsive. Figure 2e shows an overlay of Fig. 2c for multiple flies. Data from each of these flies were registered to one another by aligning the $y$ coordinates of the most dorsal and ventral neurons, as well as the $x$ coordinate of the most lateral neurons. Figure 2f is a density visualization of Fig. 2e. To compute the density, we set the individual pixel values where a neuron was located to its response value and summed this across flies. We then applied a Gaussian filter ($\sigma = 25$ pixels, kernel normalized such that it has a value of 1 in the centre to keep the units interpretable) and divided by the number of flies to create an 'average fly'. Extended Data Fig. 2d was generated in the same manner.

## Statistical tests.

Figure 2g–i includes a statistical analysis of neural responses. We quantified the number of activated neurons for each fly (Fig. 2g) as the neurons whose response value was positive (as in Fig. 2c). We quantified the fraction of activated neurons for each fly (Fig. 2h) by dividing the number of activated neurons by the number of neurons detected in the recording. In Fig. 2i, we quantified the summed $\Delta F/F$ as the sum of the response values of neurons that were positively activated (see the red line in Fig. 2d). Here we ignored neurons with negative response values because reductions in GCaMP fluorescence should not be interpreted as reflecting inhibition (see above). We used two-sided Mann–Whitney $U$-tests (scipy.stats.mannwhitneyu[74]) to statistically analyse these comparisons. Sample sizes and $P$ values are described in the figure legends. The Mann–Whitney $U$-test is a ranked test. Thus, comparing three samples against three samples (for example, aDN2 versus control), where all samples are at identical relative positions (that is, ranks), will yield the same $P$ value, even if the absolute values are slightly different. This leads the $P$ values to be identical across Fig. 2g–i, reflecting the conservative choice of a rank test that does not assume an underlying distribution.

Figures 4b–e and 5f,g and Extended Data Figs. 5 and 6 show statistical tests comparing the behavioural responses of intact and headless flies. Figures 4f,g and 5f,g and Extended Data Figs. 5 and 6 show statistical tests comparing the behavioural responses of headless experimental flies with headless control flies. In each case, we used two-sided Mann–Whitney $U$-tests (scipy.stats.mannwhitneyu[74]) to compare the average value within the first 2.5 s after stimulus onset. We averaged across technical replicates (trials) and only compared biological replicates (individual flies) using statistical tests. Exact $P$ values rounded to three digits are indicated in Supplementary Table 5.

Statistical tests in Extended Data Fig. 10 show comparison of the behavioural responses of leg-amputated experimental flies with intact experimental flies, and leg-amputated experimental flies with leg-amputated control flies. In each case, we used two-sided Mann–Whitney $U$-tests (scipy.stats.mannwhitneyu[74]) to compare the total displacement after 5 s of stimulation. We averaged across technical replicates (trials) and only compared biological replicates (individual flies) using statistical tests. Exact $P$ values rounded to three digits are in Supplementary Table 6.

Extended Data Fig. 2a–c (right) and 2e show the Pearson correlation between neural responses to optogenetic stimulation and neural activity during natural (spontaneous or puff-elicited) behaviours. The two-sided significance of the correlation is measured as the probability that a random sample has a correlation coefficient as high as the one reported (scipy.stats.pearsonr v1.4.1 (ref. 74)).

In all figures showing statistical tests, significance levels are indicated as follows: \*\*\*$P < 0.001$, \*\*$P < 0.01$, \*$P < 0.05$ and not significant (NS) $P \geq 0.05$.

## Brain connectome analysis

**Loading connectome data.** We used the female adult fly brain (FAFB) connectomics dataset[7] from Codex[75] (version hosted on Codex as of 3 August 2023, FlyWire materialization snapshot 630; https://codex.flywire.ai/api/download) to generate all figures. We merged the 'neurons', 'morphology clusters', 'connectivity clusters', 'classification', 'cell stats', 'labels', 'connections' and 'connectivity tags' tables. We then found DNs by filtering for the attribute super_class=descending. We identified DNs with known, named (for example, DNp09) genetic driver lines from Namiki et al.[14] by checking the 'cell type', 'hemibrain type' and 'community labels' attributes (in this priority) and using the following rules. Otherwise, we used the consensus cell type[38] (for example, DNpe078). We semi-automatically assigned names using the following rules:

1. For special neurons, we manually labelled root IDs 720575940610236514, 720575940640331472, 720575940631082808 and 720575940616026939 as MDNs based on community labels from S. Bidaye (consensus cell type DNpe078); root IDs 720575940616185531 and 720575940624319124 as aDN1 based on community labels from K. Eichler and S. Hampel (consensus cell type DNge197); and root IDs 720575940624220925 and 720575940629806974 as aDN2 based on community labels from K. Eichler and S. Hampel (consensus cell type DNge078). We verified visually that the shape of the neurons corresponded to published light-level microscopy images[2,4].

2. Otherwise, if both the hemibrain_type attribute and the cell_type attribute followed the Namiki format ('DN{1 lowercase letter} {2 digits}', for example, 'DNp16') and they are identical, we used this as the cell name. If they are both in this format but are not identical, we marked this neuron for manual intervention.

3. Otherwise, if the hemibrain_type attribute follows the Namiki format, we used this as the cell name. In addition, if the hemibrain_type attribute follows the Namiki format, but the cell_type attribute has a different value following the consensus cell-type format ('DN{at least 1 lowercase letter} {at least 1 digit}', such as 'DNge198'), we marked the cell as requiring manual attention.

4. Otherwise, if the cell_type attribute follows the Namiki format, we used this as the cell name.

5. Otherwise, if the cell_type attribute follows the consensus cell-type format, we used this as the cell name.

6. Otherwise, we marked the cell as requiring manual intervention.

7. Wherever manual intervention was required (mostly in which the hemibrain_type is the Namiki format, but the cell_type is in the consensus cell-type format), we manually assigned the consensus cell type. However, we assigned the Namiki type if there was no other DN in this Namiki cell type or if the cell type was still missing a pair of DNs[14].

Next, we stored the connectome as a graph using SciPy sparse matrix[74] and NetworkX DirectedGraph[76] representations. We identified DNs with somas in the GNG by checking the third letter of the consensus cell type to be 'g' (that is, DNgeXXX)[38].

**Analysing connectivity.** We only considered neurons with at least five synapses to be connected and computed the number of connected DNs based on this criterion (Figs. 3, 5b,c,e and 6a–c and Extended Data Figs. 4–9). This is the same value as the default in Codex, the connectome data explorer provided by the FlyWire community[37,75]. Analysis of connectivity across three brain hemispheres (two brain halves from the FAFB dataset[7] and one from the hemibrain dataset[77]) revealed that connections "stronger than ten synapses or 1.1% of the target's inputs have a greater than 90% change to be preserved"[38]. We visualized all DNs connected to a given DN (Figs. 3a,b and 5d and Extended Data Figs. 5 and 6) using the neuromancer interface, and manually coloured neurons depending on whether they are in the GNG.

Neurotransmitter identification was available from the connectome dataset based on classification of individual synapses with an average accuracy of 87%[39]. Here we report neurotransmitter identity for a given presynaptic–postsynaptic connection. To define neurotransmitter identity for a given presynaptic–postsynaptic pair, we asserted that the neurotransmitter type would be unique using a majority vote rule. This was chosen as a tradeoff between harmonizing neurotransmitters for a neuron (especially GABA, acetylcholine and glutamate[78]) and avoiding the propagation of classification errors.

**DN network visualizations and DN hierarchy.** We used the networkx library[76] to plot networks of DNs in Figs. 3c,d and 5e and Extended Data Fig. 5–9. Again, we considered neurons to be connected if they had at least five synapses. In the circular plots, we show summed connectivity of multiple DNs. For example, the network for DNp09 in Fig. 3c shows only one green circle in the centre representing two DNp09 neurons. All connections shown as arrows are the sum of those two neurons. DNs are considered excitatory if they have the neurotransmitter acetylcholine and inhibitory if they have the neurotransmitter GABA. Whether glutamate is excitatory or inhibitory is unclear; this depends on the receptor subtype[60], which is unknown in most cases. To emphasize this, we highlight glutamatergic network edges in a different colour (pink).

In Fig. 3e, we show the cumulative distribution of the number of DNs reachable within up to $n$ synapses. Statistics on DN connectivity across multiple synapses were computed using matrix multiplication

with the numpy library on the adjacency matrix of the network. Lines in colour represent a DN network traversal starting at specific comDNs. The black trace represents the median of all neurons. Only a maximum of approximately 800 DNs can be reached because the others have maximally one DN input. In Fig. 5b,c, we sorted DNs by the number of monosynaptic connections that they make to other DNs. In Fig. 5b, the same sorting is applied to show the number of connected GNG DNs (orange).

In Extended Data Fig. 4, we show the effect of the choice of different constraints of the underlying connectome network on DN–DN connectivity degree. Statistics on DN connectivity across multiple synapses were computed using matrix multiplication with the numpy library on the adjacency matrices of the network. The segregation of excitatory and inhibitory connections was obtained by applying a mask on the direct connection signs. This implies that an inhibitory neuron acting on another inhibitory neuron would not be counted as excitatory but simply ignored in Extended Data Fig. 4d–f.

**Fitting network models to connectivity degree distribution.** In Fig. 6a, we generated a shuffled network of the same size by keeping the number of neurons constant and keeping the number of connections constant. Then, we randomly shuffled (that is, reassigned) those connections. Here we only considered the binary measure of whether a neuron was connected (number of synapses > 5) and not its synaptic weight. We then fit a power law or an exponential to the connectivity degree distribution using the scipy.optimize[74] library. Histograms of the degree distributions for all four distributions are shown in Fig. 6a using constant bin widths of five neurons. The quality of the fits are quantified using linear regression ($R^2$).

**Detection of DN clusters.** We applied the Louvain method[48] with resolution parameter $\gamma = 1$ to detect clusters in the undirected network of DNs (that is, connections between two neurons are scaled by their synaptic strength and neurotransmitter identity, but the directionality of the connection is not taken into account). Here all connections—feedforward, lateral and feedback—are taken into account. In brief, the Louvain method is a greedy algorithm that maximizes modularity (that is, the relative density of connections within clusters compared with between clusters). To simplify analysing the network during the optimization, we did not consider the directionality of connections between neurons. If there is reciprocal connectivity between neurons, we add up the number of synapses (positive if excitatory, negative if inhibitory; here glutamate is considered inhibitory and neuromodulators are disregarded for the sake of simplicity). The Louvain method finds different local optima of cluster assignments due to its stochastic initialization and greedy nature. Therefore, we ran the algorithm 100 times. On the basis of the outcomes of these 100 runs, we defined a co-clustering matrix: the matrix has the same size as the connectivity matrix (number of DNs × number of DNs). Each entry represents how often two DNs end up in the same cluster. This matrix assigns each pair of DNs a probability to be in the same cluster. Using this meta-clustering, we could be sure that the sorting of DNs that we found through clustering is not a local optimum and that it is reproducible. We then applied hierarchical clustering to this matrix (using the 'ward' optimization method from the scipy.cluster.hierarchy library[74]) to get the final sorting of DNs shown in Fig. 6b. We used this final sorting to detect the clusters shown in grey in Fig. 6b as follows: we started from one side of the sorted DNs and sequentially grew the cluster. If the next DN was in the same Louvain clusters at least 25% of the time, we assigned it to the same cluster as the previous DN. If not, we started a new cluster with this DN and kept testing subsequent DNs to determine whether they fulfil the criteria for this new cluster. Finally, we only kept clusters that had at least ten neurons. This yielded 12 clusters (grey squares). We applied this same meta-clustering and sorting approach to analyse the shuffled network (same number of DNs, same number of connections and same number

of synapses, but shuffled connections). On this shuffled network, we found 34 clusters of much smaller size (Fig. 6c), hinting at a better clustering in our network than in a shuffled control (modularity = 0.27 for the original network and modularity = 0.12 for the shuffled network). The number of synapses is shown as positive (red) if it is excitatory and as negative (blue) if it is inhibitory.

We then analysed the connectivity within and between clusters. To do this, we accumulated the number of synapses between two clusters (positive for excitatory and negative for inhibitory). To be able to compare this quantity between clusters of different sizes, we divided this number of synapses by the number of DNs in the cluster that receives the synaptic connections. This quantity is visualized in Fig. 6d for the original DN–DN network clusters and Fig. 6e for the shuffled network as the 'normalized number of synapses'. If positive (red), then connections from one cluster to another are predominantly excitatory. If negative (blue), then connections are predominantly inhibitory. We did not mirror connectome data before clustering because it requires resolving discrepancies between left and right neuron pairs, which, in many cases, are also not identifiable as corresponding cell classes across the brain.

**Statistical comparison of original versus shuffled DN–DN clusters.** As detailed above, we applied the Louvain algorithm 100 times to increase the robustness of clustering. We computed statistics on the clustering of this dataset (mean and standard deviation) specifically on metrics including the size and number of clusters. We then compared these distributions with those for the shuffled graph using one-sided Welch's $t$-tests (scipy.stats.ttest_ind[74] with equal_var = False). The resulting statistics are a conservative quantification of the difference between the original network and the shuffled control, as each data point is taken independently. When performing the hierarchical clustering across 100 iterations, the large clusters from the biological network are preserved, whereas the random associations of the shuffled network become incoherent. In practice, the difference in cluster sizes reported statistically underestimates the difference between the resulting matrices shown in Fig. 6b,c. The 100 iterations result from random seed initialization, on the condition that the algorithm converges. We restarted it whenever the convergence criteria were not reached within 3 s. Indeed, we observed empirically that when the algorithm would not converge in 3 s, it would not do so for at least 30 min and was, therefore, terminated.

**Identifying DNs to test predictions.** On the basis of the cell-type data associated with each neuron in FAFB (see above), we were able to find many DNs from refs. 4,14,15,22,64 in the connectome database. We then checked which of them have either a very high number of synaptic connections to other DNs or a very low number. We then filtered for lines where a clean spGAL4 line was available. In addition, we focused on lines whose major projections in the VNC were outside of the wing neuropil, because we removed the wings in our experimental paradigm and thus might not be able to see optogenetically induced behaviours. This left us with 15 additional DNs to test our predictions. DNp01 (giant fibre) activation was reported to trigger take-off in intact and headless flies[44,45], so we did not repeat those experiments. This left us with 14 lines to test. The source and exact genotypes of those fly lines are reported in Supplementary Table 2. We then performed experiments with those 14 lines. Because intact control flies become aroused by laser illumination, but not headless control animals, to avoid false positives, we only analysed DN lines that either had a known optogenetic behaviour in intact flies (that is, DNp42, aDN1, DNa01, DNa02, oviDN and DNg11) or that had a clear phenotype in headless flies (that is, DNb02, DNg14 and Mute). Thus, we excluded Web, DNp24, DNg30, DNb01 (involved in flight saccades in ref. 79, but with no obvious phenotype on the spherical treadmill) and DNg16 as they did not fulfil either of these criteria and only analysed the remaining nine driver lines in Fig. 5 and Extended Data Figs. 5 and 6.

**Analysing DN–DN connectivity in the VNC.** We used the neuprint website to interact with the male adult nerve cord (MANC) connectome dataset[13,80]. There, we searched for neurons based on their names (MDN, DNp09, and so on) and checked whether there were any DNs among their postsynaptic neurons. We found all neurons that we used from ref. 14 (that is, DNp09, DNa01, and so on), MDN and oviDN. We were not able to find aDN2, aDN1, Mute, Web and DNp42.

**Analysing VNC targets of DN clusters.** We used data shown in Cheong et al.[13] (figure 3, supplement 2) to define whether a DN known from Namiki et al.[14] was projecting to a particular VNC neuropil. In brief, a DN is considered as projecting to a given neuropil if at least 5% of its presynaptic sites are in that region. We manually found the MDNs in the MANC dataset and determined the regions that they connect to using the same criterion. To generate Fig. 6f, for each cluster, we accumulated the number of known DNs that project to a given VNC region. We then divided this by the number of known DNs to obtain the fraction of known DNs within a cluster that project to a given region. The number of unknown DNs per cluster is also shown next to the plot. The raw data of associations between DNs and VNC neuropils are shown in Supplementary Table 8.

**Analysing behaviours associated with DN clusters.** We examined the literature[2–4,13,16,18,19,21,30,64,70,81–84] to identify behaviours associated with DNs and grouped them into broad categories (anterior grooming, take-off, landing, walking and flight). This literature summary is available in Supplementary Table 8. Of the 35 DN types annotated, we found conflicting evidence for only two: DNg11 is reported to elicit foreleg rubbing[21] while targeting mostly flight-related neuropils[13]; DNa08 targets flight power control circuits[13] but has been reported to be involved in courtship under the name aSP22 (ref. 23). In Fig. 6g, we assigned DNg11 to 'anterior' and DNa08 to 'flight'. We accumulated the number of known DNs that are associated with a given behaviour for each cluster. We then divided by the number of known DNs in the respective cluster to get a fraction of DNs within a cluster that have a known behaviour. The number of unknown DNs per cluster is also shown next to the plot. The raw data of associations between DNs and behaviours are shown in Supplementary Table 8.

**Analysing brain input neuropils for each DN cluster.** We used data from FAFB to identify the brain input neuropils for each DN cluster based on the neuropil annotation for each DN–DN synapse. Thus, localization information is given by the position of each synaptic connection and not the cell body of the presynaptic partner. This allows us to account for local processing and modularity of neurons. The acronyms of brain regions are detailed in Supplementary Table 7, with 'L' and 'R' standing for the left and right brain hemispheres, respectively. Results are reported as the fraction of synapses made in a neuropil out of all the postsynaptic connections made by DNs of a given cluster.

**Ethical compliance.** All experiments were performed in compliance with relevant national (Switzerland) and institutional (EPFL) ethical regulations. Characteristics of animals such as sex, age and husbandry are detailed in the Methods.

## Reporting summary

Further information on research design is available in the Nature Portfolio Reporting Summary linked to this article.

## Data availability

Data are available at: https://dataverse.harvard.edu/dataverse/dn_networks. The DOI are: https://doi.org/10.7910/DVN/6IL0X3, https://doi.org/10.7910/DVN/K0WMM4, https://doi.org/10.7910/DVN/TZK8FA,

https://doi.org/10.7910/DVN/INYAYV and https://doi.org/10.7910/DVN/HNGVGA. These repositories include processed data required to reproduce the figures for each fly. Owing to data storage limits, these do not include raw behaviour camera images or raw two-photon imaging files, which are available on reasonable request. This repository includes: all behavioural and neural time series required to reproduce figures describing experimental data, acquisition metadata files, confocal images and the SLEAP pose estimation model. The FAFB connectomics dataset from Codex (version hosted on Codex as of 3 August 2023, FlyWire materialization snapshot 630) can be found at: https://codex.flywire.ai/api/download. Source data are provided with this paper.

## Code availability

The analysis code is available at: https://github.com/NeLy-EPFL/dn_networks.

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

**Acknowledgements** We thank J. Phelps and S. Boy-Röttger for help with confocal dissections and staining; L. Goffinet for help with the initial identification of DNs in the FAFB connectome; D. Morales for generating transgenic fly lines; J. Simpson, O. Akin, S. Bidaye and B. McCabe for sharing transgenic fly lines; the Princeton FlyWire team and members of the Murthy and Seung laboratories, as well as members of the Allen Institute for Brain Science, for development and maintenance of FlyWire (supported by BRAIN Initiative grants MH117815 and NS126935 to Murthy and Seung laboratories); members of the Princeton FlyWire team and the FlyWire consortium for neuron proofreading and annotation; and the *Drosophila* Connectomics Group (Principal Investigator G. Jefferis) for sharing their large-scale proofreading and annotation of descending neurons in FAFB-FlyWire before publication. Proofreading and annotation in Cambridge were supported by Wellcome Trust Collaborative Awards (203261/Z/16/Z and 220343/Z/20/Z) to G. Jefferis; NIH BRAIN Initiative grant 1RF1MH120679-01 to D. Bock and G. Jefferis; and a Neuronex2 award to D. Bock and G. Jefferis (NSF 2014862 and MRC MC-EX-MR/T046279/1). We are grateful for pre-publication access to the FlyWire dataset. We thank K. Eichler, S. Hampel and S. Bidaye for annotating DNs in the FAFB connectome; and the members of the Neuroengineering Laboratory for helpful discussions and comments on the manuscript. J.B. acknowledges support from a Boehringer Ingelheim Fonds PhD stipend. F.H. acknowledges support from a Boehringer Ingelheim Fonds PhD stipend. S.W.-C. acknowledges support from a Boehringer Ingelheim Fonds PhD stipend. P.R. acknowledges support from an SNSF project grant (175667) and an SNSF Eccellenza grant (181239).

**Author contributions** Conceptualization, methodology, software, software validation, formal analysis, investigation, data curation, data validation, original draft preparation, reviewing and editing the manuscript, and visualization were done by J.B. In addition, J.B. performed the following technical contributions: visualization for Fig. 1; experiments, analysis and visualization for Fig. 2; some visualization for Fig. 3; experiments, analysis and visualization for Fig. 4; some experiments, some data analysis and some visualization for Fig. 5; as well as additional contributions: experiments, analysis and visualization for Extended Data Figs. 1–3. Conceptualization, methodology, software, software validation, formal analysis, investigation, data curation, data validation, reviewing and editing the manuscript, and visualization were done by F.H. Furthermore, F.H. performed the following technical contributions: most connectomics data analysis and most visualization for Fig. 3; most experiments, most data analysis and most visualization for Fig. 5; connectomics modelling and visualization for Fig. 6; as well as additional contributions: experiments, analysis and visualization for Extended Data Figs. 4–10. Conceptualization, methodology, software, software validation, formal analysis, investigation, data curation, data validation, and reviewing and editing of the manuscript were done by S.W.-C. In addition, S.W.-C. performed the following technical contributions: some connectomics data analysis for Figs. 3 and 5. Conceptualization, methodology, resources, original draft preparation, reviewing and editing of the manuscript, supervision, project administration and funding acquisition were done by P.R.

**Competing interests** The authors declare no competing interests.

**Additional information**
**Correspondence and requests for materials** should be addressed to Pavan Ramdya.

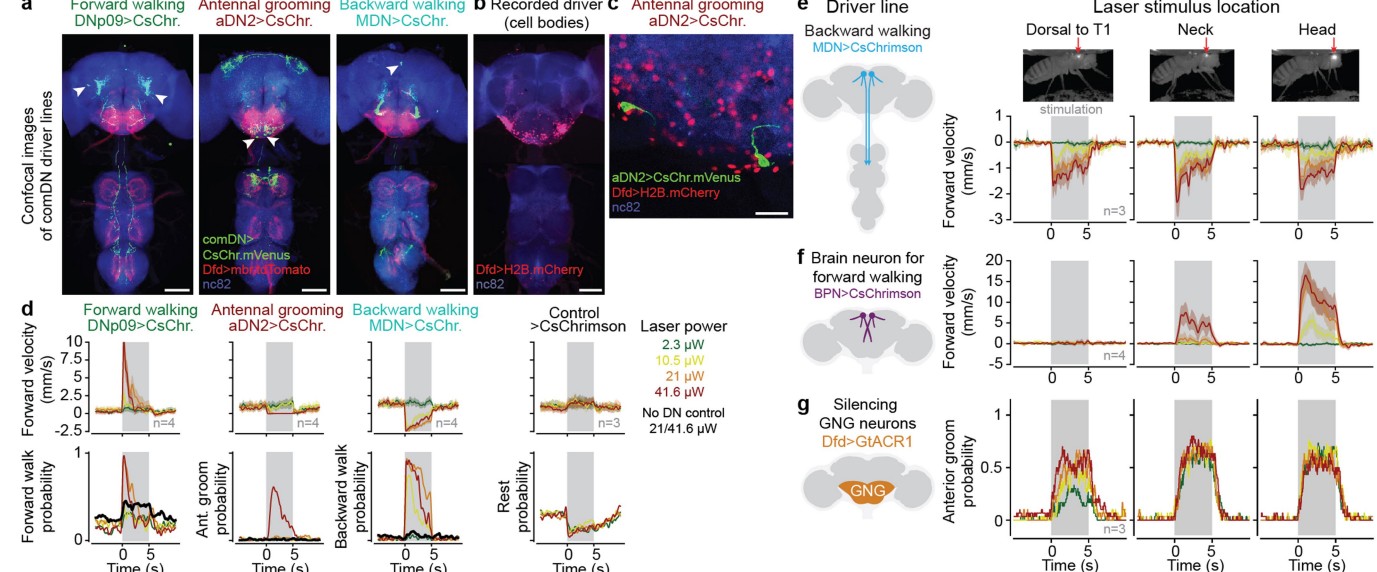

**Extended Data Fig. 1 | DN driver lines and optogenetic stimulation strategy.**
**(a)** Z-projected confocal images of the brain (top) and VNC (bottom) show
the expression of UAS-CsChrimson.mVenus (green) in command-like DNs,
membrane-bound tdTomato in the Dfd driver line (red), and neuropil ('nc82',
blue). The location of command-like DN cell bodies is indicated (white
arrowheads). Scalebars are 100 μm. **(b)** Z-projected confocal image of Dfd driver
line expression of soma-targeted mCherry. Only brain neurons in the GNG are
labeled. Scalebar is 100 μm. **(c)** Confocal image of the posterior GNG with Dfd
driver line expression of soma-targeted mCherry and aDN2 expression of UAS-
CsChrimson.mVenus (green). The two GNG-DNs in the aDN2 driver line are not
targeted by the Dfd driver line. Scalebar is 20 μm. Immunohistochemistry and
confocal imaging experiments in **(a-c)** were performed once due to the reliability
of these methods. **(d)** Behavioral responses to optogenetic stimulation of the
neck connective at different laser intensities for DNp09 (left; 4 flies, total 49
trials per condition), aDN2 (left-middle; 4 flies, total 60 trials per condition), MDN
(right-middle; 4 flies, total 50 trials per condition), and no DN control (right; 3
flies, total 60 trials per condition) animals. Flies reliably (i) walk forward upon

DNp09 stimulation for stimuli ≥ 21 μW, (ii) groom upon aDN2 stimulation only
for the highest stimulation power (41.6 μW) but rest at 21 μW, and (iii) walk
backward upon MDN stimulation for stimuli ≥ 10.5 μW. For all stimulation
intensities, control flies walk more and rest less. Thus, we selected 21 μW as our
default laser stimulation power and 41.6 μW for aDN2 stimulation specifically.
**(e)** MDN stimulation with focused laser light elicits backward walking when
illuminating the anterior dorsal thorax (left, as in Figs. 4 and 5), the neck (middle,
as in Fig. 2) or the head (right). 3 flies, total 30 stimulation trials per condition.
**(f)** Stimulation of a brain-specific neuron ('Bolt protocerebral neurons' or BPN)
known to drive forward walking[3] with focused laser light elicits forward
walking when illuminating the head (right), but not the thorax (left). Laser light
focused on the neck (middle) can only elicit weak forward walking at 41.6 μW.
4 flies, total 40 stimulation trials per condition. **(g)** Silencing GNG neurons
(Dfd-LexA > LexAop-GtACR1) with focused 561 nm laser light elicits anterior
grooming when illuminating the head (right), neck (middle), or thorax (left).
3 flies, total of 30 stimulation trials per condition. All velocity traces in **(d,e,f)**
show mean ± 95% confidence interval of the mean across stimulation trials.

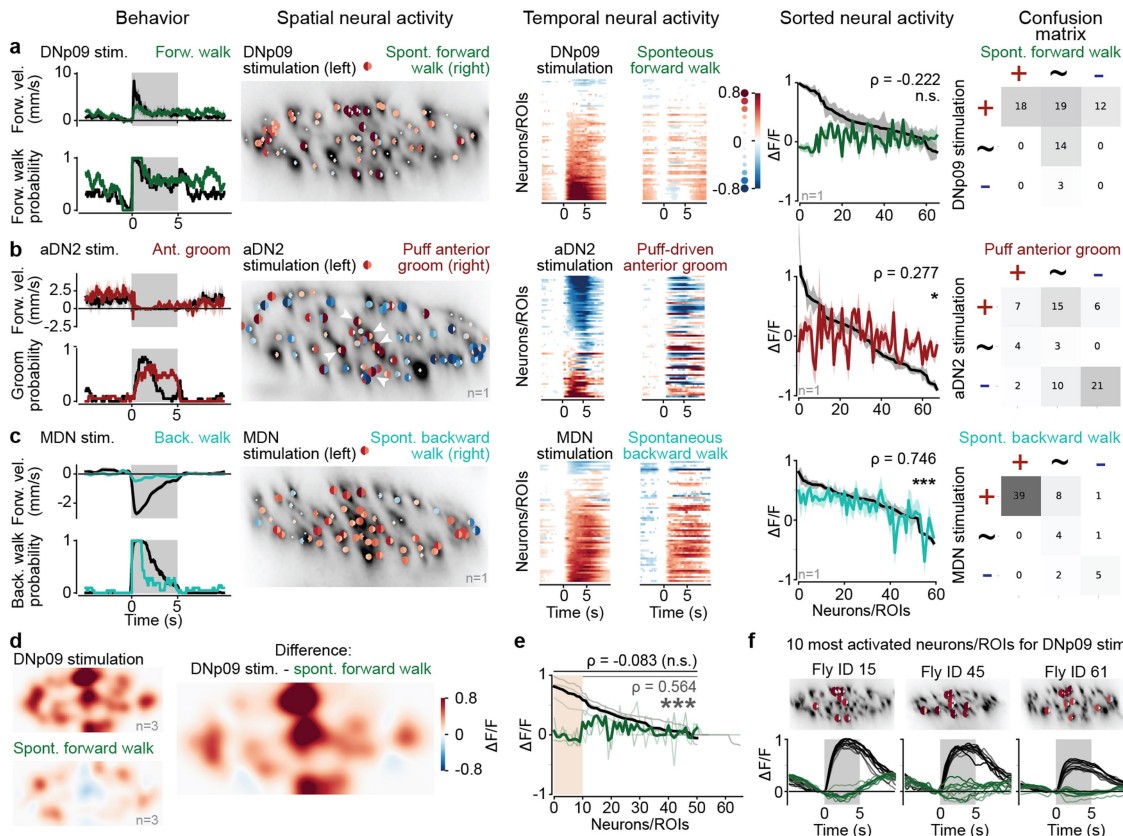

**Extended Data Fig. 2 | Comparison of GNG-DN population neural activity during optogenetic stimulation versus corresponding natural behaviors. (a-c)** For **(a)** DNp09 and forward walking, **(b)** aDN2 and anterior grooming, or **(c)** MDN and backward walking: **(left)** behavioral responses to optogenetic stimulation of command-like DNs (black) versus natural occurrences of the behavior in question (color); **(middle left)** single neuron/ROI responses (analyzed as in Fig. 2c). Here the left half-circle reflects the response to optogenetic activation and the right half-circle the activity during natural behavior; **(middle)** single neuron average responses as in Fig. 2d; **(middle right)** Comparing the activity of individual neurons between optogenetic stimulation (black) and natural behavior (color). Neurons/ROIs are sorted by the magnitude of their responses to optogenetic activation. Shaded areas indicate 95% confidence interval of the mean across trials. Pearson correlation between optogenetic and spontaneous response and significance of test against null-hypothesis (the two variables are uncorrelated, see Methods) are shown; **(right)** Confusion matrix comparing the number of active neurons/ROIs that were more active (+), similar (-), or less active (−) upon optogenetic stimulation versus during natural behavior. **(a)** DNp09: for one fly n=23 optogenetic stimulation trials (not forward walking before stimulus) and 28 instances of spontaneous forward walking in which the fly was not walking forward for at least 1 s and then walking forward for at least 1 s (correlation: $\rho = -0.022$, $p = 0.356$, $N = 66$ neurons, two-sided test, see Methods). **(b)** aDN2: for one fly, n = 20 optogenetic stimulation trials (pre-stimulus behavior not

restricted) and 16 instances of anterior grooming elicited by a 5 s humidified air puff (correlation: $\rho = 0.277$, $p = 0.022$, $N = 68$ neurons, two-sided test, see Methods). Indicated are central neurons/ROIs with strong activation during aDN2 stimulation of the neck cervical connective as in Fig. 2f. **(c)** MDN: for one fly, n = 80 optogenetic stimulation trials (pre-stimulus behavior not restricted) and 21 instances of spontaneous backward walking on a cylindrical treadmill in which the fly was not walking backward for 1 s and then walked backward for at least 1 s (correlation: $\rho = 0.746$, $p < 0.001$, $N = 60$ neurons, two-sided test, see Methods). **(d)** Density visualisation (as in Fig. 2f) of neural responses to DNp09 stimulation and spontaneous forward walking across three animals. The difference in responses is primarily localized to the medial but not lateral regions of the connective. To maximize comparability, only trials where the fly was not walking forward before stimulus onset were selected. **(e)** Same plot as in **a, middle right** but for three animals with DNp09 stimulation and forward walking. Indicated are the correlation values when including ($\rho = -0.083$, $p = 0.564$, $n = 172$ neurons across three flies, two-sided test, see Methods) or excluding ($\rho = 0.564$, $p < 0.001$, $n = 142$ neurons across three flies, two-sided test, see Methods) the ten neurons most activated by optogenetic stimulation (orange region). **(f)** The locations of ten neurons indicated in **e** within the connective of three flies (top) and their single neuron responses to optogenetic stimulation (bottom, black traces) or during natural backward walking (bottom, green traces).

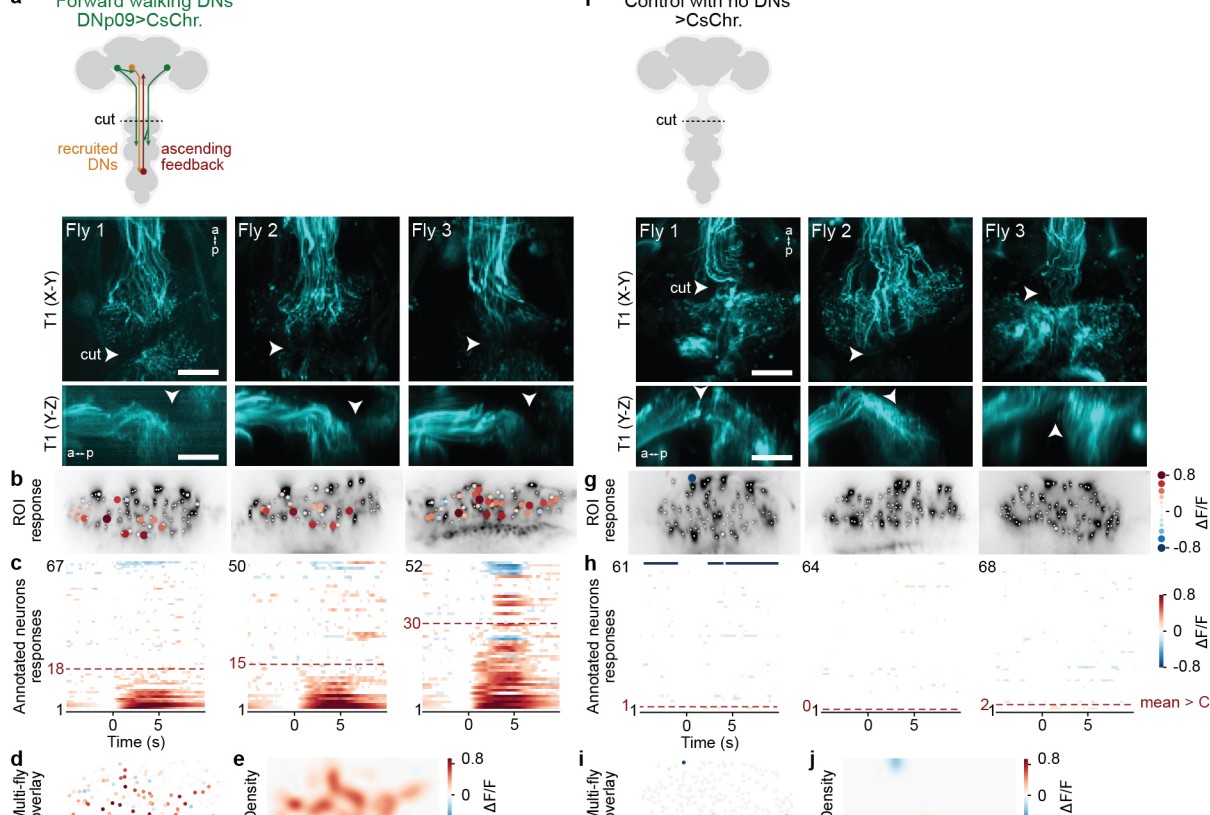

**Extended Data Fig. 3 | GNG-DNs are recruited by command-like DNs despite resection of ascending axons from the VNC.** Experimental data (three flies each) showing anatomy and functional responses of GNG-DNs upon optogenetic stimulation of **(a-e)** DNp09 > CsChrimson or **(f-j)** control flies. **(a,f)** Horizontal (top) and side (bottom) projections of the cervical connective and VNC for three flies after resecting the VNC T1 neuropil. Arrowheads indicate the locations of cuts. Scale bars are 50 $\mu$m. **(b,g)** Single neuron/region-of-interest (ROI) response magnitude during laser light illumination. Each circle is scaled and color-coded to represent the maximum change in fluorescence (normalized $\Delta F/F$) of one detected DN axon/ROI relative to the level of activity 1 s prior to illumination. Small white dots are shown if the response magnitude is smaller than the 95%

confidence interval of the mean across trials. The background image is a standard-deviation projection across time of raw fluorescence microscopy data. **(c,h)** Trial-averaged single neuron/ROI responses across time, aligned to illumination onset and ordered by response magnitude. Data are color-coded according to the magnitude of activity, or white if the response is smaller than the 95% confidence interval of the mean. Indicated are the number of neurons/ROIs with a positive response magnitude larger than the 95% confidence interval of the mean across trials (horizontal red line). **(d,i)** A registered overlay of the data from all three flies shown in panel **b,g**. **(e,j)** A density visualization of the data from all three flies shown in panel **b,g**.

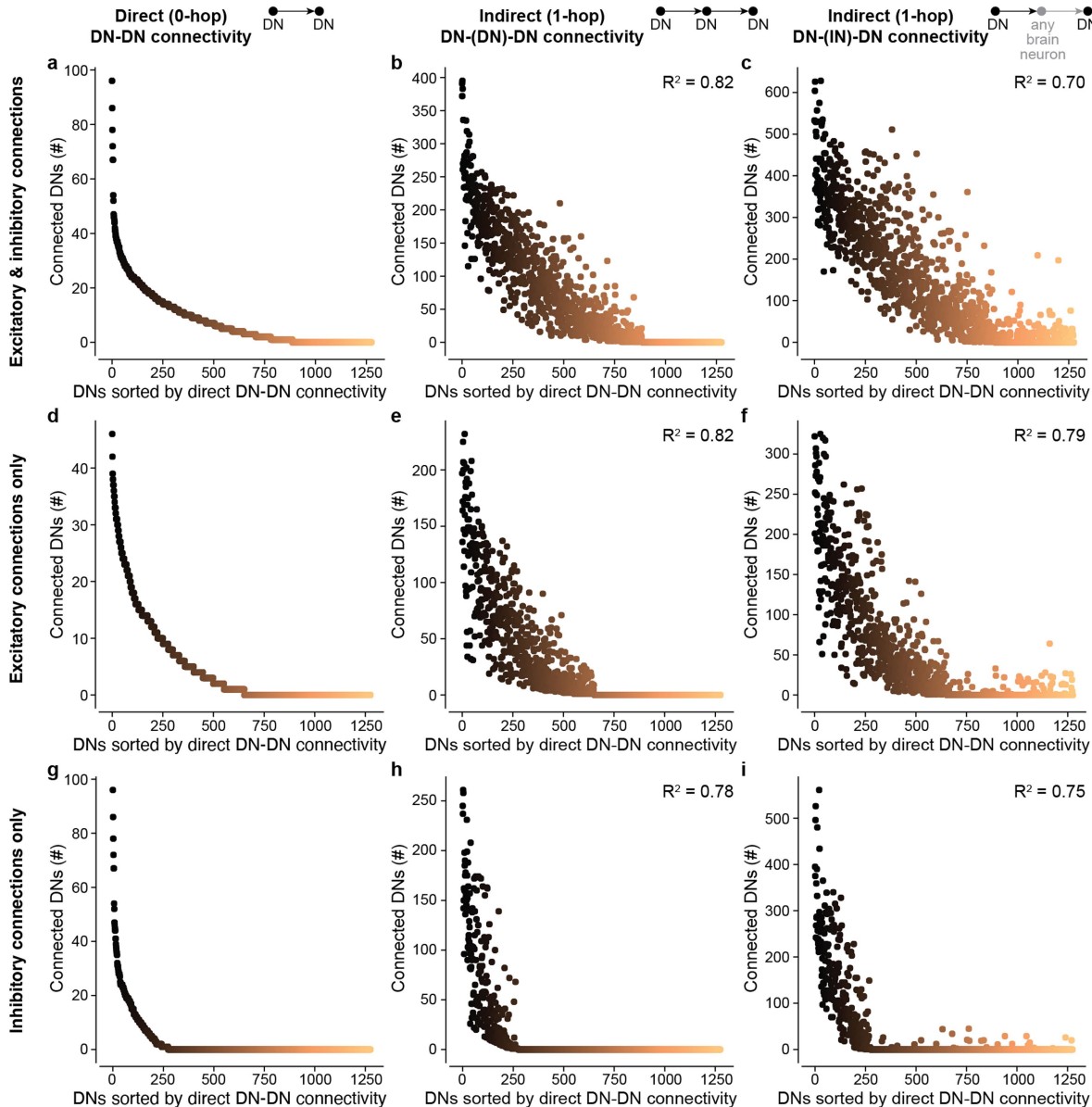

**Extended Data Fig. 4 | DN-DN connectivity statistics when also including interneurons in the underlying connectome network. (a)** The number of DNs monosynaptically (directly) downstream of every DN, taking into account both excitatory and inhibitory synapses. Data are identical to those in figure 5. **(b)** The number of DNs disynaptically downstream of each DN, allowing for at most one additional intervening DN. Sorting of the x axis identical to panel **(a)**. Correlation coefficient compares the distributions of panels **(a)** and **(b)**. **(c)** The number of DNs disynaptically downstream of each DN, allowing for at most one additional intervening interneuron of any type. Sorting of the x axis identical to panel **(a)**. Correlation coefficient compares the distributions of panels **(a)** and **(c)**. **(d-f)** Identical to panels **(a-c)** but restricted only to excitatory connections between individual neurons. Sorting of the x axis identical in panels **(d-f)**. **(g-i)** Identical to panels **(a-c)** but restricted only to inhibitory connections between individual neurons. Sorting of the x axis identical in panels **(g-i)**.

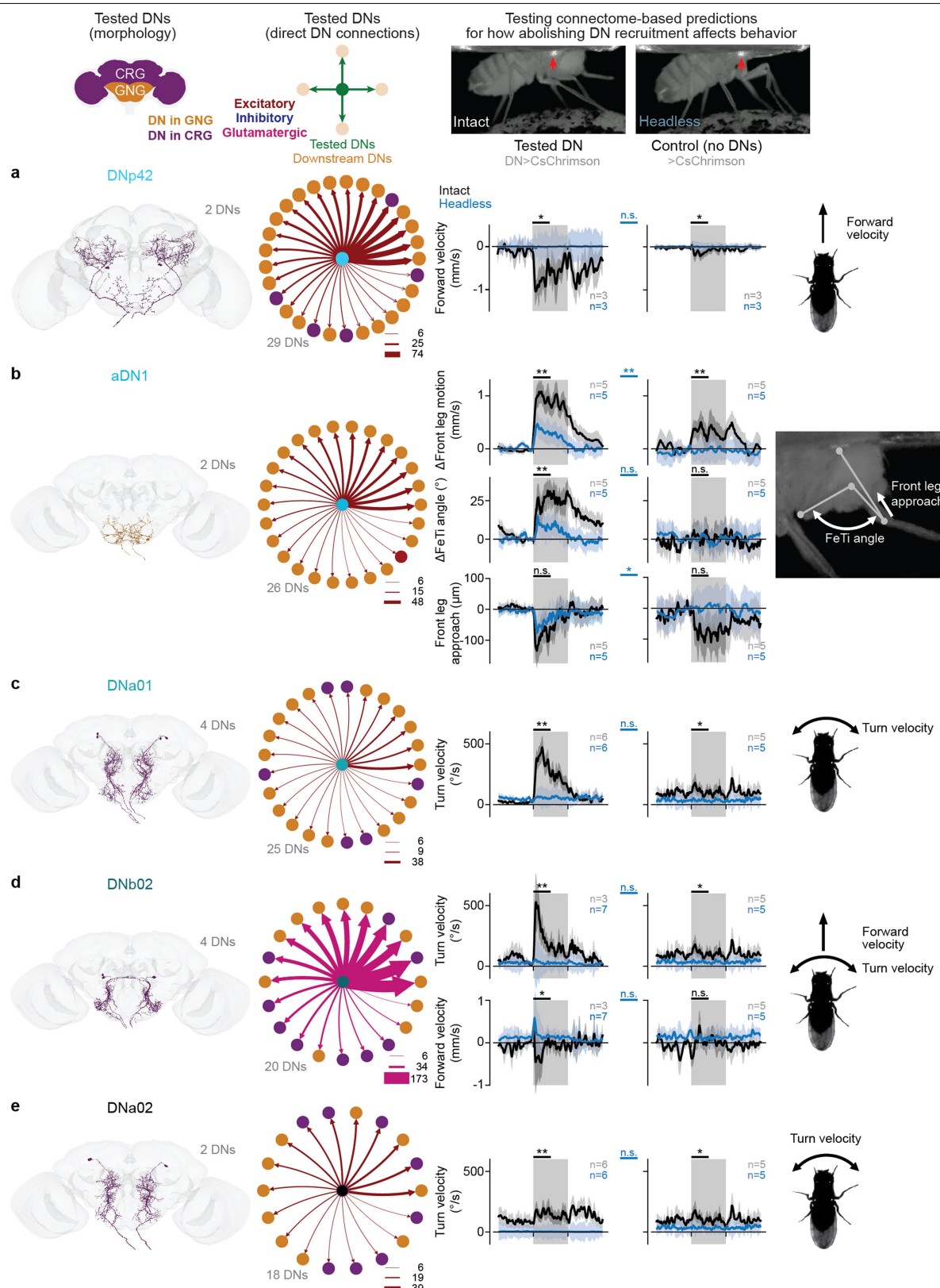

**Extended Data Fig. 5** | See next page for caption.

**Extended Data Fig. 5 | Testing the connectome-based prediction for broadcaster DNs that behaviors depend strongly on downstream DNs. (a-e)(first column)** The morphology of tested DNs in the adult female brain connectome. DNs are color-coded based on their somata localization within the cerebral ganglia (purple) or gnathal ganglia (orange). The number of DNs is indicated. **(second column)** A network schematic of direct connections from tested to downstream DNs. Edge widths reflect the number of synapses and is consistent across plots. Edge colors denote excitatory (red), inhibitory (blue), or glutamatergic (pink) which can be excitatory or inhibitory depending on receptor type[60]. **(third column)** Quantitative analyses of optogenetically-driven behaviors and movements in intact (black traces) and headless animals (blue traces). The number of flies for each condition are indicated. Each fly is optogenetically stimulated ten times. Thus, the average and 95% confidence interval of the mean for a total of $n*10$ trials is shown. **(fourth column)** Identical behavioral analysis for control flies without DN opsin expression. Note that controls for different parameters include the same five animals. Two-sided Mann-Whitney U tests comparing the trial mean of intact and headless animals (black bars, above each plot) and comparing headless experimental with headless control flies (blue, in between experimental and control plots) are shown (*** means $p < 0.001$, ** means $p < 0.01$, * means $p < 0.05$, n.s. means $p \geq 0.05$; for exact p-values see Supplementary Table 5). **(fifth column)** An illustration of the behavioral parameter(s) being quantified. **(a)** DNp42 has monosyaptic connections to 29 other DNs and triggers backing up in intact animals[64]. This behavior is not observed in headless flies, as quantified by fictive forward walking velocity. **(b)** aDN1 has monosynaptic connections to 26 other DNs and triggers grooming in intact animals. By contrast, headless animals produce mostly uncoordinated front leg movements. These occur more slowly at a lower frequency (top) with a smaller change in femur-tibia angle (middle). The 'front leg approach' to the head—the change in Euclidean distance between the neck and tibia-tarsus joint relative to 1 s before stimulus onset—is similar between intact and headless animals (bottom). **(c)** DNa01 has monosynaptic connections to 25 other DNs and triggers in place turning. This is quantified as an increase in turn velocity. This behavior is lost in headless animals. **(d)** DNb02 has monosynaptic connections to 20 other DNs and weakly triggers turning. This is quantified as an increase in turning velocity (top), a phenotype that is lost in headless animals. Instead, a flexion of the front legs can be observed in headless animals. This is quantified as a short spike in forward velocity (bottom). These data partially overlap with those in Fig. 5d–g. **(e)** DNa02 has monosynaptic connections to 18 other DNs and weakly triggers turning. This is quantified as an increase in turning velocity. This behavior is lost in headless animals.

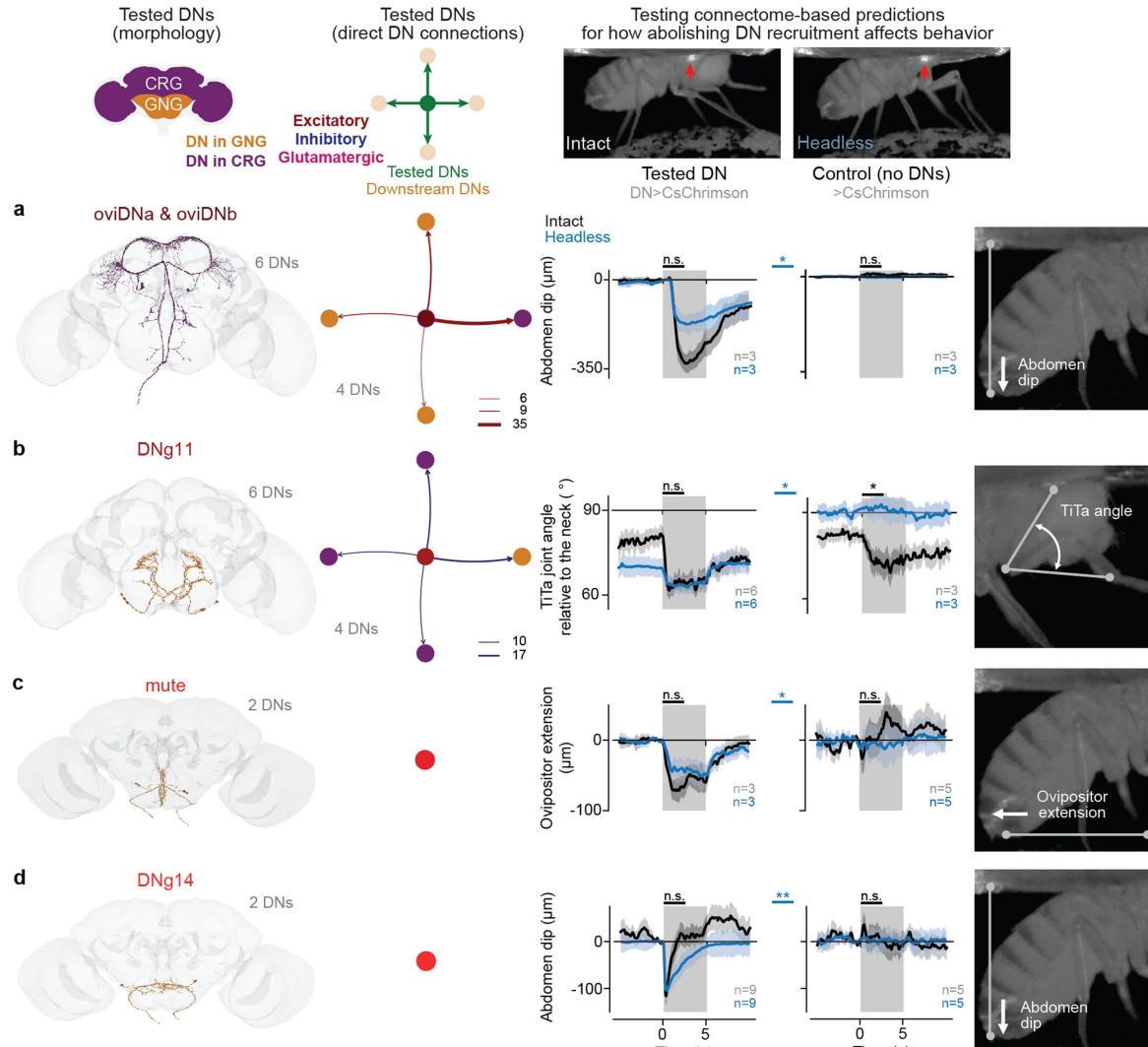

**Extended Data Fig. 6 | Testing the connectome-based prediction for stand-alone DNs that behaviors depend weakly on downstream DNs. (a-d)(first column)** The morphology of tested DNs in the adult female brain connectome. DNs are color-coded based on their somata localization within the cerebral ganglia (purple) or gnathal ganglia (orange). The number of DNs is indicated. **(second column)** A network schematic of direct connections from tested to downstream DNs. Edge widths reflect the number of synapses and is consistent across plots. Edge colors denote excitatory (red), inhibitory (blue), or glutamatergic (pink) which can be excitatory or inhibitory depending on receptor type[60]. **(third column)** Quantitative analyses of optogenetically-driven behaviors and movements in intact (black traces) and headless animals (blue traces). The number of flies for each condition are indicated. Each fly is optogenetically stimulated ten times. Thus, the average and 95% confidence interval of the mean for a total of $n*10$ trials is shown. **(fourth column)** Identical behavioral analysis for control flies without DN opsin expression. Note that controls for different parameters include the same five animals. Two-sided Mann-Whitney U tests comparing the trial mean of intact and headless animals (black bars, above each plot) and comparing headless experimental with

headless control flies (blue, in between experimental and control plots) are shown (*** means p < 0.001, ** means p < 0.01, * means p < 0.05, n.s. means p≥0.05; for exact p-values see Supplementary Table 5). **(fifth column)** An illustration of the behavioral parameter(s) being quantified. **(a)** oviDNs have four direct downstream partners and trigger curling of the abdomen in both intact and headless animals. This movement is quantified as a change in the vertical positioning of the ovum during optogenetic stimulation. **(b)** All together, six DNg11 neurons have four downstream partners and trigger foreleg rubbing[21]. This movement is quantified by the angle drawn by the axis between the coxa and front legs' tibia-tarsus joint, and the coxa-neck axis. This metric allows to compare positions across flies. **(c)** The DN 'Mute' has no monosynaptic connections to other DNs and triggers ovipositor extension in both intact and headless animals. This movement is quantified as a change in the horizontal position of the ovipositor relative to the 1 s prior to stimulus onset. **(d)** DNg14 has no monosynaptic connections to other DNs and triggers abdominal dipping and vibration in both intact and headless animals. This movement is quantified as a change in the vertical position of the anal plate relative to 1 s before stimulus onset. These are the same data as in Fig. 5d–f).

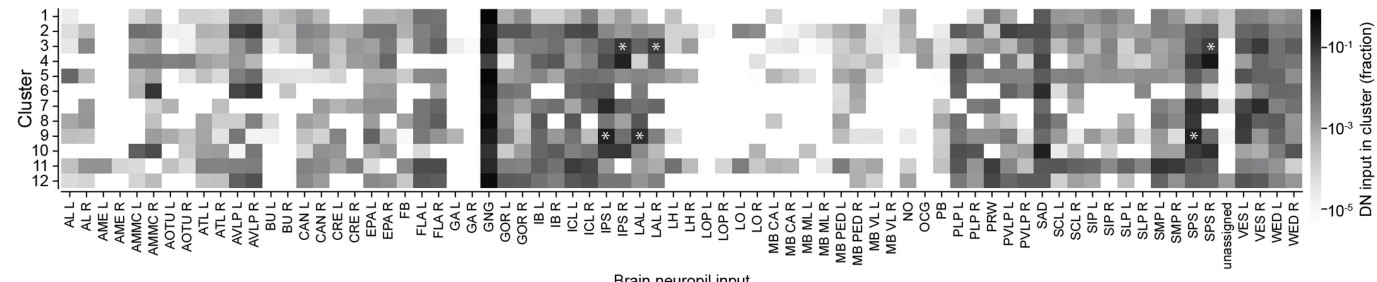

**Extended Data Fig. 7 | Inputs to clusters by brain region.** Fraction of DN input synapses from different brain neuropils within each cluster. Although there is largely no clear link between the source of DN inputs in the brain and DN clusters, there is one exception: Among 'walking' or 'steering' clusters 3 & 9 we find a bias with neurons in the right hemisphere being assigned mainly to cluster 3 and those in the left hemisphere being assigned to cluster 9. This was due to differences in connectivity between the two brain hemispheres, both in terms of bilateral symmetry in the brain as well as a left-right imbalance of inputs from the inferior posterior slope (IPS), superior posterior slope (SPS) and the lateral accessory lobe (LAL) (white asterisks). Neuropil names are listed in Supplementary Table 7.

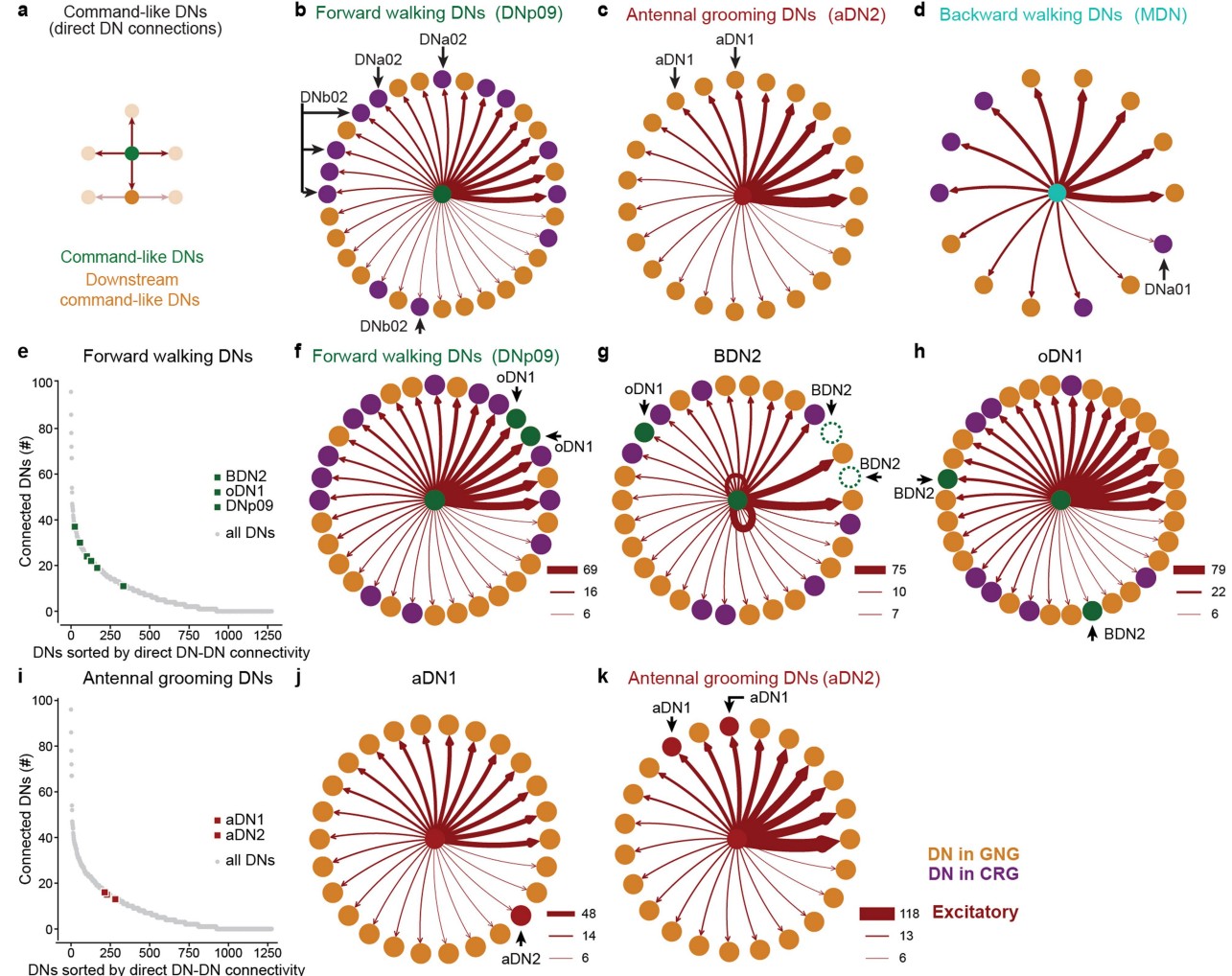

**Extended Data Fig. 8 | DN-DN connectivity for multiple DNs driving similar behaviors. (a-d)** DNs used to test predictions and that are directly downstream of our studied command-like DNs (DNp09, aDN2 and MDN). **(a)** Schematic illustrating that command-like DNs can recruit other command-like DNs involved in related behaviors. **(b)** Downstream partners of DNp09 include DNa02 and DNb02 neurons. **(c)** Downstream partners of aDN2 include aDN1 neurons. **(d)** Downstream of MDN is one DNa01 neuron. **(e,i)** Command-like DNs whose artificial stimulation are known to evoke **(e)** forward walking

(DNp09[3], BDN2[49], and oDN1[49]), or **(i)** antennal grooming (aDN1 and aDN2[4]) are all well-connected to other DNs. **(f-h,j,k)** Direct connectivity diagrams showing downstream partners of **(f)** DNp09, **(g)** BDN2, **(h)** oDN1, **(j)** aDN1, and **(k)** aDN2. Command-like DNs are shown at the center of each plot. Edge widths indicate the strength of the synaptic connections. Peripheral neurons highlighted in **(f-h)** green or **(j-k)** red are the interconnected DNs evoking forward locomotion or antennal grooming, respectively. Dashed circles in **(g)** represent internal connections among BDN2 neurons, grouped in the center through self loops.

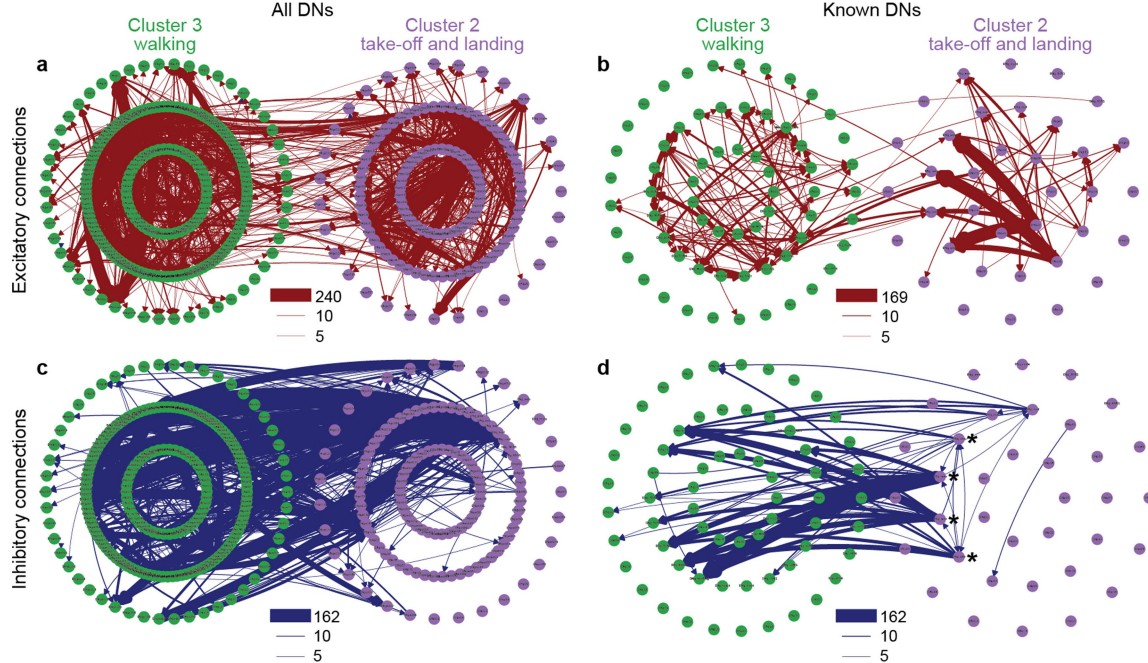

**Extended Data Fig. 9 | Node-wise connectivity between two clusters controlling walking versus take-off and landing. (a-b)** Excitatory connections between **(a)** all or only **(b)** experimentally studied (prior to[37]) nodes from cluster 2 implicated in take-off and landing (purple) or cluster 3 implicated in walking (green). Synapse counts are indicated by edge weights. Each cluster is organized such that DNs with outputs only within the cluster are on the inner ring, DNs with both inputs and outputs to the same cluster are on the middle ring, and remaining cluster DNs are on the outer ring. Most excitatory connections are within a given cluster. **(c-d)** As in panels **a-d** but including only inhibitory connections. Most connections project across clusters 2 and 3. In panel **d**, four Web DNs[15] are indicated (black asterisks). These neurons receive strong excitatory input from within their cluster 2 (panel **b**) and inhibit many DNs in cluster 3.

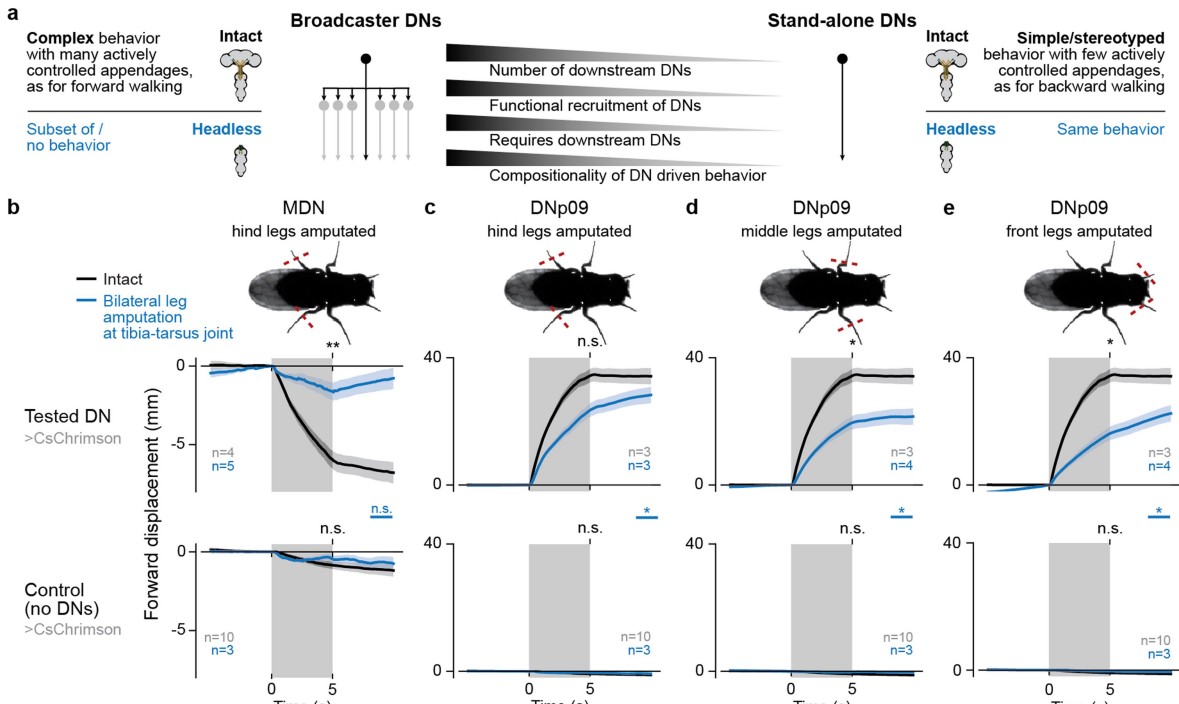

**Extended Data Fig. 10 | Backward locomotion depends on the active actuation of fewer appendages than forward locomotion. (a)** Illustration of the hypothesis that behavioral complexity/compositionality correlates with underlying DN network size. **(b-e, top row)** Cartoon schema illustrating legs that were bilaterally amputated at the level of the tibia-tarsus joint. Indicated are optogenetically activated DNs. Shown below is the cumulative fictive forward displacement for tethered flies before, after, and during optogenetic stimulation (gray region) for either optogenetic stimulation of **(b-e, middle row)** the DN in question, or **(b-e, bottom row)** a control animal with no GAL4 driver. Data are shown for traces for both amputated (blue) and intact control (black traces) flies. Flies were optogenetically stimulated 10 times. Shaded areas represent the 95% confidence interval of the mean. Shown are two-sided Mann-Whitney U tests comparing the trial-wise mean of intact versus leg amputated animals (black asterisks and 'n.s.') as well as the leg amputated DN > GAL4 versus leg amputated control flies (blue asterisks and 'n.s.'). *** Indicates $p < 0.001$, ** indicates $p < 0.01$, * indicates $p < 0.05$, n.s. indicates $p \geq 0.05$; for exact p-values see Supplementary Table 6. **(b)** Amputation of the hind legs is sufficient to prevent flies from walking backward upon MDN optogenetic stimulation. Residual backward displacement results from struggle-associated noise and is not statistically distinguishable from control animal backward displacement. **(c-e)** Amputation of either the hind-, mid- or forelegs does not prevent forward walking but only reduces forward walking velocity.

# Reporting Summary

## Statistics

For all statistical analyses, confirm that the following items are present in the figure legend, table legend, main text, or Methods section.

| n/a | Confirmed | |
|---|---|---|
| ☐ | ☒ | The exact sample size (*n*) for each experimental group/condition, given as a discrete number and unit of measurement |
| ☐ | ☒ | A statement on whether measurements were taken from distinct samples or whether the same sample was measured repeatedly |
| ☐ | ☒ | The statistical test(s) used AND whether they are one- or two-sided *Only common tests should be described solely by name; describe more complex techniques in the Methods section.* |
| ☐ | ☒ | A description of all covariates tested |
| ☐ | ☒ | A description of any assumptions or corrections, such as tests of normality and adjustment for multiple comparisons |
| ☐ | ☒ | A full description of the statistical parameters including central tendency (e.g. means) or other basic estimates (e.g. regression coefficient) AND variation (e.g. standard deviation) or associated estimates of uncertainty (e.g. confidence intervals) |
| ☐ | ☒ | For null hypothesis testing, the test statistic (e.g. *F*, *t*, *r*) with confidence intervals, effect sizes, degrees of freedom and *P* value noted *Give P values as exact values whenever suitable.* |
| ☒ | ☐ | For Bayesian analysis, information on the choice of priors and Markov chain Monte Carlo settings |
| ☒ | ☐ | For hierarchical and complex designs, identification of the appropriate level for tests and full reporting of outcomes |
| ☐ | ☒ | Estimates of effect sizes (e.g. Cohen's *d*, Pearson's *r*), indicating how they were calculated |

*Our web collection on statistics for biologists contains articles on many of the points above.*

## Software and code

Policy information about availability of computer code

| Data collection | Two-photon microscope images were acquired using ThorImage 3.2 software. Data synchronization was performed using ThorSync 3.2 software. Behavior images were acquired using custom Python scripts. Optogenetic stimulation was controlled using a custom Arduino 1.8.13 script. |
|---|---|
| Data analysis | Data analyses were performed using custom code written in Python 3. The code is available in the following repository: https://github.com/NeLy-EPFL/dn_networks Fiji v.2.9.0 software was used to sum z-projections of image z-stacks and to combine monochromatic images into RGB images. SLEAP v1.3.0 was used to perform 2D pose estimation. |

For manuscripts utilizing custom algorithms or software that are central to the research but not yet described in published literature, software must be made available to editors and reviewers. We strongly encourage code deposition in a community repository (e.g. GitHub). See the Nature Portfolio guidelines for submitting code & software for further information.

## Data

Policy information about availability of data

All manuscripts must include a data availability statement. This statement should provide the following information, where applicable:
- Accession codes, unique identifiers, or web links for publicly available datasets
- A description of any restrictions on data availability
- For clinical datasets or third party data, please ensure that the statement adheres to our policy

Data are available at:
https://dataverse.harvard.edu/dataverse/dn_networks
DOI are:
https://doi.org/10.7910/DVN/6IL0X3
https://doi.org/10.7910/DVN/K0WMM4
https://doi.org/10.7910/DVN/TZK8FA
https://doi.org/10.7910/DVN/INYAYV
https://doi.org/10.7910/DVN/HNGVGA
These repositories include processed data required to reproduce the figures for each fly. Due to data storage limits, these do not include raw behavior camera images or raw two-photon imaging files which are available upon reasonable request. This repository includes: all behavioral and neural time series required to reproduce Figures describing experimental data; Acquisition Metadata files; Confocal images; SLEAP pose estimation model.
The female adult fly brain (FAFB) connectomics dataset from Codex (version hosted on Codex as of August 3, 2023, FlyWire materialization snapshot 630) can be found at:
https://codex.flywire.ai/api/download.

## Research involving human participants, their data, or biological material

Policy information about studies with human participants or human data. See also policy information about sex, gender (identity/presentation), and sexual orientation and race, ethnicity and racism.

| Reporting on sex and gender | N/A |
| --- | --- |
| Reporting on race, ethnicity, or other socially relevant groupings | N/A |
| Population characteristics | N/A |
| Recruitment | N/A |
| Ethics oversight | N/A |

Note that full information on the approval of the study protocol must also be provided in the manuscript.

# Field-specific reporting

Please select the one below that is the best fit for your research. If you are not sure, read the appropriate sections before making your selection.

☒ Life sciences          ☐ Behavioural & social sciences          ☐ Ecological, evolutionary & environmental sciences

For a reference copy of the document with all sections, see nature.com/documents/nr-reporting-summary-flat.pdf

# Life sciences study design

All studies must disclose on these points even when the disclosure is negative.

| Sample size | Sample sizes were chosen based on convention in the field based on expected inter-animal variability from published results and pilot experiments. In total, we performed experiments with 176 flies:<br>(1) 20 flies for two-photon recordings during optogenetic stimulation after the fly was walking Fig 2). We have at least 3 samples per genotype.<br>(2) 25 flies to fine-tune and characterize the optogenetic stimulation system (Supp Fig 1; 9 overlap with (1)). We have at least 3 samples per genotype.<br>(3) 5 flies to examine spontaneous behavioral responses (Supp. Fig 2., 3 overlap with (1))<br>(4) 6 Flies to test DN recruitment upon VNC resection (Supp. Fig. 3). We have 3 samples per genotype.<br>(5) 20 flies for headless experiments (Fig. 4). We have at least 5 samples per genotype.<br>(6) 69 flies to test model predictions (Fig. 5 and Supp. Figs. 6,7). We have at least 3 samples per genotype.<br>(7) 35 flies to test forward and backward walking with amputated tarsi (Supp. Fig. 4). We have at least 3 samples per genotype.<br>(6) 5 flies for confocal imaging (Supp. Fig. 1)<br>(7) 11 flies for two-photon recordings during optogenetic responses after the fly was resting (Supp. File 1; 7 overlap with (1)) |
| --- | --- |

| Data exclusions | Data from two-photon recordings of behaving flies were excluded for animals and trials in which we observed abnormal limb movements, or low vitality. Two-photon imaging data were also excluded if they suffered from optical occlusions due to tissue debris, or extreme motion artifacts resulting from animal behavior. Both of these exclusions were performed based on a quantitative scoring system. In experiments with headless flies, we only retained data from flies that were not visibly injured after decapitation (i.e., all limbs were still able to move). |
|---|---|
| Replication | For the two-photon recordings in Figure 2, 3-9 replicates were recorded for each genotype.<br>For the headless experiments shown in figure 4, five replicates were recorded for each genotype. For the experiments testing model predictions in Figure 5 and Supp. Figure 3, 3-9 replicates were recorded for each genotype.<br>For other control experiments (Supp. Fig 1d-g, Supp Fig 3,  Supp Fig 4,) at least 3 replicates were recorded for each genotype. |
| Randomization | Experiments were not randomized because of the automated nature of the data analysis. |
| Blinding | Experimenters were not blinded in this study due to obvious behavioral phenotypes for specific genotypes during optogenetic activation. Additionally, for specific experiments, a behavioral phenotype was required to establish the health of the animals in question. |

# Reporting for specific materials, systems and methods

We require information from authors about some types of materials, experimental systems and methods used in many studies. Here, indicate whether each material, system or method listed is relevant to your study. If you are not sure if a list item applies to your research, read the appropriate section before selecting a response.

## Materials & experimental systems

| n/a | Involved in the study |
|---|---|
| ☐ | ☒ Antibodies |
| ☒ | ☐ Eukaryotic cell lines |
| ☒ | ☐ Palaeontology and archaeology |
| ☐ | ☒ Animals and other organisms |
| ☒ | ☐ Clinical data |
| ☒ | ☐ Dual use research of concern |
| ☒ | ☐ Plants |

## Methods

| n/a | Involved in the study |
|---|---|
| ☒ | ☐ ChIP-seq |
| ☒ | ☐ Flow cytometry |
| ☒ | ☐ MRI-based neuroimaging |

## Antibodies

| Antibodies used | Anti-Bruchpilot (mouse) NC82, Dev. Studies Hybridoma Bank (DSHB), NC82 1ml supernatant<br>GFP Tag Rabbit, ThermoFisher, G10362<br>Goat anti-Mouse Alexa 633, ThermoFisher, A21052<br>Goat anti-Rabbit Alexa 488, ThermoFisher, A11008<br>Living Colors DsRed, Takara, 632496<br>Chicken to GFP Anti-GFP, abcam, ab13970<br>Goat Anti-Rabbit (Cy3), abcam, ab6939<br>Goat Anti-Chicken (Alexa 488), abcam, ab150169 |
|---|---|
| Validation | Primary antibodies were validated by the suppliers as follows:<br>GFP Tag Rabbit (G10362) was verified by Relative expression<br>Living Colors DsRed, (632496) was validated by western blot.<br>Chicken to GFP (ab13970) was validated by western blot.<br>No manufacturer notes are available for the validation of other primary antibodies. No additional<br>validation was performed.<br>- DSHB - https://dshb.biology.uiowa.edu/nc82<br>- Thermofisher - https://www.thermofisher.com/antibody/product/<br>- Takara Biomedical Technology - https://www.takarabio.com/products/antibodies-and-elisa/fluorescent-protein-antibodies/red-fluorescent-protein-antibodies<br>- https://www.abcam.com/products |

## Animals and other research organisms

Policy information about studies involving animals; ARRIVE guidelines recommended for reporting animal research, and Sex and Gender in Research

| Laboratory animals | Female Drosophila melanogaster flies 3-8 days post-eclosion (dpe) from the following driver lines were used in this study:<br>Split GAL4 lines (targeting DNp09, aDN2, MDN, aDN1, DNa01, DNa02, DNb02, DNg14, Mute, Web, DNp24, DNg30, DNg11, oviDN, DNb01, DNg16, DNp42).<br>20xUAS-CsChr.mVenus (attP40), 13xLexAop-opGCaMP6s (su(Hw)attP5); DfdLexA / TM6B<br>20xUAS-CsChr.mVenus (attP40), 13xLexAop-opGCaMP6s (su(Hw)attP5); 13xLexAop-CD4-tdTomato (VK00033),<br>DfdLexA / TM6B |
|---|---|

LexOp-myr-TdTomato / CyO; DfdLexA / TM6B
LexOP-H2B::mCherry / CyO; DfdLexA / TM6B
LexAop-GtACR1 / CyO; DfdLexA / TM6
20xUAS-CsChr.[mVenus]attP18; spGAL4-AD; spGAL4-DBD (see Methods)
PR (Phinney Ridge wild type) flies, Canton-S wild type flies

| | |
|---|---|
| Wild animals | No wild animals were used. |
| Reporting on sex | All studies were performed on female flies due to their larger body size. This property facilitates neural data analysis and behavioral quantification. |
| Field-collected samples | No field-collected samples were used |
| Ethics oversight | All experiments were performed in compliance with relevant national (Switzerland) and institutional (EPFL) ethical regulations |

Note that full information on the approval of the study protocol must also be provided in the manuscript.

