## [Peer Review File · Nature]

Manuscript Title: Descending networks transform command signals into population motor control

Reviewer Comments & Author Rebuttals

Reviewer Reports on the Initial Version:

Referees' comments:

Referee #1 (Remarks to the Author):

The finding that the descending neurons that carry signals from the brain to the ventral nerve cord are often connected to each other in the gnathal ganglion complexifies our understanding of how motor programs are governed. The results combine brand new connectome data with technically challenging, large scale functional imaging of neural activity in the neck connective of behaving flies. The quality of the data and significance of the results are high; the conceptual implications are thought-provoking and broadly significant.

In some cases, the interpretation outruns what can be fully supported by the experiments – specifically the requirement for recruitment of the downstream, population DNs and in the distinction between flexible vs stereotyped behaviors. This can be addressed either by some additional experiments, or by text changes moving some of what is now in “results” into the “discussion” section with an expansion of the potential caveats and acknowledgement of space for alternative possibilities.

Summary

Descending neurons (DNs) connect the brain to the ventral nerve cord. Some DNs act in a command-like manner: activation initiates behaviors, ranging from forward walking to grooming to jump-escape. Anatomical analysis of DNs suggested a division between individual/unique ones – where there is only a single bilaterally-symmetric pair – and populations, where there is a group of morphologically similar ones from a common developmental origin that share neurotransmitter identity, overall connectivity, and potentially behavioral functions. The current manuscript attempts to reconcile whether behavior performance commands are conveyed by single neurons or by populations onto the anatomical framework of pairs vs groups by providing evidence that single command-like descending neurons often connect to other descending neurons, linking them to population codes.

The major issues are

1. The effort to show that the “broadcaster” command DN to population DN connections are required for the induced behavior rests on “silencing” the post-synaptic DNs - by decapitation. Removing the head before activating the command-like neuron does stop that DN from activating other DNs in the brain, but does a lot of other things to the animal as well, and these limit its behavioral repertoire. It might be better to test this hypothesis on behaviors that can be more fully performed without the head – leg rubbing, oviposition,...? Normal antennal grooming depends on the sensory signals arriving in the head, forward walking usually occurs in response to visual or

olfactory cues received there, and even postural balance is altered, so there are things the head does that contribute to those behaviors other than house some downstream DN neurons. Is there an alternative way to silence the DN synapses in the brain specifically? The optogenetic effort with *Dfd>GtACR* looked promising. If that produced interfering behavioral phenotypes, what about a more specific test targeting one of the previously identified population DNs? Is there a way to silence the synaptic output of the DNs specifically in the brain while leaving their outputs in the VNC intact? (Experimental request at best, inclusion of other roles for the head/result explanations in text acceptable)

2. Is there a way to image activity in headless flies to assess how descending neuron activity is altered? That activation is still working as expected? Imaging at the most posterior part of the neck connective or in the upper VNC?

3. To make the argument that it is the connectivity pattern that is critical, it would be ideal to compare DNs that evoke similar behaviors but have different degrees of DN-to-DN connectivity. What about aDN1 vs. aDN2, both of which command antennal grooming but show different intra-DN connectivity? What about comparing multiple DNs evoking forward locomotion? (experimental request)

4. The observation that optogenetic activation of command neurons and “spontaneous” performance of the behavior result in different patterns of activity in neck neurons is problematic. If there are many different reasons to walk, reflected in the spontaneous activity, but only one is conveyed by optogenetic activation of this command neuron, where are the other signals conveyed? It is surprising that there is so little activity in the neck connective in the controls (Figure 2B) even though the flies are walking. One would think that the flies decided to walk and activated the appropriate command neurons themselves, with downstream DN connections, even if the experimenter didn't do it with optogenetics. There are actually three conditions – optogenetic activation, spontaneous initiation of behavior, and sensory induction (the most natural?) Please explain or discuss this.

5. What is the purpose of analyzing the activity after spontaneous behaviors? (line 131; 156-157) Quiescent periods? Comparison? Please justify this more clearly. (text change)

6. Do the sparsely connected DNs connect to other DNs in brain regions other than the GNG? It sounded like there are additional brain connections (line 189). What about downstream, in the VNC? Or do they connect indirectly, perhaps with one interneuron intermediary? “single hop”. Would this make them functionally similar to the current broadcaster neurons? The division is logical and appealing, but may not be that absolute. (data analysis and text change)

7. The classification of behaviors as flexible and stereotyped is subjective. Why is backward walking considered more stereotyped? It seems to be rarer, and the connection to the motor neurons that implement it somewhat more direct, but foot placement is still somewhat variable. The best position for grooming along this continuum probably depends on the temporal scale/complexity of the movements considered: leg rubs might be similar each time (stereotyped) but the trajectory of the head sweeps depends on the dust distribution and the alternation and number of head sweeps and

leg rubs in a cycle might be stochastic – is that flexible? (text change to explain the classification decision and available behavioral quantification data as evidence for this judgement call.)

8. Is it possible to activate and image from the same DNs? It would be a nice confirmation that the optogenetic manipulation was inducing neural activity and the GCaMP signal time locked to activation would give landmarks in the neck connective pattern (a way to correlate regions of fluorescence change to anatomical locations of known neurons). If none of the command neuron reagents overlap with Dfd, as stated, perhaps including a UAS-GCaMP or RGecko is an option. While the temporal resolution of GCaMP might not permit, the onset of change in fluorescence could indicate that the “broadcaster” command neuron’s activity precedes that of “receiver populations” and help discriminate which DNs are active because of direct synaptic recruitment rather than as a consequence of the behavior performance, sensory feedback, or other “intentions” of the fly. (experimental request, but expanded discussion acceptable)

9. Figure 6 is confusing. If the DNs make excitatory connections within clusters, how does inhibition between clusters work? Are there additional GABAergic or glutamatergic neurons with outputs from the cluster that receive many inputs? The directed and undirected graphs were hard to parse. Clusters 3 and 9 are both walking – are the left and right? Would it make sense to mirror and then cluster? (Either re-analyze or expand explanation)

The figures are nice. The schematics help connect the functional and anatomical definitions.

The discovery of this framework and then the selection of other DNs along the connectivity continuum is a nice test of the predictions.

Is “Cerebral Ganglia” the best term for the brain regions above the GNG? Is that the Ito standard nomenclature? Otherwise, it might be confusing for readers.

Minor: in the introduction line 60, please clarify meanings of “high- and low- dimensional” in these contexts. In the list of possible functions for the DN-DN connections, what is meant by “gating” (line 209)?

Referee #2 (Remarks to the Author):

The study by Braun and colleagues addresses a highly timely and unanswered question in the neurosciences, especially for the fields of motor control and neuroethology: is there a systematic logic or motif for the descending control of an animal’s behavior by the brain. The authors use the fruit fly as a model organism as it allows for their methodological tour de force used in this hypothesis-driven study. They use two-photon calcium imaging of axons of descending neurons (DN) in the fly’s neck connective during the execution of three motor behaviors, i.e. forward walking, backward walking, and antennal grooming. Furthermore, the authors do a connectome analysis of individual descending neurons (DNs) involved in these behaviors as well as the postsynaptic connectivity of all known DNs in the fruit fly to other DNs. The three behaviors focused on can be elicited by stimulation of three individual DNs (DNp09 for forward walking; aDN2 for antennal

grooming; MDN for backward walking; all existing in 1 or 2 pairs). These DNs are also active during the spontaneous generation of their respective behavior. Using the recently published connectome data for the brain and VNC in the fruit fly the authors show that the number of DNs that are postsynaptic to the three DNs in focus differs, which they relate to the level of flexibility of the behavior that they drive. To extend the conclusions drawn to a more general level of control in the fruit fly the authors show that there exists a continuum of number of postsynaptic DNs for each of the more than 1250 DNs from zero to about 100. They conclude from their study that the number of DNs that are downstream partners of individual command-like DNs depends on the level of flexibility a behavior associated with such a command-like neuron has: DNp09 has the most downstream partners, because forward walking is in comparison the most flexible behavior of the three. aDN2 for antennal grooming and MDN for backward walking come in second and third with respect to the number of postsynaptic DNs and their correlated behavioral flexibility. The authors test their conclusion for two other DNs commanding two further behaviors (DNb02 and DNg14). The individual parts of the study have been conducted thoroughly, which accounts for data collection, evaluation, and statistical analysis as far as the reviewer can judge based on the materials and methods descriptions and the inspection of text and figures.

Such study could be of high general interest, because it is the first account linking connectivity patterns of DNs in the brain with specifics of elicited behaviors commanded by these. Still, I have the following general and specific concerns, questions, and suggestions:

1. Abstract, li 10-17; and subsequent places: The authors rate the studied behaviors, i.e. forward walking, backward walking, and antennal grooming, as inherently and clearly differing in flexibility. However, no argument is given for this classification, which would, however, be of utmost relevance for the conclusions drawn by the authors.

2. Introduction: The three DNs that were selected differ significantly from each other: MDN has been shown to be necessary and sufficient for commanding backward walking (Bidaye et al. 2014), i.e. inactivation/silencing of MDN results in the inability to walk backward. To the knowledge of the reviewer no such proof of necessity as command neuron exists for the other two DNs (Bidaye et al. 2020; Guo et al. 2022; Hampel et al. 2015).

3. One important argument for the conclusions that are drawn here, i.e. that DNs commanding flexible behaviors rely on many other downstream DNs for executing the motor behavior, are based on activating the three DNs in headless flies (Fig. 4 and text). Following decapitation only optogenetic activation of MDNs, which have the fewest direct postsynaptic DN connections among the three DNs that were studied, was still sufficient to drive backward walking, while the other two DNs could no longer initiate the associated behavior. I have difficulties to follow this conclusion: (i) on the methodological level the question arises, on how the authors can be sure that the remaining axons of all three DNs can still be activated. (ii) The fact that headless flies can still walk backwards indicates that ascending information (sensory, for instance) from the VNC is not necessary. This might, however, well be the case of forward walking or antennal grooming: in headless flies not only downstream DNs to DNp09 are missing, but also all ascending information from the VNC is removed. Similarly, in headless flies not only downstream neurons to aDN2 are missing, but also feedback signals from the antennae are missing, potentially relevant for organizing and maintaining this motor behavior. Presently, these concerns are not addressed by the authors.

4. In Fig. 5b and c the authors plot the number of connected DNs to one single DN as a function of numbers of DNs with a given connectivity. It becomes clear that there is a systematic range from very few DNs connecting to up to approx. 100 other DNs to many DNs connecting to fewer and fewer DNs, down to some that connect to no other DNs. The three DNs that were studied here are in the range of up to approx. 30 connected DNs. This is also true for the two additional DNs tested later in the study. Given the conclusion of the authors that the number of connected DNs depends on the flexibility of a given motor behavior the question arises, why the authors did not test for DNs with a high number of postsynaptic partner DNs, i.e. left to the DNs selected in the range of 30 upwards. From their conclusion it can be expected that these DNs upon activation will be able to command highly flexible motor behaviors. Finally, and in general, the notion of the authors to conclude from five DNs that were inspected in detail out of more than 1000 towards being able to prove a general “ground plan of DN connectivity and behavioral flexibility” appears rather far reaching; too far in the opinion of this reviewer.

5. Results, lines 205-211, lines 243-247, lines; and Fig.1a; Discussion, lines 440-442, lines 457-462: the authors reason that DNs commanding more flexible behaviors need further and more DNs to recruit individual aspects of a given behavior. Compelling evidence for this notion would come from silencing some of the downstream DNs of DNp09 or aDN02 and then monitor the motor output generated upon activation of these two DNs. Such experiments were not performed, which renders this conclusion by the authors as not justified, in particular the discussion of behavioral building blocks and motor primitives.

Specific criticisms/suggestions:

Lines 160-161: what is meant by “...non-ethological rather than reflecting a natural process,.....”? please specify

Line 168: “...that populations of GNG-DNs.....” were these the same?

Line 179-180: “...a few command-like neurons are sufficient to drive behaviors but that larger DN populations are active during spontaneous behaviors.” It is not clear, what the difference between both situations is intended to be.

Line 209: “third, additional DNs may ‘gate’ a behavior that is initiated by command-like DNs....” Please explain for clarity.

Line 234-235: “....could still behave.” What is meant by “behave”?

Line 320-333: The authors mainly take feedforward and lateral connections and influences into account, however, not feedback influences between neurons and levels of processing.

Line 457: “these behavioral building blocks” I was not able to find a part in the results section dealing with this topic.

Line 475-480: given that no DNs recruited by the three DNs studied in detail have been tested for their contribution, this paragraph has unfortunately no justification and should be removed.

Fig.5b: it does not become clear, why DNs are separated in two groups “all DNs” and “GNG DNs”

Author Rebuttals to Initial Comments:

Response to Reviewers for Braun et al., 2023 (2023-09-16274A)

We thank the editor and the reviewers for their valuable comments. In addition to addressing their comments, we have also increased the number of biological replicates for control experiments in **Extended Data Figures 1d-f**.

Referee #1 (Remarks to the Author):

The finding that the descending neurons that carry signals from the brain to the ventral nerve cord are often connected to each other in the gnathal ganglion complexifies our understanding of how motor programs are governed. The results combine brand new connectome data with technically challenging, large scale functional imaging of neural activity in the neck connective of behaving flies. The quality of the data and significance of the results are high; the conceptual implications are thought-provoking and broadly significant.

We thank the Referee for their positive assessment of our study.

In some cases, the interpretation outruns what can be fully supported by the experiments – specifically the requirement for recruitment of the downstream, population DNs and in the distinction between flexible vs stereotyped behaviors. This can be addressed either by some additional experiments, or by text changes moving some of what is now in “results” into the “discussion” section with an expansion of the potential caveats and acknowledgement of space for alternative possibilities.

We address the Referee’s comments on limitations below. We appreciate the specifications of how the reviewer suggests individual points be addressed.

Summary

Descending neurons (DNs) connect the brain to the ventral nerve cord. Some DNs act in a command-like manner: activation initiates behaviors, ranging from forward walking to grooming to jump-escape. Anatomical analysis of DNs suggested a division between individual/unique ones – where there is only a single bilaterally-symmetric pair – and populations, where there is a group of morphologically similar ones from a common developmental origin that share neurotransmitter identity, overall connectivity, and potentially behavioral functions. The current manuscript attempts to reconcile whether behavior performance commands are conveyed by single neurons or by populations onto the anatomical framework of pairs vs groups by providing evidence that single command-like descending neurons often connect to other descending neurons, linking them to population codes.

The major issues are

1. The effort to show that the “broadcaster” command DN to population DN connections are required for the induced behavior rests on “silencing” the post-synaptic DNs - by decapitation. Removing the head before activating the command-like neuron does stop that DN from activating other DNs in the brain, but does a lot of other things to the animal as well, and these limit its behavioral repertoire. It might be better to test this hypothesis on behaviors that can be more fully performed without the head – leg rubbing, oviposition,...

We appreciate the suggestion to analyze DNs controlling behaviors that can be performed without the head. Indeed, command-like DNs have been identified for both leg rubbing (DN_{g11} from Guo et al., *Current Biology*, 2022) and oviposition (oviDN_a and oviDN_b from Wang et al., *Nature* 2020). We analyzed DN-DN connectivity for these neuron types and observed that they have only a few downstream synaptic partners (**updated Figure 5c**). There are four oviDN_a neurons, two of them have two downstream DN partners each and two have no downstream DN partners; As well, the two oviDN_b neurons have no downstream DN partners (**new analysis: Extended Data Figure 7a, left**). DN_{g11} comprises six neurons, among which two each have two downstream DN partners, while the others have no DN partners (**new analysis: Extended Data Figure 7b, left**). Thus, neither of these two classes of DNs are ‘broadcasters’ but are rather ‘standalone’ DNs. Therefore, in our model, we predict that they would be able to drive the complete behavioral output (i.e., leg rubbing and abdominal dipping, respectively) even in headless animals. In the revised manuscript, we made this prediction (**updated Figures 5a,c**) and observed that, indeed, both behaviors remain intact in headless animals (**new experiments: Extended Data Figures 7a-b, right**). These results are consistent with our model.

Normal antennal grooming depends on the sensory signals arriving in the head, forward walking usually occurs in response to visual or olfactory cues received there, and even postural balance is altered, so there are things the head does that contribute to those behaviors other than house some downstream DN neurons.

We agree that the brain is responsible for providing feedback during natural behavior. We note, however, that antennal grooming driven by aDN₂ stimulation –as defined by front leg movements towards the front of the head– can proceed without sensory feedback from the antenna. We illustrate this through a set of new experiments, in which we compare antennal grooming upon aDN₂ stimulation with or without the antennae (**new experiments: Reviewer Figure 1**). The message of this experiment is not that the brain is dispensable for natural behaviors but that, because descending neurons normally integrate these sensory signals to direct motor actions, we can focus on the roles of descending neurons in controlling downstream motor circuits and movements by bypassing natural sensory inputs and optogenetically stimulating descending neurons directly.

In the revised manuscript, we clarify this point by stating:

“...we needed to stimulate command-like DNs while preventing the recruitment of additional DN populations. Sensory neurons in the brain are responsible for providing sensory inputs to initiate and regulate natural behaviors –antennal mechanosensing elicits antennal grooming and visual or olfactory input can drive walking– whereas DNs are thought to integrate these signals to drive specific motor actions. In this experiment we aimed to focus solely on the roles of DNs in controlling behavior by bypassing natural sensory inputs to the brain and optogenetically stimulating premotor DN axons in the VNC directly. This would allow us to identify which elements of behavioral kinematics directly and solely result from optogenetic stimulation of command-like DNs alone (i.e., a low-dimensional signal coming from the brain), without also recruiting other DNs in the brain.”

Is there an alternative way to silence the DN synapses in the brain specifically? The optogenetic effort with Dfd>GtACR looked promising. If that produced interfering behavioral phenotypes, what about a more specific test targeting one of the previously identified population DNs? Is there a way to silence the synaptic output of the DNs specifically in the brain while leaving their outputs in the VNC intact? (Experimental request at best, inclusion of other roles for the head/result explanations in text acceptable)

We agree that, ideally, one would be able to silence DN-DN synapses in the brain specifically. Unfortunately, in addition to expressing transgenes in GNG-DNs, Dfd also expresses in other interneurons within the GNG. As well, even if we could genetically isolate DNs in the GNG, to the best of our knowledge there exists no genetic reagent that allows one to selectively silence a neuron's synaptic output in one region (i.e., in the GNG) while retaining its output elsewhere (i.e., in the VNC). As the Referee mentions, we initially attempted to silence a large fraction of GNG neurons optogenetically using GtACR1 but observed confounding anterior grooming behaviors. We illustrate this interfering behavioral phenotype in the revised manuscript (**new analyses: Extended Data Figure 1g**). We now include in the revised main text an explanation of the other roles for the head/brain as we detail in response to the previous point.

2. Is there a way to image activity in headless flies to assess how descending neuron activity is altered? That activation is still working as expected? Imaging at the most posterior part of the neck connective or in the upper VNC?

In the revised manuscript, we performed additional experiments to address this question. We asked whether one can observe calcium signals in MDN and DNp09 axons during optogenetic stimulation in intact and headless animals. Indeed, one can observe calcium signals in intact MDN and DNp09 neurons (**new experiments: Reviewer Figures 2f-g, black**). Importantly, in headless flies, DNp09 stimulation could still yield abdominal contraction (**Reviewer Figure 2h, blue**) and MDN stimulation could still yield backward stepping movements of the hind legs (**Reviewer Figure 2i, blue**). However, in headless animals, although optogenetic stimulation could still drive a behavioral phenotype, calcium signals were no longer apparent in cut DN axons (**Reviewer Figures 2f-g, blue**). These results may be due to the absence of anterograde Ca^{2+} transport in severed axons without their cell-bodies (note that severed DN axons with cell bodies attached still show Ca^{2+} signals in the new Extended Data Figure 3a-d). We are nevertheless certain that we can still optogenetically activate DN axons in headless animals based on numerous observations: (i) MDN stimulation still drives backward walking in headless animals (**Figure 4d**); (ii) although DNp09 and aDN2 do not drive forward walking and anterior grooming in headless animals, these 'broadcaster' DNs still reliably drive abdominal contraction (**Figure 4f**) and front leg movements (**Figure 4g**), respectively in headless animals; and (iii) oviDN, DN_g11, mute, and DN_g14 still drive abdomen dipping, tibia-tarsus flexion, ovipositor extension, and abdomen dipping, respectively (**Extended Data Figure 7**). Thus, we can be sure that DN axons in the VNC are capable by themselves of driving downstream VNC motor circuits.

3. To make the argument that it is the connectivity pattern that is critical, it would be ideal to compare DNs that evoke similar behaviors but have different degrees of DN-to-DN connectivity. What about aDN1 vs. aDN2, both of which command antennal grooming but show different intra-DN connectivity? What about comparing multiple DNs evoking forward locomotion? (experimental request)

We agree with the Referee that this experiment would be ideal. To assess the feasibility of this approach, we first performed connectomics analysis for multiple DNs known to drive the same behavior: either forward walking (DNp09 from Bidaye et al., *Neuron* 2020; BDN2 and oDN1 from Sapkal et al., *bioRxiv* 2023), or antennal grooming (aDN1 and aDN2 from Hampel et al., *Elife* 2015) (**new analyses: Extended Data Figure 9**). These results illustrate two reasons that we unfortunately cannot compare 'broadcaster' and 'standalone' DNs for the same behavior. First, DNs that are known to drive walking (**Extended Data Figure 9a-d**) as well as those known to drive antennal grooming (**Extended Data Figure 9e-g**) all show similar degrees of DN-to-DN connectivity. They are all substantially connected to downstream DNs (i.e., they are all 'broadcasters'). Second, as might be expected from our whole brain connectome cluster analysis (**Figure 6**), DNs that drive similar behaviors are part of the same larger DN networks: (i) oDN1 and BDN2 are reciprocally connected to one another, (ii) DNp09 targets oDN1, and (iii) aDN1 and aDN2 are reciprocally connected to one another.

4. The observation that optogenetic activation of command neurons and "spontaneous" performance of the behavior result in different patterns of activity in neck neurons is problematic. If there are many different reasons to walk, reflected in the spontaneous activity, but only one is conveyed by optogenetic activation of this command neuron, where are the other signals conveyed?

To answer this question, we note that in our previous manuscript examining DN population activity (Aymanns et al., *Elife* 2022) we observed that a large fraction of DNs in the cerebral ganglia (CRG), rather than DNs in the gnathal ganglia (GNG) that were studied in this manuscript, become active during both spontaneous as well as odor-evoked forward walking. Thus, it is likely that spontaneous walking is principally governed by CRG descending neurons. By contrast, DNp09 neurons are thought to mediate courtship-related forward walking because this class of neurons "receives inputs from central courtship-promoting neurons and visual projection neurons, and is necessary for a male to pursue a female during courtship" (from Bidaye et al., *Neuron* 2020 referring to these 'P9' DNs). The possibility that DNp09 might be active only in specific courtship-related behavioral contexts but inactive during spontaneous walking is also supported by the recent electrophysiological finding that DNp09 are not active during spontaneous tethered walking (Yang et al., *bioRxiv* 2023). Thus, it appears that forward walking can be controlled using diverse DNs depending on the behavioral context. We further clarify this point in the Discussion of the revised manuscript.

We state:

"We find that the additional DN activity elicited by DNp09 stimulation is recruited beyond what is normally seen during spontaneous walking. This suggests a distinction between DN populations becoming active during spontaneously generated, sensory-induced, and optogenetically activated walking. We note that we previously observed a large fraction of DNs in the CRG (rather than GNG-DNs recorded in this study) becoming active during spontaneous and odor-evoked forward walking (Aymanns et al., *Elife* 2022). Thus, spontaneously generated and sensory-induced walking may principally be driven by CRG-DNs. For example, DNp09 are thought to mediate courtship-related forward walking (Bidaye et al., *Neuron* 2020). The possibility that DNp09 is active only in specific courtship-related behavioral contexts but inactive during spontaneous walking is supported by recent electrophysiological evidence

(Yang et al., bioRxiv 2023). Thus, it appears that forward walking can be controlled by distinct DNs depending on the context.”

It is surprising that there is so little activity in the neck connective in the controls (Figure 2B) even though the flies are walking. One would think that the flies decided to walk and activated the appropriate command neurons themselves, with downstream DN connections, even if the experimenter didn't do it with optogenetics. There are actually three conditions – optogenetic activation, spontaneous initiation of behavior, and sensory induction (the most natural?) Please explain or discuss this.

We apologize for the confusion. The absence of neural activity in ‘no DNs’ control animals (**Figure 2b, right column**) can be explained by the fact that our $\Delta F/F$ is normalized and compared with the period prior to optogenetic stimulation during which animals were already spontaneously walking. Thus, the increased neural activity seen by DNp09 stimulation represents an additional recruitment of DNs beyond what is seen during spontaneous walking.

5. What is the purpose of analyzing the activity after spontaneous behaviors? (line 131; 156-157) Quiescent periods? Comparison? Please justify this more clearly. (text change)

We apologize for the lack of clarity. Indeed, because flies were quite spontaneously active, we analyzed neural activity following walking periods to increase the number of trials available for trial-averaging. Additionally, in otherwise quiescent control animals laser light exposure often increased animal arousal, resulting in behaviors which drove increases in DN population activity that would obscure our analysis. This was another reason to focus on animals that were already walking prior to optogenetic stimulation. We note, however, that we include examples of data analyzed from trials following resting periods as well (**Supporting Information File 1, pages 6-10**). Importantly, regardless of whether we analyzed trials with pre-stimulus walking or resting, we consistently observed the recruitment of additional DNs upon command-like DN optogenetic stimulation. We now further justify our methodological decision in the Methods section of the revised manuscript.

We state:

“Because flies were quite spontaneously active, analyzing trials for which flies were previously walking increased the data available for trial-averaging. It also allowed us to avoid laser light causing quiescent control animals to behave, obscuring our analyses.”

6. Do the sparsely connected DNs connect to other DNs in brain regions other than the GNG? It sounded like there are additional brain connections (line 189).

We apologize for the confusion. In our original manuscript, we described DN connections to other GNG-DNs (**Fig. 5B, orange ‘GNG DNs’**) but also showed connections to all other DNs including those outside of the GNG (**Fig. 5B, gray ‘All DNs’**). We found that the trend of standalone DNs being sparsely connected is maintained when examining DN-DN connections across all brain regions. We had stated: “we observed a continuum of DN-DN connectivity for DNs across the entire brain (Fig 5b, gray) that was also present in GNG-based DNs specifically Fig. 5b, orange).”

What about downstream, in the VNC?

In our original manuscript, we stated (l425-l427) that “Direct DN-DN connectivity is rare in the VNC (only 2% of all DN post-synaptic partners in the VNC are DNs (Cheong et al., *bioRxiv* 2023)) compared to strong and prominent DN-DN connections within the brain (32% of all DN post-synaptic partners in the brain are DNs).”

In the revised manuscript, we analyzed the driver lines from our study and now state: “Additionally, a preliminary examination of how DNs in this study connect to one another in the male VNC connectome (Cheong et al., *bioRxiv* 2023; Marin et al., *bioRxiv* 2023) shows that DNp09, giant fiber, DNa01, DNa02, DNg11, and DNg14 do not synapse onto other DNs in the VNC. oviDN, MDN and DNb02 connect to one, three, and three DN cell types in the VNC, respectively. Thus, we can conclude that DN-DN interactions in the VNC are rare.”

Or do they connect indirectly, perhaps with one interneuron intermediary? “single hop”. Would this make them functionally similar to the current broadcaster neurons? The division is logical and appealing, but may not be that absolute. (data analysis and text change)

Indeed, this is an important further question: whether standalone DNs can actually recruit many other DNs via a single intermediary neuron. We examined this question in the revised manuscript (**new analyses: Extended Data Figure 5**). We observed that, when accounting for one hop via one other DN (**Extended Data Figures 5b,e,h**) or one hop via one interneuron of any type (**Extended Data Figures 5c,f,i**), the status of all *Drosophila* DNs as well-connected broadcasters versus sparsely connected standalones holds true. Qualitatively, this is true when taking into account all possible connections (**Extended Data Figures 5a-c**), excitatory connections only (**Extended Data Figures 5d-f**), and inhibitory connections only (**Extended Data Figures 5g-i**). Quantitatively, for all intermediary connections the correlation to the ‘zero-hop’ graph with direct connectivity (**Extended Data Figures 5a,d,g**) is high. For example, when taking into account all possible connections, $R^2 = 0.82$ with one DN intermediary and $R^2 = 0.70$ for any intermediary interneuron.

7. The classification of behaviors as flexible and stereotyped is subjective. Why is backward walking considered more stereotyped? It seems to be rarer, and the connection to the motor neurons that implement it somewhat more direct, but foot placement is still somewhat variable. The best position for grooming along this continuum probably depends on the temporal scale/complexity of the movements considered: leg rubs might be similar each time (stereotyped) but the trajectory of the head sweeps depends on the dust distribution and the alternation and number of head sweeps and leg rubs in a cycle might be stochastic – is that flexible? (text change to explain the classification decision and available behavioral quantification data as evidence for this judgement call.)

We agree with the Referee that our original wording of flexible versus stereotyped deserved further attention and justification. Our original wording was based on the consideration that forward walking is a flexible behavior because foot placement is known to be adjusted based on visual feedback (Fujiwara et al., *Neuron* 2022). By contrast, backward walking must be relatively ballistic in nature with no course correction because terrain and obstacles behind the animal are not visible. However, after careful consideration, we decided it would be more appropriate and quantitative to distinguish between behaviors as ‘complex’ versus ‘simple’. Thus, we have removed most references to behavioral ‘flexibility’ in the revised manuscript.

Instead, we define ‘complex’ behaviors as those requiring the control of many active degrees of freedom whereas ‘simple’ behaviors are those using only a few (or only one) actively controlled degrees of freedom. This is in line with the spirit of defining ‘complex’ as composed of two or more parts.

We performed experiments to further justify this classification of DNp09-driven forward walking as complex and MDN-driven backward walking as simple (**new experiments: Extended Data Figure 4 and associated Supplementary Videos 13-16**). To distinguish the number of actively controlled degrees of freedom for these two distinct behaviors, we amputated different leg pairs at the tibia-tarsus joints (to remove mechanical coupling with the spherical treadmill) and optogenetically stimulated either DNp09-driven forward walking or MDN-driven backward walking. We found that DNp09-driven forward walking can be generated in the absence of any pair of legs (i.e., fore, mid, or hindlegs). By contrast, MDN-driven backward walking is abolished upon removal of only the hind pair of legs. Thus, MDN-driven backward walking depends on fewer actively controlled degrees of freedom (one hind leg pair) whereas DNp09-driven forward walking uses more actively controlled degrees of freedom (all three leg pairs). Notably, our other tested ‘simple’ behaviors require the control of only a few degrees of freedom: (i) abdomen dipping (via oviDN and DN_g14 stimulation), (ii) ovipositor extension (via Mute stimulation), and (iii) foreleg rubbing (via DN_g11 stimulation).

We now clarify this justification in our revised manuscript where we state:

“To explore DN control for a wide range of simple to complex actions, we performed this experiment for three sets of command-like DNs driving diverse behaviors. First, we stimulated forward walking, a behavior that is classically considered complex –because it requires the active control of numerous degrees of freedom (DoFs) for six legs– and is also flexible with foot placement adjusted in closed-loop based on visual feedback (Fujiwara et al., *Neuron* 2022)(DNp09-spGAL4, green). Second, we studied anterior grooming, a less complex behavior that requires the control of fewer DoFs–neck, and two forelegs –to switch between grooming one antenna and then the other (Seeds et al., *Elife* 2014)(aDN2-spGAL4, red). Finally, we studied backward walking. Although backward walking appears to be as kinematically complex as forward walking, it has been shown to arise principally from stereotyped oscillations of only the two hindlegs; amputation of these legs suppresses backward walking (Feng et al., *Nature Communications* 2020)(MDN3-spGAL4, cyan). Additionally, unlike forward walking, backward walking is ballistic in nature with no course correction; terrain and obstacles behind the animal are not visible.”

And later we state:

“These two command-like DNs may also be distinguished by the complexity of the behaviors they control: DNp09-driven forward walking may require the coordination of many additional limb DoFs than MDN-driven backward walking which is thought to regulate backward walking through hindleg movements alone (Feng et al., *Nature Communications* 2020). Indeed, through leg amputation experiments, we confirmed that MDN-driven backward walking is generated via active control of only one pair of hind legs: amputation at the tibia-tarsus joint of these legs abolishes the behavior. By contrast, we discovered that DNp09-driven forward walking can be accomplished using any two pairs of legs. Thus, DNp09-driven forward walking employs active control of six legs whereas MDN-driven backward walking only relies on active control of two (hind) legs.”

8. Is it possible to activate and image from the same DNs? It would be a nice confirmation that the optogenetic manipulation was inducing neural activity and the GCaMP signal time locked to activation would give landmarks in the neck connective pattern (a way to correlate regions of fluorescence change to anatomical locations of known neurons). If none of the command neuron reagents overlap with *Dfd*, as stated, perhaps including a *UAS-GCaMP* or *RGecko* is an option.

As previously mentioned in response to point #2, in new experiments we confirmed that MDN and DNp09 neural activity can be observed during optogenetic stimulation of MDNs and DNp09 in intact animals (**Reviewer Figures 2f-g, black traces**). Regarding the possibility of deriving landmarks in the connective, it would indeed be a fantastic challenge for future work to measure DN population activity and identify corresponding known neurons. As the Referee correctly states, in the case of our command-like DNs, we show that these are not among those expressing transgenes in *Dfd-Gal4* (**Extended Data Figures 1a,c**). Thus, we would not be able to identify them during simultaneous optogenetic stimulation and *Dfd* neural imaging. If we added a *UAS-GCaMP* to our *comDN-Gal4>UAS-CsChrimson; Dfd-LexA>LexAOp-GCaMP* flies, we would not be able to use their respective emission spectra to distinguish between command-like neurons directly activated by *CsChrimson* versus other additionally recruited DNs in our neural recordings. On the other hand, adding *UAS-jRGECO* to our transgenic fly might be an interesting future avenue. This is hampered by our current stimulation-imaging approach: a PMT with a red light bandpass filter cannot be used for *jRGECO* imaging because the red laser used for *CsChrimson* stimulation would saturate and could damage the PMT. Instead, in future work, one potential avenue for identifying DN cell types within population imaging data could be by performing post hoc volumetric imaging of the VNC to obtain full DN axonal morphologies. In principle, this might then permit matching between light-level imaging data and connectomics data akin to what has recently been done in the brain (Brezovec et al., *bioRxiv* 2023).

While the temporal resolution of *GCaMP* might not permit, the onset of change in fluorescence could indicate that the “broadcaster” command neuron’s activity precedes that of “receiver populations” and help discriminate which DNs are active because of direct synaptic recruitment rather than as a consequence of the behavior performance, sensory feedback, or other “intentions” of the fly. (experimental request, but expanded discussion acceptable)

As the Referee correctly states, the low temporal resolution of *GCaMP6s* and two-photon microscopy does not permit a discrimination between the relative onset of ‘broadcaster’ command-like DNs and downstream recruited DN populations. However, we agree that it is of the utmost importance to determine if DN-DN recruitment can arise through direct excitation (as expected from brain connectomics data) as opposed to more indirectly via sensory feedback during behavior. We see two major pieces of evidence pointing towards a direct recruitment. First, one strong piece of evidence against sensory feedback-based DN recruitment is that in our DNp09 experiments flies were already walking prior to optogenetic stimulation (**Figure 2b and Extended Data Figure 2**). Thus, any change in DN population activity results only from optogenetic stimulation of DNp09. Second, in the revised manuscript we have performed new experiments to further test the importance of direct DN-DN recruitment (**new experiments: Extended Data Figure 3**). We cut the anterior T1 neuromere of the VNC –and thus ascending sensory feedback to the brain– and examined GNG-DN population activity during DNp09 optogenetic stimulation. There we observed that DNs were

recruited whereas, in control animals with no DNp09 CsChrimson expression, there was no DN recruitment. We note that the spatial pattern of DN recruitment is not as consistent as in intact flies, but attribute this to the invasiveness of severing the VNC. These data strongly support direct DN-DN recruitment in the brain rather than via behavior and sensory feedback.

9. Figure 6 is confusing. If the DNs make excitatory connections within clusters, how does inhibition between clusters work? Are there additional GABAergic or glutamatergic neurons with outputs from the cluster that receive many inputs?

We apologize for the lack of clarity. In the revised manuscript, we now highlight that inhibition between clusters (as shown in **Figure 6b,d,i**) is generated via excitation of inhibitory DNs within each cluster that project to another cluster (**new analyses: Extended Data Figure 10**). We highlight this by illustrating the prominent inhibition between Cluster 2 (take-off and landing) and Cluster 3 (walking). When examining only known DNs, we see that four inhibitory “Web” DNs (**Extended Data Figure 10d, asterisks**) receive strong excitation from within their own Cluster 2 (**Extended Data Figure 10b**) and target numerous (96, 86, 45, and 41) other DNs, especially in Cluster 3 (**Extended Data Figure 10d**). We note that there is likely also inhibition between clusters via local interneurons. However, in-depth analysis of local interneurons in the GNG is beyond the scope of this study.

The directed and undirected graphs were hard to parse.

We apologize for the lack of clarity. Indeed, the short phrase in the original Results section was an encapsulation of a more detailed explanation in the Methods. We have now moved all relevant text to the Methods section “Detection of DN clusters” where the description is more thorough.

Clusters 3 and 9 are both walking – are the left and right? Would it make sense to mirror and then cluster? (Either re-analyze or expand explanation)

Indeed, clusters 3 and 9 both include walking DNs but this is more pronounced for cluster 3 than for cluster 9 which also includes neurons involved in steering during flight (see original submission, Figure 6g). In the original submission (I364-I369) we pointed out that “neurons in the right hemisphere [are] assigned mainly to cluster 3 and those of the left hemisphere [are] assigned to cluster 9. This split of DNs associated with walking between clusters 3 and 9 was due to differences in connectivity between the two brain hemispheres, both in terms of bilateral symmetry in the brain as well as from the localization of the inputs coming from the inferior posterior slope (IPS), superior posterior slope (SPS) and the lateral accessory lobe (LAL) (Fig. 6h, white asterisks).” We hypothesize that cluster 9 may be distinct from cluster 3 in that it may include DNs used for steering both during walking and during flight whereas cluster 3 may be limited to DNs involved in walking. We did not mirror the FAFB whole brain connectome data because it would involve resolving discrepancies between left and right neuron pairs for which corresponding cells often cannot be identified across the brain. As requested by the Referee, we now expand this explanation in the revised manuscript.

In the Results section we state:

“...we hypothesize that cluster 9 may be further distinct from cluster 3 in that it may involve DNs used for steering both during walking and flight whereas cluster 3 may be limited to DNs involved in walking.”

In the Methods section we state:

“We did not mirror connectome data prior to clustering because it requires resolving discrepancies between left and right neuron pairs which, in many cases, are also not identifiable as corresponding cell classes across the brain.”

The figures are nice. The schematics help connect the functional and anatomical definitions.

We thank the reviewer for appreciating our efforts to make the visualizations informative and helpful for the reader.

The discovery of this framework and then the selection of other DNs along the connectivity continuum is a nice test of the predictions.

We thank the reviewer for their positive evaluation of our work.

Is “Cerebral Ganglia” the best term for the brain regions above the GNG? Is that the Ito standard nomenclature? Otherwise, it might be confusing for readers.

Indeed, “Cerebral Ganglia” is the nomenclature suggested by Ito and colleagues to describe, in a neuromere-based manner, brain regions above the gnathal ganglia or GNG (Ito et al., 2014: Fig 2D). However, we note that Cerebral Ganglia is abbreviated as CRG rather than CG. Thus, in the revised manuscript, we have fixed this abbreviation in the text and Figures. To minimize confusion for the reader, in the revised manuscript we now also mention the previous nomenclature of CRG and GNG (supraesophageal ganglion and subesophageal ganglion) in the figure legend of **Figure 1**.

We state:

“Two coarse subdivisions of the adult *Drosophila* brain are the cerebral ganglia (CRG, previously referred to as the supraesophageal ganglion) and gnathal ganglia (GNG, previously referred to as the subesophageal ganglion).”

Minor: in the introduction line 60, please clarify meanings of “high- and low- dimensional” in these contexts.

We apologize for the confusion. In the revised manuscript, we now clarify the meaning of ‘high- and low-dimensional’.

We state:

“All of these observations imply that DN control, rather than being low-dimensional –with each DN pair conveying a simple but reliable drive signal– multiple classes of DNs can work together to control behavior in a population-based manner. In other words, individual DNs might represent single dimensions of a high-dimensional control signal which are combined like building blocks to construct complete behaviors from simpler motor primitives.”

In the list of possible functions for the DN-DN connections, what is meant by “gating” (line 209)?

We apologize for the confusion. In the revised manuscript, we now clarify the meaning of ‘gating’.

We state:

“...additional DNs may ‘gate’ a behavior that is initiated by command-like DNs, for instance, by increasing the excitability of downstream motor circuits to make them more likely to exceed a threshold for firing upon activation by command-like DNs. In this model, they would act as one element of an AND gate, being necessary but not sufficient to drive a motor behavior.”

Referee #2 (Remarks to the Author):

The study by Braun and colleagues addresses a highly timely and unanswered question in the neurosciences, especially for the fields of motor control and neuroethology: is there a systematic logic or motif for the descending control of an animal’s behavior by the brain. The authors use the fruit fly as a model organism as it allows for their methodological tour de force used in this hypothesis-driven study. They use two-photon calcium imaging of axons of descending neurons (DN) in the fly’s neck connective during the execution of three motor behaviors, i.e. forward walking, backward walking, and antennal grooming. Furthermore, the authors do a connectome analysis of individual descending neurons (DNs) involved in these behaviors as well as the postsynaptic connectivity of all known DNs in the fruit fly to other DNs. The three behaviors focused on can be elicited by stimulation of three individual DNs (DNp09 for forward walking; aDN2 for antennal grooming; MDN for backward walking; all existing in 1 or 2 pairs). These DNs are also active during the spontaneous generation of their respective behavior. Using the recently published connectome data for the brain and VNC in the fruit fly the authors show that the number of DNs that are postsynaptic to the three DNs in focus differs, which they relate to the level of flexibility of the behavior that they drive. To extend the conclusions drawn to a more general level of control in the fruit fly the authors show that there exists a continuum of number of postsynaptic DNs for each of the more than 1250 DNs from zero to about 100. They conclude from their study that the number of DNs that are downstream partners of individual command-like DNs depends on the level of flexibility a behavior associated with such a command-like neuron has: DNp09 has the most downstream partners, because forward walking is in comparison the most flexible behavior of the three. aDN2 for antennal grooming and MDN for backward walking come in second and third with respect to the number of postsynaptic DNs and their correlated behavioral flexibility. The authors test their conclusion for two other DNs commanding two further behaviors (DNb02 and DNg14). The individual parts of the study have been conducted thoroughly, which accounts for data collection, evaluation, and statistical analysis as far as the reviewer can judge based on the materials and methods descriptions and the inspection of text and figures.

Such study could be of high general interest, because it is the first account linking connectivity patterns of DNs in the brain with specifics of elicited behaviors commanded by these. Still, I have the following general and specific concerns, questions, and suggestions:

We thank the reviewer for their positive assessment of our work. We also note that, in addition to the two DNs stated by the Referee (DNb02 and DNg14), in the original manuscript we had

tested four more. The full list of six DN types were listed in the original manuscript's Figure 5c: aDN1, DNa02, DNa01, DNb02, mute, and DNg14. Due to lack of space, we only included DNb02 and DNg14 in the original submission's Figure 5d-g. However, data and videos for all six DNs further tested were in the original submission's Extended Data Figure 3.

In the revised manuscript we have performed additional experiments and analyses for DNp42, DNg11, and oviDN. Thus, in total we derived our hypothesis from 3 sets of DNs (DNp09, aDN2, and MDN; see **Figure 5a**) and tested its predictions with a total of 9 additional sets of DNs (DNp42, aDN1, DNa01, DNb02, DNa02, oviDN, DNg11, mute, and DNg14; see **Extended Data Figures 6 and 7**) for a total of 12 sets of DNs investigated with connectomics analysis and optogenetics behavioral experiments.

1. Abstract, li 10-17; and subsequent places: The authors rate the studied behaviors, i.e. forward walking, backward walking, and antennal grooming, as inherently and clearly differing in flexibility. However, no argument is given for this classification, which would, however, be of utmost relevance for the conclusions drawn by the authors.

We agree with the Referee that our original classification of behavioral flexibility deserved further attention and justification. Our original wording was based on the consideration that forward walking is a flexible behavior because foot placement is known to be adjusted based on visual feedback (Fujiwara et al., *Neuron* 2022). By contrast, backward walking must be relatively ballistic in nature with no course correction because terrain and obstacles behind the animal are not visible. However, after careful consideration, we decided it would be more appropriate and quantitative to distinguish between behaviors as 'complex' versus 'simple'. Thus, we have removed most references to behavioral 'flexibility' in the revised manuscript. Instead, we define 'complex' behaviors as those requiring the control of many active degrees of freedom whereas 'simple' behaviors are those using only a few (or only one) actively controlled degrees of freedom. This is in line with the spirit of defining 'complex' as composed of two or more parts.

We performed experiments to further justify this classification of DNp09-driven forward walking as complex and MDN-driven backward walking as simple (**new experiments: Extended Data Figure 4 and associated Supplementary Videos 13-16**). To distinguish the number of actively controlled degrees of freedom for these two distinct behaviors, we amputated different leg pairs at the tibia-tarsus joints (to remove mechanical coupling with the spherical treadmill) and optogenetically stimulated either DNp09-driven forward walking or MDN-driven backward walking. We found that DNp09-driven forward walking can be generated in the absence of any pair of legs (i.e., fore, mid, or hindlegs). By contrast, MDN-driven backward walking is abolished upon removal of only the hind pair of legs. Thus, MDN-driven backward walking depends on fewer actively controlled degrees of freedom (one hind leg pair) whereas DNp09-driven forward walking uses more actively controlled degrees of freedom (all three leg pairs). Notably, our other tested 'simple' behaviors require the control of only a few degrees of freedom: (i) abdomen dipping (via oviDN and DNg14 stimulation), (ii) ovipositor extension (via Mute stimulation), and (iii) foreleg rubbing (via DNg11 stimulation).

We now clarify this justification in our revised manuscript. We state:

"To explore DN control for a wide range of simple to complex actions, we performed this experiment for three sets of command-like DNs driving diverse behaviors. First, we stimulated forward walking, a behavior that is classically considered complex –because it requires the

active control of numerous degrees of freedom (DoFs) for six legs— and is also flexible with foot placement adjusted in closed-loop based on visual feedback (Fujiwara et al., *Neuron* 2022)(DNp09-spGAL4, green). Second, we studied anterior grooming, a less complex behavior that requires the control of fewer DoFs —neck, and two forelegs— to switch between grooming one antenna and then the other (Seeds et al., *Elife* 2014)(aDN2-spGAL4, red). Finally, we studied backward walking. Although backward walking appears to be as kinematically complex as forward walking, it has been shown to arise principally from stereotyped oscillations of only the two hindlegs; amputation of these legs suppresses backward walking (Feng et al., *Nature Communications* 2020)(MDN3-spGAL4, cyan). Additionally, unlike forward walking, backward walking is ballistic in nature with no course correction; terrain and obstacles behind the animal are not visible.”

And later we state:

“These two command-like DNs may also be distinguished by the complexity of the behaviors they control: DNp09-driven forward walking may require the coordination of many additional limb DoFs than MDN-driven backward walking which is thought to regulate backward walking through hindleg movements alone (Feng et al., *Nature Communications* 2020). Indeed, through leg amputation experiments, we confirmed that MDN-driven backward walking is generated via active control of only one pair of hind legs: amputation at the tibia-tarsus joint of these legs abolishes the behavior. By contrast, we discovered that DNp09-driven forward walking can be accomplished using any two pairs of legs. Thus, DNp09-driven forward walking employs active control of six legs whereas MDN-driven backward walking only relies on active control of two (hind) legs.”

2. Introduction: The three DNs that were selected differ significantly from each other: MDN has been shown to be necessary and sufficient for commanding backward walking (Bidaye et al. 2014), i.e. inactivation/silencing of MDN results in the inability to walk backward. To the knowledge of the reviewer no such proof of necessity as command neuron exists for the other two DNs (Bidaye et al. 2020; Guo et al. 2022; Hampel et al. 2015).

We apologize for the confusion. The term ‘command neuron’ was originally proposed for neurons that are both sufficient and necessary for a given behavior and encode the behavior as well (see Kupfermann and Weiss, *Behavioral and Brain Sciences* 1978). More recently, this definition of a command neuron has been criticized as too narrow. The term ‘command-like neuron’ has been used instead for *Drosophila* (e.g., Carreira-Rosario et al., *Elife* 2018) and other species (e.g. Eaton et al. *Progress in Neurobiology* 2001) to describe neurons whose activation is sufficient (but not strictly necessary) to elicit a behavior. We note that it was recently shown that odor-induced backward walking can be generated while silencing MDNs (Israel et al., *Current Biology* 2022). Thus, the ‘command’ nature of MDN for backward walking (i.e., being both necessary and sufficient for backward walking) appears to be more restricted than previously believed (Bidaye et al., *Science* 2014). Therefore, all three classes of DNs that we study in detail can be considered at least sufficient to elicit a behavior and, thus, ‘command-like’. We now clarify this in the Introduction of the revised manuscript.

We state:

“...so-called ‘command-like’ DNs can be sufficient to drive a complete behavior (but not both necessary and sufficient as required to be considered ‘command’ neurons (Kupfermann and Weiss, *Behavioral and Brain Sciences* 1978).”

We also clarify this in the Results section where we state:

“In some contexts (Bidaye et al., *Science* 2014) but not others (Israel et al., *Current Biology* 2022), MDNs have been shown to be both necessary and sufficient to elicit backward walking.”

3. One important argument for the conclusions that are drawn here, i.e. that DNs commanding flexible behaviors rely on many other downstream DNs for executing the motor behavior, are based on activating the three DNs in headless flies (Fig. 4 and text). Following decapitation only optogenetic activation of MDNs, which have the fewest direct postsynaptic DN connections among the three DNs that were studied, was still sufficient to drive backward walking, while the other two DNs could no longer initiate the associated behavior. I have difficulties to follow this conclusion:

(i) on the methodological level the question arises, on how the authors can be sure that the remaining axons of all three DNs can still be activated.

We are certain that DN axons are still activated in headless animals for the following reasons. First, as the Referee acknowledges, the optogenetic activation MDN axons in the VNC of headless flies still triggers full backward walking. Thus, a full behavior can, in principle, be elicited by stimulation of DN axons in a headless animal. Second, stimulation of DNp09 axons triggers abdominal contraction in both intact and headless flies (**Figure 4f**), stimulation of aDN2 axons in headless flies triggers a front leg approach similar to that observed in intact animals (**Figure 4g**). Third, in the original submission we showed that stimulation of Mute axons drives ovipositor extension in headless animals (now **Extended Data Figure 7c**) and stimulation of DN_g14 axons drives abdomen dipping in headless animals (now **Extended Data Figure 7d**). In the revised manuscript, we now also show that stimulation of oviDN axons drives abdomen dipping in headless animals (**new experiments: Extended Data Figure 7a**) and stimulation of DN_g11 axons drives foreleg rubbing in headless animals (**new experiments: Extended Data Figure 7b**). Many of these movements are not observed in headless control animals lacking DN>CsChrimson expression (**Extended Data Figures 6 & 7, right column**). Taken together, these results confirm that optogenetically stimulating DN axons in the VNC is sufficient to drive downstream VNC motor circuits and behavior in headless animals. We have further clarified this early in the Results section of the revised manuscript.

We state:

“These observations confirm that DN axons in the VNC alone are capable of activating downstream VNC motor circuits in headless animals: (i) MDN stimulation in headless flies still triggers full backward walking, (ii) DNp09 stimulation triggers abdominal contraction in both intact and headless flies, and (iii) aDN2 stimulation in headless animals triggers a front leg approach similar to that observed in intact animals. None of these behaviors were observed in headless control animals.”

(ii) The fact that headless flies can still walk backwards indicates that ascending information (sensory, for instance) from the VNC is not necessary. This might, however, well be the case of forward walking or antennal grooming: in headless flies not only downstream DNs to DNp09 are missing, but also all ascending information from the VNC is removed.

Similarly, in headless flies not only downstream neurons to aDN2 are missing, but also feedback signals from the antennae are missing, potentially relevant for organizing and maintaining this motor behavior. Presently, these concerns are not addressed by the authors.

We agree with the Referee that organizing and maintaining motor behaviors may involve sensory feedback. However, we note that headless flies can, in principle, generate something akin to forward walking by activating the appropriate VNC neurons (Harris et al., *Elife* 2015 - see Video 14 for Lineage 12A neuron activation). Additionally, we have performed new experiments to confirm that aDN2-driven antennal grooming can be generated even after the antennae and its sensory apparatus have been removed (**new experiments: Reviewer Figure 1**).

4. In Fig. 5b and c the authors plot the number of connected DNs to one single DN as a function of numbers of DNs with a given connectivity. It becomes clear that there is a systematic range from very few DNs connecting to up to approx. 100 other DNs to many DNs connecting to fewer and fewer DNs, down to some that connect to no other DNs. The three DNs that were studied here are in the range of up to approx. 30 connected DNs. This is also true for the two additional DNs tested later in the study. Given the conclusion of the authors that the number of connected DNs depends on the flexibility of a given motor behavior the question arises, why the authors did not test for DNs with a high number of postsynaptic

partner DNs, i.e. left to the DNs selected in the range of 30 upwards. From their conclusion it can be expected that these DNs upon activation will be able to command highly flexible motor behaviors.

We agree with the Referee that how DNs with the largest number of DN-DN connections regulate animal behavior is an interesting question. We first note that our model does not predict that all DNs with a high number of postsynaptic DNs ('broadcaster') should generate a complex behavior. More precisely, our model predicts that if a broadcaster DN drives a behavior, this behavior (i) would be complex and (ii) lost in headless animals. Not all DNs trigger clear and reliable behaviors (Cande et al., *eLife* 2018) possibly because they are modulatory or sensory in nature.

To address this Referee request to investigate the roles of even more highly interconnected DNs (more than DNp09, aDN2, and MDN), we used specific criteria to identify new sets of DNs with a large number of downstream partners. Namely, we searched for DNs that (i) can be targeted with existing sparse, clean genetic driver lines, (ii) are not associated with flight, and (iii) are excitatory –inhibitory DNs might not drive de novo behaviors but rather inhibit other DNs for action selection. With these criteria we identified DNb01, DNp42, and DNg16. We then performed optogenetic activation experiments on these three sets of neurons to test if they drive reliable behaviors. Of these, we only observed reliable retreat behaviors in DNp42 animals. Consistent with our model's prediction concerning such a highly interconnected 'broadcaster' DN, DNp42 neurons generate retreat in intact but not in headless animals (**new experiments: Extended Data Figure 6a**).

Finally, and in general, the notion of the authors to conclude from five DNs that were inspected in detail out of more than 1000 towards being able to prove a general "ground plan of DN connectivity and behavioral flexibility" appears rather far reaching; too far in the opinion of this reviewer.

We apologize to the Referee for giving the impression of overreaching. We would like to clarify that we derived our hypothesized model relating DN interconnectivity and behavioral complexity from a first group of 3 command-like DNs: DNp09, aDN2, and MDN (**Figures 1-4**). With these results in hand, we then made predictions based on connectivity patterns within the brain for an additional 9 sets of command-like DNs (including 3 new sets of DNs added during revisions): DNp42, aDN1, DNa01, DNb02, DNa02, oviDN, DNg11, mute, and DNg14 (**Figures 5a-c; Extended Data Figures 6 & 7**). To test these predictions, we performed optogenetics experiments and found the results consistent with our model. Thus, in summary, we have tested our model (derived from the original 3 sets of DNs) on 9 additional distinct sets of DNs. Nevertheless, in the revised manuscript we emphasize in the Discussion section that this relationship may not hold for some other kinds of DNs.

We state:

"We note that there is also functional heterogeneity within the large population of ~1300 DNs in the brain. Although many DNs may be excitatory and recruit others to construct behaviors, some DNs may be modulatory, controlling behavioral vigor or persistence. Other DNs are inhibitory, potentially playing a role in action selection. Finally, other DNs may have a role in 'gating' behaviors by increasing the excitability of downstream motor circuits."

5. Results, lines 205-211, lines 243-247, lines; and Fig.1a; Discussion, lines 440-442, lines 457-462: the authors reason that DNs commanding more flexible behaviors need further and more DNs to recruit individual aspects of a given behavior. Compelling evidence for this notion would come from silencing some of the downstream DNs of DNp09 or aDN02 and then monitor the motor output generated upon activation of these two DNs. Such experiments were not performed, which renders this conclusion by the authors as not justified, in particular the discussion of behavioral building blocks and motor primitives.

We agree that activating command-like DNs while silencing downstream DNs would be an excellent future avenue for testing our proposed model for DN behavioral control. Conceptually, this would require that DNs driving motor primitives are not redundant. At the moment whether this is the case remains unknown. As well, technically, one would need to be able to target DNs using both the UAS-GAL4 system for GtACR2 silencing as well as an orthogonal system like the LexA-LexAOp system for CsChrimson activation. To the best of our knowledge the only DN-targeting LexA lines available are those for MDNs and our decapitation experiments reveal that DN recruitment is not necessary for MDN-driven backward walking (**Figure 4d**). However, we note that after our work became available as a preprint, another became available (Yang et al., *bioRxiv* 2023). This other work is complementary in that it provides supporting evidence for DNs as motor primitives. They stimulated and recorded from two DNs downstream of our command-like DNs: DNg13 (in the walking cluster of our original manuscript) and DNa02 (with which we also performed experiments in our original manuscript). Activity of these neurons is correlated with subtle changes in limb movements during walking which they term “gestures” –a term akin to “motor primitives.” As well, another preprint appearing after our own (Sapkal et al., *bioRxiv* 2023) provides evidence that DNs downstream of DNp09 (BDN1, BDN3, BDN4, oDN1, oDN2) may regulate the forward and turning components of DNp09-driven walking. In the Discussion section of our revised manuscript we cite these new studies and discuss the Referee’s suggestion of future experiments that could directly test our notion of behavioral building blocks and motor primitives.

We state:

“Further evidence for a framework in which DNs command complex behaviors by recruiting additional DNs driving simpler motor primitives could come from activating command-like DNs while silencing downstream DNs. For example, DNp09 is connected to and requires the actions of a large number of DNs to drive elaborate movements of the six legs for goal-directed walking during courtship (Bidaye et al., *Neuron* 2020). Both DNp09 (for forward walking) and MDN (for backward walking) synapse upon DNa01 and DNa02, two other DNs involved in turning (Chen et al., *Nature Communications* 2018; Rayshubskiy et al., *bioRxiv* 2020). Thus, one might silence DNa01 or DNa02 to test the prediction that they control specific leg kinematics for turning during both forward or backward walking.”

Specific criticisms/suggestions:

Lines 160-161: what is meant by “...non-ethological rather than reflecting a natural process,.....”? please specify

Here we intend to convey that optogenetic stimulation may drive neurons to a firing rate that is not normally attained during natural activity patterns in response to ethological circumstances. For example, aDN2 may be driven to a specific firing rate or with a specific

temporal activity pattern during natural antennal grooming to remove antennal debris. This may not be reflected by the potentially high firing rate and relatively static temporal activity pattern generated by optogenetic stimulation of the same neurons. An abnormally high firing rate may also abnormally recruit other DNs. We clarify this in the revised manuscript.

We state:

“Recruitment of GNG-DNs by optogenetic stimulation may be non-ethological rather than reflecting what is seen during natural behaviors. For example, aDN2 will have a specific firing rate with a specific temporal activity pattern when animals groom to remove antennal debris. This may not be reflected by the potentially high firing rate and relatively static temporal activity pattern driven by optogenetic stimulation of the same neurons. Thus, an unusually high firing rate may abnormally recruit other DNs. To address this potential concern, we compared the activity of GNG-DN populations in the same animals both during optogenetic stimulation and during the corresponding natural behavior.”

Line 168: “....that populations of GNG-DNs.....” were these the same?

They are indeed the same populations of GNG-DNs (**Extended Data Figure 2**). We compare GNG-DN neural activity for the same individual animal(s) during both (i) optogenetic stimulation of command-like DNs and (ii) natural behaviors.

We state:

“...we compared the activity of GNG-DN populations in the same animals both during optogenetic stimulation and during the corresponding natural behavior.”

Line 179-180: “....a few command-like neurons are sufficient to drive behaviors but that larger DN populations are active during spontaneous behaviors.” It is not clear, what the difference between both situations is intended to be.

We apologize for the confusion. Here we intend to reconcile two observations concerning descending control that at first appear to be incompatible. First, that the activation of only a few command-like DNs is sufficient to drive a full behavior (e.g., in **Figure 2** DNp09 drives forward walking, aDN2 drives grooming, and MDN drives backward walking). Second, when observing the activity of nearly one hundred DNs during natural (i.e., non-optogenetically elicited) spontaneous and odor-driven behaviors we see activity across many dozens of DNs and not simply a few command-like DNs (Aymanns et al., *Elife* 2022). Our data reconcile these two observations by showing that the stimulation of command-like DNs leads to the recruitment of many additional DNs in the GNG in a manner that is similar to DN population activity during natural behaviors. We clarify this in the revised manuscript.

We state:

“Taken together, these data can reconcile two prominent models of descending control that at first appear to be incompatible. First, that the activation of only a few command-like DNs is sufficient to drive a full behavior (e.g., DNp09 drives forward walking, aDN2 drives grooming, and MDN drives backward walking). Second, that many dozens of DNs (not simply a few command-like DNs) are active during natural behaviors (Aymanns et al., 2022). Our data suggest a framework whereby optogenetic stimulation of command-like DNs leads to the recruitment of many additional DNs in a manner that, particularly for backward walking and antennal grooming, is similar to DN population activity during natural behavior.”

Line 209: “third, additional DNs may ‘gate’ a behavior that is initiated by command-like DNs...”
Please explain for clarity.

We apologize for the confusion. In the revised manuscript, we now clarify the meaning of ‘gate’.

We state:

“Third, additional DNs may ‘gate’ a behavior that is initiated by command-like DNs, for instance, by increasing the excitability of downstream motor circuits to make them more likely to exceed a threshold for firing upon activation by command-like DNs. In this model, they would act as one element of an AND gate, being necessary but not sufficient to drive a motor behavior.”

Line 234-235: “...could still behave.” What is meant by “behave”?

We apologize for the lack of clarity. The purpose of this phrase is to convey that headless animals are not just completely immobile. In general, we define a ‘behavior’ as the coordinated movement of multiple degrees-of-freedom in a stereotyped or reproducible manner. To make this more clear, in the revised manuscript we link this sentence with the subsequent sentence.

We state:

“Importantly, these headless animals could still behave (i.e., coordinate the movements of body part DoFs in a stereotyped or reproducible manner): outside of optogenetic stimulation periods we observed episodes of spontaneous grooming in headless flies that resembled those generated by intact animals.”

Line 320-333: The authors mainly take feedforward and lateral connections and influences into account, however, not feedback influences between neurons and levels of processing.

We apologize for the confusion. In this section, we do take into account feedback as well as feedforward and lateral connections. We have now clarified this in the Methods section of the revised manuscript.

We state:

“Here all connections –feedforward, lateral, and feedback– are taken into account.”

Line 457: “these behavioral building blocks” I was not able to find a part in the results section dealing with this topic.

We thank the Referee for pointing this out. Indeed, we did not convey this idea in the Results of our initial submission. We have removed instances of the phrase “behavioral building blocks” from the revised manuscript except to state (metaphorically) in the Introduction that: “individual DNs might represent single dimensions of a high-dimensional control signal which are combined like building blocks to construct complete behaviors from simpler motor primitives.”

Line 475-480: given that no DNs recruited by the three DNs studied in detail have been tested for their contribution, this paragraph has unfortunately no justification and should be removed.

We apologize for the confusion. In the original submission we did indeed study DNs recruited by the three command-like DNs. We clarify this in the revised manuscript where we show the names of DNs directly connected downstream of our three command-like DNs (**new Extended Data Figure 8**). In our original submission Extended Data Figure 3 (now **Extended Data Figure 6 and 7**), we showed the results of investigating two DNs downstream of DNp09: DNa02 (turning DNs) and DNb02 (forward/turning DNs). As well we reported results from aDN1 (antennal grooming DNs) downstream of aDN2. Finally, we reported results from DNa01 (turning DNs) which is downstream of MDN. Because, in the cases of forward and backward locomotion, downstream DNs controlling turning are intuitively related to locomotion, we believe that this Discussion paragraph is justified. Nevertheless, in the revised manuscript, we reword to emphasize the speculative nature of our statement.

We state:

“...a framework in which command-like DNs potentially recruit additional DNs, each of which control simpler motor primitives, **could** provide an effective substrate for the evolutionary modification of behaviors (e.g., diversification of species-specific courtship displays), or the generation of entirely new behaviors through the *de novo* coupling or uncoupling of DNs and motor primitives. Thus, it is likely that a similar mechanism **might** be leveraged for descending control in other species including mammals...”

Fig.5b: it does not become clear, why DNs are separated in two groups “all DNs” and “GNG DNs”

We apologize for the confusion. We make this distinction here for several reasons. First, our functional imaging experiments (**Figure 2**) are restricted to DNs in the GNG (using the Dfd-GAL4 driver line). Second, the GNG has the most varicose DN processes (i.e., putative axon terminals) in the brain. This suggests a high degree of interconnectivity among DNs with projections in this brain region (Namiki et al., *Elife* 2018: Figure 6a). Third, we show both of these two groupings to illustrate that even for the subset of only GNG-DNs, the overall pattern of DN-DN connectivity remains consistent.

Reviewer Reports on the First Revision:

Referees' comments:

Referee #1 (Remarks to the Author):

I am satisfied that the authors have conscientiously address all reviewer concerns and strengthened an already impressive manuscript.

Referee #2 (Remarks to the Author):

The revision of the manuscript by Braun and colleagues has thoroughly addressed most of my initial concerns. Still the following issue remains and needs the attention of and further revision by the authors.

1. (Former point 1 and belonging rebuttle of the authors): The authors have revised the classification of behaviors in focus from more or less flexible to “complex” and “simple”. They do so by inferring the degrees of freedom to be controlled for in a particular motor behavior. While such classification can be regarded as sensible on first sight, it turns out as premature, when looking closer at the behaviors in focus and comparing between them, especially between those which are generated by the appendages, i.e. the legs, as these are forward walking, backward walking and grooming. The reviewer agrees, that it is tempting to combine the qualitative observation of a behavior with the number of DNs the three DNs connect too, to the conclusion drawn by the authors. Still, this cannot be the base of a causal argument. The general problem of the intended classification with respect to these three DNs arises from the fact, that the number of degrees of freedom to be controlled for in the three behaviors, i.e. forward walking, backward walking and grooming, is presently unknown. Just to make this explicit, here is one of the many unresolved issues: what aspects of the two walking behaviors need to be controlled for individually by descending signals from the brain and which aspects arise from neural network action synergies in the VNC? This fact becomes also clear from the additional experiments the authors have conducted on two DNs, i.e. MDN and DNp09. The authors did so to be able to assess “complexity” of the behavior commanded by these two DNs. However, what the additional experiments at best show is, that MDN-induced backward walking relies on presence and action of the hindlegs. It does not show that “backward walking is generated via active control of only one pair of hind legs”. This result might simply arise from a gating function of related neural signals for other control channels, rendering the control of backward walking depending on the presence of the hindlegs. Furthermore this finding does not prove that this behavior is “controlled by fewer degrees of freedom” compared to forward walking as concluded by the authors. Furthermore, while the argument raised by the authors that forward walking is under visual control is correct, the reviewer cannot follow their reasoning that this shows that backward walking is simpler. Please cite a study, which has proven that, for example, no foot placement control or tactile exploration of hindlegs is happening in backward walking fruit flies, or in other insects. The reviewer strongly suggests that the authors refrain from trying to push for the general result of their study, that they here show a continuum within “broadcaster DNs” relating the number of other DNs connected to them to some quantifiable complexity of motor behavior controlled. This study cannot show that the complexity of one of these three motor behaviors correlates with the

number of downstream DNs for each of the DNs studied in detail. However, I agree that the data presented can give rise to such hypothesis, which one may carefully formulate in light of the other DNs investigated.

Author Rebuttals to First Revision:

Response to Reviewers for Braun et al., 2023 (2023-09-16274A)

We thank both Reviewers for valuable comments which have led to substantial improvements in the manuscript.

Referee #1 (Remarks to the Author):

I am satisfied that the authors have conscientiously address all reviewer concerns and strengthened an already impressive manuscript.

We thank the Referee for their positive evaluation of our study.

Referee #2 (Remarks to the Author):

The revision of the manuscript by Braun and colleagues has thoroughly addressed most of my initial concerns.

We thank the Referee for appreciating our efforts to address these initial concerns.

Still the following issue remains and needs the attention of and further revision by the authors.

1. (Former point 1 and belonging rebuttle of the authors): The authors have revised the classification of behaviors in focus from more or less flexible to “complex” and “simple”. They do so by inferring the degrees of freedom to be controlled for in a particular motor behavior. While such classification can be regarded as sensible on first sight, it turns out as premature, when looking closer at the behaviors in focus and comparing between them, especially between those which are generated by the appendages, i.e. the legs, as these are forward walking, backward walking and grooming. The reviewer agrees, that it is tempting to combine the qualitative observation of a behavior with the number of DNs the three DNs connect too, to the conclusion drawn by the authors. Still, this cannot be the base of a causal argument. The general problem of the intended classification with respect to these three DNs arises from the fact, that the number of degrees of freedom to be controlled for in the three behaviors, i.e. forward walking, backward walking and grooming, is presently unknown. Just to make this explicit, here is one of the many unresolved issues: what aspects of the two walking behaviors need to be controlled for individually by descending signals from the brain and which aspects arise from neural network action synergies in the VNC? This fact becomes also clear from the additional experiments the authors have conducted on two DNs, i.e. MDN and DNp09. The authors did so to be able to assess “complexity” of the behavior commanded by these two DNs. However, what the additional experiments at best show is, that MDN-induced backward walking relies on presence and action of the hindlegs. It does not show that “backward walking is generated via active control of only one pair of hind legs”. This result might simply arise from a gating function of related neural signals for other control channels, rendering the control of backward walking depending on the presence of the hindlegs. Furthermore this finding does not prove that this behavior is “controlled by fewer degrees of freedom” compared to forward walking as concluded by the authors. Furthermore, while the argument raised by the authors that forward walking is under visual control is correct, the reviewer cannot follow their reasoning that this shows that backward walking is simpler. Please cite a study, which has proven that, for example, no foot placement control or tactile exploration of hindlegs is

happening in backward walking fruit flies, or in other insects. The reviewer strongly suggests that the authors refrain from trying to push for the general result of their study, that they here show a continuum within “broadcaster DNs” relating the number of other DNs connected to them to some quantifiable complexity of motor behavior controlled. This study cannot show that the complexity of one of these three motor behaviors correlates with the number of downstream DNs for each of the DNs studied in detail. However, I agree that the data presented can give rise to such hypothesis, which one may carefully formulate in light of the other DNs investigated.

We agree with the Reviewer that the state of the art in motor control does not allow us yet to quantify precisely the number of actively controlled degrees of freedom, hindering a rigorous evaluation of ‘behavioral complexity’. Thus, we have accepted the suggestion to only pose this hypothesis in the Discussion and to remove it from the Abstract, Introduction, and Results sections. Instead, we focus on reporting the discovery of functional and topological DN networks as well as their requirement for generating complete behaviors. We have made substantial changes to the text and several figures in our revised manuscript.

These can be found in the “Difference_manuscript.pdf” file at the following line numbers (‘L#’):

Abstract

L7 – Removed reference to degrees-of-freedom.

L6-7; L14-16; L19 – Removed reference to the dichotomy between behaviors that are complex versus simple.

L16-17 – Focused on the requirement for DN recruitment in driving complete behaviors when downstream networks are large.

Introduction

L24-28; L84; L90-97; L100; L105-106 – Removed reference to the dichotomy between behaviors that are complex versus simple.

L92-97; L103-107 – Focused on the requirement for DN recruitment in driving complete behaviors when downstream networks are large.

Results

L125-137; L305-307; L319-327; L330; L332; L337; L381-387 – Removed reference to the dichotomy between behaviors that are complex versus simple, flexibility of behavior, and the control of specific degrees of freedom.

L745-756 – We have edited **Figure 5a**, to focus strictly on the necessity of DN network recruitment for driving behaviors.

Discussion

L485-486; L489-492; L565 – Removed reference to the dichotomy between behaviors that are complex versus simple.

L569-583; L1673-1688; L1892-1918 – Here we mention that command DNs connected to fewer downstream DNs appear to control relatively simpler behaviors (“We speculate that...”). This is accompanied by moving the leg amputation experiments here as well (**Extended Data Figure 10** formerly Extended Data Figure 4; **Supplementary Videos 21-25** formerly Supplementary Videos 13-16) and adding an illustration of this hypothesis (**ED Fig 10a**).